# SEMI-SUPERVISED COMMUNITY DETECTION VIA STRUCTURAL SIMILARITY METRICS

**Yicong Jiang, Tracy Ke**
Department of Statistics
Harvard University
Cambridge, MA 02138, USA

## ABSTRACT

Motivated by social network analysis and network-based recommendation systems, we study a semi-supervised community detection problem in which the objective is to estimate the community label of a new node using the network topology and partially observed community labels of existing nodes. The network is modeled using a degree-corrected stochastic block model, which allows for severe degree heterogeneity and potentially non-assortative communities. We propose an algorithm that computes a 'structural similarity metric' between the new node and each of the $K$ communities by aggregating labeled and unlabeled data. The estimated label of the new node corresponds to the value of $k$ that maximizes this similarity metric. Our method is fast and numerically outperforms existing semi-supervised algorithms. Theoretically, we derive explicit bounds for the misclassification error and show the efficiency of our method by comparing it with an ideal classifier. Our findings highlight, to the best of our knowledge, the first semi-supervised community detection algorithm that offers theoretical guarantees.

## 1 INTRODUCTION

Nowadays, large network data are frequently observed on social media (such as Facebook, Twitter, and LinkedIn), science, and social science. Learning the latent community structure in a network is of particular interest. For example, community analysis is useful in designing recommendation systems (Debnath et al., 2008), measuring scholarly impacts (Ji et al., 2022), and re-constructing pseudo-dynamics in single-cell data (Liu et al., 2018). In this paper, we consider a semi-supervised community detection setting: we are given a symmetric network with $n$ nodes, and denote by $A \in \mathbb{R}^{n \times n}$ the adjacency matrix, where $A_{ij} \in \{0, 1\}$ indicates whether there is an edge between nodes $i$ and $j$. Suppose the nodes partition into $K$ non-overlapping communities $\mathcal{C}_1, \mathcal{C}_2, \ldots, \mathcal{C}_K$. For a subset $\mathcal{L} \subset \{1, 2, \ldots, n\}$, we observe the true community label $y_i \in \{1, 2, \ldots, K\}$ for each $i \in \mathcal{L}$. Write $m = |\mathcal{L}|$ and $Y_{\mathcal{L}} = (y_i)_{i \in \mathcal{L}}$. In this context, there are two related semi-supervised community detection problems: (i) *in-sample classification*, where the goal is to classify all the existing unlabeled nodes; (ii) *prediction*, where the goal is to classify a new node joining the network. Notably, the in-sample classification problem can be easily reduced to prediction problem: we can successively single out each existing unlabeled node, regard it as the "new node", and then predict its label by applying an algorithms for the prediction problem. Hence, for most of the paper, we focus on the prediction problem and defer the study of in-sample classification to Section 3. In the *prediction* problem, let $X \in \{0, 1\}^n$ denote the vector consisting of edges between the new node and each of the existing nodes. Given $(A, Y_{\mathcal{L}}, X)$, our goal is to estimate the community label of the new node.

This problem has multiple applications. Consider the news suggestion or online advertising push for a new Facebook user (Shapira et al., 2013). Given a big Facebook network of existing users, for a small fraction of nodes (e.g., active users), we may have good information about the communities to which they belong, whereas for the majority of users, we just observe who they link to. We are interested in estimating the community label of the new user in order to personalize news or ad recommendations. For another example, in a co-citation network of researchers (Ji et al., 2022), each community might be interpreted as a group of researchers working on the same research area. We frequently have a clear understanding of the research areas of some authors (e.g., senior authors), and we intend to use this knowledge to determine the community to which a new node (e.g., a junior author) belongs.

The statistical literature on community detection has mainly focused on the *unsupervised* setting (Bickel & Chen, 2009; Rohe et al., 2011; Jin, 2015; Gao et al., 2018; Li et al., 2021). The *semi-supervised* setting is less studied. Leng & Ma (2019) offers a comprehensive literature review of semi-supervised community detection algorithms. Liu et al. (2014) and Ji et al. (2016) derive systems of linear equations for the community labels through physics theory, and predict the labels by solving those equations. Zhou et al. (2018) leverages on the belief function to propagate labels across the network, so that one can estimate the label of a node through its belief. Betzel et al. (2018) extracts several patterns in size and structural composition across the known communities and search for similar patterns in the graph. Yang et al. (2015) unifies a number of different community detection algorithms based on non-negative matrix factorization or spectral clustering under the unsupervised setting, and fits them into the semi-supervised scenario by adding various regularization terms to encourage the estimated labels for nodes in $\mathcal{L}$ to match with the clustering behavior of their observed labels. However, the existing methods still face challenges. First, many of them employ the heuristic that a node tends to have more edges with nodes in the same community than those in other communities. This is true only when communities are *assortative*. But non-assortative communities are also seen in real networks (Goldenberg et al., 2010; Betzel et al., 2018); for instance, Facebook users sharing similar restaurant preferences are not necessarily friends of each other. Second, real networks often have severe degree heterogeneity (i.e., the degrees of some nodes can be many times larger than the degrees of other nodes), but most semi-supervised community detection algorithms do not handle degree heterogeneity. Third, the optimization-based algorithms (Yang et al., 2015) solve non-convex problems and face the issue of local minima. Last, to our best knowledge, none of the existing methods have theoretical guarantees.

Attributed network clustering is a problem related to community detection, for which many algorithms have been developed (please see Chunaev et al. (2019) for a nice survey). The graph neural networks (GNN) reported great successes in attributed network clustering. Kipf & Welling (2016) proposes a graph convolutional network (GCN) approach to semi-supervised community detection, and Jin et al. (2019) combines GNN with the Markov random field to predict node labels. However, GNN is designed for the setting where each node has a large number of attributes and these attributes contain rich information of community labels. The key question in the GNN research is how to utilize the graph to better propagate messages. In contrast, we are interested in the scenario where it is infeasible or costly to collect node attributes. For instance, it is easy to construct a co-authorship network from bibtex files, but collecting features of authors is much harder. Additionally, a number of benchmark network datasets do not have attributes (e.g. Caltech (Red et al., 2011; Traud et al., 2012), Simmons (Red et al., 2011; Traud et al., 2012) , and Polblogs (Adamic & Glance, 2005)). It is unclear how to implement GNN on these data sets. In Section 4, we briefly study the performance of GNN with self-created nodal features from 1-hop representation, graph topology and node embedding. Our experiments indicate that GNN is often not suitable for the case of no node attributes.

We propose a new algorithm for semi-supervised community detection to address the limitations of existing methods. We adopt the DCBM model (Karrer & Newman, 2011) for networks, which models degree heterogeneity and allows for both assortative and non-assortative communities. Inspired by the viewpoint of Goldenberg et al. (2010) that a 'community' is a group of 'structurally equivalent' nodes, we design a *structural similar metric* between the new node and each of the $K$ communities. This metric aggregates information in both labeled and unlabeled nodes. We then estimate the community label of the new node by the $k$ that maximizes this similarity metric. Our method is easy to implement, computationally fast, and compares favorably with other methods in numerical experiments. In theory, we derive explicit bounds for the misclassification probability of our method under the DCBM model. We also study the efficiency of our method by comparing its misclassification probability with that of an ideal classifier having access to the community labels of all nodes.

## 2 SEMI-SUPERVISED COMMUNITY DETECTION

Recall that $A$ is the $n \times n$ adjacency matrix on the existing nodes and $Y_{\mathcal{L}}$ contains the community labels of nodes in $\mathcal{L}$. Write $[n] = \{1, 2, \ldots, n\}$ and let $\mathcal{U} = [n] \setminus \mathcal{L}$ denote the set of unlabeled nodes. We index the new node by $n + 1$ and let $X \in \mathbb{R}^n$ be the binary vector consisting of the edges between the new node and existing nodes. Denote by $\bar{A}$ the adjacency matrix for the network of $(n + 1)$ nodes.

**2.1 The DCBM model and structural equivalence of communities** We model $\bar{A}$ with the degree-corrected block model (DCBM) (Karrer & Newman, 2011). Define a $K$-dimensional membership

matrix $\pi_i \in \{e_1, e_2, \ldots, e_K\}$, where $e_k$'s are the standard basis vectors of $\mathbb{R}^K$. We encode the community labels by $\pi_i$, where $\pi_i = e_k$ if and only if $y_i = k$. For a symmetric nonnegative matrix $P \in \mathbb{R}^{K \times K}$ and a degree parameter $\theta_i \in (0, 1]$ for each node $i$, we assume that the upper triangle of $\bar{A}$ contains independent Bernoulli variables, where

$$\mathbb{P}(\bar{A}_{ij} = 1) = \theta_i \theta_j \cdot \pi_i' P \pi_j, \qquad \text{for all } 1 \leq i \neq j \leq n+1. \tag{1}$$

When $\theta_i$ are equal, the DCBM model reduces to the stochastic block model (SBM). Compared with SBM, DCBM is more flexible as it accommodates degree heterogeneity. For a matrix $M$ or a vector $v$, let $\mathrm{diag}(M)$ and $\mathrm{diag}(v)$ denote the diagonal matrices whose diagonals are from the diagonal of $M$ or the vector $v$, respectively. Write $\theta = (\theta_1, \theta_2, \ldots, \theta_{n+1})'$, $\Theta = \mathrm{diag}(\theta)$, and $\Pi = [\pi_1, \pi_2, \ldots, \pi_{n+1}]' \in \mathbb{R}^{n \times K}$. Model (1) yields that

$$\bar{A} = \Omega - \mathrm{diag}(\Omega) + W, \qquad \text{where } \Omega = \Theta \Pi P \Pi' \Theta \text{ and } \bar{W} = \bar{A} - \mathbb{E}\bar{A}. \tag{2}$$

Here, $\Omega$ is a low-rank matrix that captures the 'signal', $W$ is a generalized Wigner matrix that captures 'noise', and $\mathrm{diag}(\Omega)$ yields a bias to the 'signal' but its effect is usually negligible.

The DCBM belongs to the family of block models for networks. In block models, it is not necessarily true that the edge densities within a community are higher than those between different communities. Such communities are called assortative communities. However, non-assortative communities also appear in many real networks (Goldenberg et al., 2010; Betzel et al., 2018). For instance, in news and ad recommendation, we are interested in identifying a group of users who have similar behaviors, but they may not be densely connected to each other. Goldenberg et al. (2010) introduced an intuitive notion of *structural equivalence* - two nodes are structurally equivalent if their connectivity with similar nodes is similar. They argued that a 'community' in block models is a group of structurally equivalent nodes. This way of defining communities is more general than assortative communities.

We introduce a rigorous description of structural equivalence in the DCBM model. For two vectors $u$ and $v$, define $\psi(u, v) = \arccos\langle \frac{u}{\|u\|}, \frac{v}{\|v\|} \rangle$, which is the angle between these two vectors. Let $\bar{A}_i$ be the $i$th column of $\bar{A}$. This vector describes the 'behavior' of node $i$ in the network. Recall that $\Omega$ is as in (2). When the signal-to-noise ratio is sufficiently large, $\bar{A}_i \approx \Omega_i$, where $\Omega_i$ is the $i$th column of $\Omega$. We approximate the angle between $\bar{A}_i$ and $\bar{A}_j$ by the angle between $\Omega_i$ and $\Omega_j$. By DCBM model, for a node $i$ in community $k$, $\Omega_i = \theta_i \Theta \Pi P e_k$, where $e_k$ is the $k$th standard basis of $\mathbb{R}^K$. It follows that for $i \in \mathcal{C}_k$ and $j \in \mathcal{C}_\ell$, the degree parameters $\theta_i$ and $\theta_j$ cancel out in our structural similarity:

$$\cos \psi(\Omega_i, \Omega_j) = \frac{\langle \theta_i \Theta \Pi P e_k, \theta_j \Theta \Pi P e_\ell \rangle}{\|\theta_i \Theta \Pi P e_k\| \cdot \|\theta_j \Theta \Pi P e_\ell\|} = \frac{M_{k\ell}}{\sqrt{M_{kk} M_{\ell\ell}}}, \quad \text{with } M := P \Pi' \Theta^2 \Pi P. \tag{3}$$

It is seen that $\cos \psi(\Omega_i, \Omega_j)$ does not depend on the degree parameters of nodes and is solely determined by community membership. When $k = \ell$ (i.e., $i$ and $j$ are in the same community), $\cos \psi(\Omega_i, \Omega_j) = 1$, which means the angle between these two vectors is zero. When $k \neq \ell$, as long as $P$ is non-singular and $\Pi$ has a full column rank, $M$ is a positive-definite matrix. It follows that $\cos \psi(\Omega_i, \Omega_j) < 1$ and that the angle between $\Omega_i$ and $\Omega_j$ is nonzero.

**Example 1.** *Suppose $K = 2$, $P \in \mathbb{R}^{2 \times 2}$ is such that the diagonal entries are 1 and off-diagonal entries are $b$, for some $b > 0$ and $b \neq 1$, and $\max_i \{\theta_i\} < \min\{1/b, 1\}$ (to guarantee that all entries of $\Omega$ are smaller than 1). For simplicity, we assume $\sum_{i \in \mathcal{C}_1} \theta_i^2 = \sum_{i \in \mathcal{C}_2} \theta_i^2$. It can be shown that $M$ is proportional to the matrix whose diagonal entries are $(1 + b^2)$ and off-diagonal entries are $2b$. When $b < 1$, the communities are assortative, and when $b > 1$, the communities are non-assortative. However, regardless of the value of $b$, the off-diagonal entries of $M$ are always strictly smaller than the diagonal entries, so that $\cos \psi(\Omega_i, \Omega_j) < 1$, for nodes in distinct communities.*

**2.2 Semi-supervised community detection** Inspired by (3), we propose assigning a community label to the new node based on its 'similarity' to those labeled nodes. For each $1 \leq k \leq K$, assume that $\mathcal{L} \cap \mathcal{C}_k \neq \emptyset$ and define a vector $A^{(k)} \in \mathbb{R}^n$ by $A_j^{(k)} = \sum_{i \in \mathcal{L} \cap \mathcal{C}_k} A_{ij}$, for $1 \leq j \leq n$. The vector $A^{(k)}$ describes the 'aggregated behavior' of all labeled nodes in community $k$. Recall that $X \in \mathbb{R}^n$ contains the edges between the new node and all the existing nodes. We can estimate the community label of the new node by

$$\hat{y} = \arg \min_{1 \leq k \leq K} \psi(A^{(k)}, X). \tag{4}$$

We call (4) the *AngleMin* estimate. Note that each $A^{(k)}$ is an $n$-dimensional vector, the construction of which uses both $A_{\mathcal{LL}}$ and $A_{\mathcal{LU}}$. Therefore, $A^{(k)}$ aggregates information from both labeled and unlabeled nodes, and so AngleMin is indeed a semi-supervised approach.

The estimate in (4) still has space to improve. First, $A^{(k)}$ and $X$ are high-dimensional random vectors, each entry of which is a sum of independent Bernoulli variables. When the network is very sparse or communities are heavily imbalanced in size or degree, the large-deviation bound for $\psi(A^{(k)}, X)$ can be unsatisfactory. Second, recall that our observed data include $A$ and $X$. Denote by $A_{\mathcal{LL}}$ the submatrix of $A$ restricted on $\mathcal{L} \times \mathcal{L}$ and $X_{\mathcal{L}}$ the subvector of $X$ restricted on $\mathcal{L}$; other notations are similar. In (4), only $(A_{\mathcal{LL}}, A_{\mathcal{LU}}, X)$ are used, but the information in $A_{\mathcal{UU}}$ is wasted. We now propose a variant of (4). For any vector $x \in \mathbb{R}^n$, let $x_{\mathcal{L}}$ and $x_{\mathcal{U}}$ be the sub-vectors restricted to indices in $\mathcal{L}$ and $\mathcal{U}$, respectively. Let $\mathbf{1}_{(k)}$ denote the $|\mathcal{L}|$-dimensional vector indicating whether each labeled node is in community $k$. Given any $|\mathcal{U}| \times K$ matrix $H = [h_1, h_2, \ldots, h_K]$, define

$$f(x; H) = [x'_{\mathcal{L}} \mathbf{1}_{(1)}, \ \ldots, \ x'_{\mathcal{L}} \mathbf{1}_{(k)}, \ x'_{\mathcal{U}} h_1, \ldots, x'_{\mathcal{U}} h_K]' \ \in \ \mathbb{R}^{2K}. \tag{5}$$

The mapping $f(\cdot; H)$ creates a low-dimensional projection of $x$. Suppose we now apply this mapping to $A^{(k)}$. In the projected vector, each entry is a weighted sum of a large number of entries of $A^{(k)}$. Since $A^{(k)}$ contains independent entries, it follows from large-deviation inequalities that each entry of $f(A^{(k)}, H)$ has a nice asymptotic tail behavior. This resolves the first issue above. We then modify the AngleMin estimate in (4) to the following estimate, which we call (3):[1]

$$\hat{y}(H) = \arg \min_{1 \le k \le K} \psi\big(f(A^{(k)}; H), \ f(X; H)\big). \tag{6}$$

AngleMin+ requires an input of $H$. Our theory suggests that $H$ has to satisfy two conditions: (a) The spectral norm of $H'H$ is $O(|\mathcal{U}|)$. In fact, given any $H$, we can always multiply it by a scalar so that $\|H'H\|$ is at the order of $|\mathcal{U}|$. Hence, this condition says that the scaling of $H$ should be properly set to balance the contributions from labeled and unlabeled nodes. (b) The minimum singular value of $H'\Theta_{\mathcal{UU}}\Pi_{\mathcal{U}}$ has to be at least a constant times $\|H\|\|\Theta_{\mathcal{UU}}\Pi_{\mathcal{U}}\|$, where $\Theta_{\mathcal{UU}}$ is the submatrix of $\Theta$ restricted to the $(\mathcal{U}, \mathcal{U})$ block and $\Pi_{\mathcal{U}}$ is the sub-matrix of $\Pi$ restricted to the rows in $\mathcal{U}$. This condition prevents the columns of $H$ from being orthogonal to the columns of $\Theta_{\mathcal{UU}}\Pi_{\mathcal{U}}$, and it guarantees that the last $K$ entries of $f(x; H)$ retain enough information of the unlabeled nodes.

We construct a data-driven $H$ from $\mathcal{A}_{\mathcal{UU}}$, by taking advantage of the existing unsupervised community detection algorithms such as Gao et al. (2018); Jin et al. (2021). Let $\hat{\Pi}_{\mathcal{U}} = [\hat{\pi}_i]_{i \in \mathcal{U}}$ be the community labels obtained by applying a community detection algorithm on the sub-network restricted to unlabeled nodes, where $\hat{\pi}_i = e_k$ if and only if node $k$ is clustered to community $k$. We propose using

$$H = \hat{\Pi}_{\mathcal{U}}. \tag{7}$$

This choice of $H$ always satisfies the aforementioned condition (a). Furthermore, under mild regularity conditions, as long as the clustering error fraction is bounded by a constant, this $H$ also satisfies the aforementioned condition (b). We note that the information in $\mathcal{A}_{\mathcal{UU}}$ has been absorbed into $H$, so it resolves the second issue above. Combining (7) with (3) gives a two-stage algorithm for estimating $y$.

**Remark 1**: A nice property of AngleMin+ is that it tolerates an arbitrary permutation of communities in $\hat{\Pi}_{\mathcal{U}}$. In other words, the communities output by the unsupervised community detection algorithm do not need to have a one-to-one correspondence with the communities on the labeled nodes. To see the reason, we consider an arbitrary permutation of columns of $\hat{\Pi}_{\mathcal{U}}$. By (12), this yields a permutation of the last $K$ entries of $f(x; H)$, simultaneously for all $x$. However, the angle between $f(A^{(k)}; H)$ and $f(X; H))$ is still the same, and so $\hat{y}(H)$ is unchanged. This property brings a lot of practical conveniences. When $K$ is large or the signals are weak, it is challenging (both computationally and statistically) to match the communities in $\hat{\Pi}_{\mathcal{U}}$ with those in $\Pi_{\mathcal{L}}$. Our method avoids this issue.

**Remark 2**: AngleMin+ is flexible to accommodate other choices of $H$. Some unsupervised community detection algorithms provide both $\hat{\Pi}_{\mathcal{U}}$ and $\hat{\Theta}_{\mathcal{UU}}$ (Jin et al., 2022). We may use $H \propto \hat{\Theta}_{\mathcal{UU}}\hat{\Pi}_{\mathcal{U}}$,

---

[1]In AngleMin+, $H$ serves to reduce noise. For example, let $X, Y \in \mathbb{R}^{2m}$ be two random Bernoulli vectors, where $\mathbb{E}X = \mathbb{E}Y = (.1, \ldots, .1, .4, \ldots, .4)'$. As $m \to \infty$, it can be shown that $\psi(X, Y) \to 0.34 \neq 1$ almost surely. If we project $X$ and $Y$ into $\mathbb{R}^2$ by summing the first $m$ coordinates and last $m$ coordinates separately, then as $m \to \infty$, $\psi(X, Y) \to 1$ almost surely.

(subject to a re-scaling to satisfy the aforementioned condition (a)). This $H$ down-weights the contribution of low-degree unlabeled nodes in the last $K$ entries of (12). This is beneficial if the signals are weak and the degree heterogeneity is severe. Another choice is $H \propto \hat{\Xi}_{(\mathcal{U})} \hat{\Lambda}_{(\mathcal{U})}^{-1}$, where $\hat{\Lambda}_{\mathcal{U}}$ is a diagonal matrix containing the $K$ largest eigenvalues (in magnitude) of $A_{\mathcal{U}\mathcal{U}}$ and $\hat{\Xi}_{(\mathcal{U})}$ is the associated matrix of eigenvectors. For this $H$, we do not even need to perform any community detection algorithm on $\mathcal{A}_{\mathcal{U}\mathcal{U}}$. We may also use spectral embedding (Rubin-Delanchy et al., 2017).

**Remark 3**: The local refinement algorithm (Gao et al., 2018) may be adapted to the semi-supervised setting, but it requires prior knowledge on assortativity or dis-assortativity and a strong balance condition on the average degrees of communities. When these conditions are not satisfied, we can construct examples where the error rate of AngleMin+ is $o(1)$ but the error rate of local refinement is 0.5. See Section C.

**2.3 The choice of the unsupervised community detection algorithm**  We discuss how to obtain $\hat{\Pi}_{\mathcal{U}}$. In the statistical literature, there are several approaches to unsupervised community detection. The first is modularity maximization (Girvan & Newman, 2002). It exhaustively searches for all cluster assignments and selects the one that maximizes an empirical modularity function. The second is spectral clustering (Jin, 2015). It applies k-means clustering to rows of the matrix consisting of empirical eigenvectors. Other methods include post-processing the output of spectral clustering by majority vote (Gao et al., 2018). Not every method deals with degree heterogeneity and non-assortative communities as in the DCBM model. We use a recent spectral algorithm SCORE+ (Jin et al., 2021), which allows for both severe degree heterogeneity and non-assortative communities.

**SCORE+**: We tentatively write $A_{\mathcal{U}\mathcal{U}}=A$ and $|\mathcal{U}|=n$ and assume the network (on unlabeled nodes) is connected (otherwise consider its giant component). SCORE+ first computes $L=D_\tau^{-1/2} A D_\tau^{-1/2}$, where $D_\tau=\mathrm{diag}(d_1,\ldots,d_n)+0.1 d_{\max} I_n$, and $d_i$ is degree of node $i$. Let $\hat{\lambda}_k$ be the $k$th eigenvalue (in magnitude) of $L$ and let $\hat{\xi}_k$ be the associated eigenvector. Let $r=K$ or $r=K+1$ (see Jin et al. (2021) for details). Let $\hat{R} \in \mathbb{R}^{n \times (r-1)}$ by $\hat{R}_{ik} = (\hat{\lambda}_{k+1}/\hat{\lambda}_1) \cdot [\hat{\xi}_{k+1}(i)/\hat{\xi}_1(i)]$. Run k-means on rows of $\hat{R}$.

## 3 Theoretical properties

We assume that the observed adjacency matrix $\bar{A}$ follows the DCBM model in (1)-(2). From now on, let $\theta_*$ denote the degree parameter of the new node $n+1$. Suppose $k^* \in \{1, 2, \ldots, K\}$ is its true community label, and the corresponding $K$-dimensional membership vector is $\pi^* = e_{k^*}$. In (2), $\theta$ and $P$ are not identifiable. To have identifiability, we assume that all diagonal entries of $P$ are equal to 1 (if this is not true, we replace $P$ by $[\mathrm{diag}(P)]^{-\frac{1}{2}} P [\mathrm{diag}(P)]^{-\frac{1}{2}}$ and each $\theta_i$ in community $k$ by $\theta_i \sqrt{P_{kk}}$, while keeping $\Omega = \Theta\Pi P\Pi'\Theta$ unchanged). In the asymptotic framework, we fix $K$ and assume $n \to \infty$. We need some regularity conditions. For any symmetric matrix $B$, let $\|B\|_{\max}$ denote its entry-wise maximum norm and $\lambda_{\min}(B)$ denote its minimum eigenvalue (in magnitude). We assume for a constant $C_1 > 0$ and a positive sequence $\beta_n$ (which may tend to 0),

$$\|P\|_{\max} \leq C_1, \qquad |\lambda_{\min}(P)| \geq \beta_n. \tag{8}$$

For $1 \leq k \leq K$, let $\theta^{(k)} \in \mathbb{R}^n$ be the vector with $\theta_i^{(k)} = \theta_i \cdot 1\{i \in \mathcal{C}_k\}$, and let $\theta_{\mathcal{L}}^{(k)}$ and $\theta_{\mathcal{U}}^{(k)}$ be the sub-vectors restricted to indices in $\mathcal{L}$ and $\mathcal{U}$, respectively. We assume for a constant $C_2 > 0$ and a properly small constant $c_3 > 0$,

$$\max_k \|\theta^{(k)}\|_1 \leq C_2 \min_k \|\theta^{(k)}\|_1, \qquad \|\theta_{\mathcal{L}}^{(k)}\|^2 \leq c_3 \beta_n \|\theta_{\mathcal{L}}^{(k)}\|_1 \|\theta\|_1, \ \text{ for all } 1 \leq k \leq K. \tag{9}$$

These conditions are mild. Consider (8). For identifiability, $P$ is already scaled to make $P_{kk} = 1$ for all $k$. It is thus a mild condition to assume $\|P\|_{\max} \leq C_1$. The condition of $|\lambda_{\min}(P)| \geq \beta_n$ is also mild, because we allow $\beta_n \to 0$. Here, $\beta_n$ captures the 'dissimilarity' of communities. To see this, consider a special $P$ where the diagonals are 1 and the off-diagonals are all equal to $b$; in this example, $|1 - b|$ captures the difference of within-community connectivity and between-community connectivity, and it can be shown that $|\lambda_{\min}(P)| = |1 - b|$. Consider (9). The first condition requires that the total degree in different communities are balanced, which is mild. The second condition is about degree heterogeneity. Let $\theta_{\max}$ and $\bar{\theta}$ be the maximum and average of $\theta_i$, respectively. In the second inequality of (9), the left hand side is $O(n^{-1}\theta_{\max}/\bar{\theta})$, so this condition is satisfied as long as $\theta_{\max}/\bar{\theta} = O(n\beta_n)$. This is a very mild requirement.

**3.1 The misclassification error of AngleMin+** For any $|\mathcal{U}| \times K$ matrix $H$, let $\hat{\psi}_k(H) = \psi(f(A^{(k)}; H), f(X; H))$ be as in (3). AngleMin+ estimates the community label to the new node by finding the minimum of $\hat{\psi}_1(H), \ldots, \hat{\psi}_K(H)$, with $H = \hat{\Pi}_{\mathcal{U}}$. We first introduce a counterpart of $\hat{\psi}_k(H)$. Recall that $\Omega$ is as in (2), which is the 'signal' matrix. Let $\Omega^{(k)} \in \mathbb{R}^n$ by $\Omega_j^{(k)} = \sum_{i \in \mathcal{L} \cap \mathcal{C}_k} \Omega_{ij}$, for $1 \leq j \leq n$, and define

$$\psi_k(H) = \psi\big(f(\Omega^{(k)}; H), \ f(\mathbb{E}X; H)\big), \qquad \text{for } 1 \leq k \leq K. \tag{10}$$

The next lemma gives the explicit expression of $\psi_k(H)$ for an arbitrary $H$.

**Lemma 1.** *Consider the DCBM model where (8)-(9) are satisfied. We define three $K \times K$ matrices: $G_{\mathcal{L}\mathcal{L}} = \Pi'_{\mathcal{L}} \Theta_{\mathcal{L}\mathcal{L}} \Pi_{\mathcal{L}}$, $G_{\mathcal{U}\mathcal{U}} = \Pi'_{\mathcal{U}} \Theta_{\mathcal{U}\mathcal{U}} \Pi_{\mathcal{U}}$, and $Q = G_{\mathcal{U}\mathcal{U}}^{-1} \Pi'_{\mathcal{U}} \Theta_{\mathcal{U}\mathcal{U}} H$. For $1 \leq k \leq K$, $\psi_k(H) = \arccos\Big( \frac{M_{kk^*}}{\sqrt{M_{kk}}\sqrt{M_{k^*k^*}}} \Big)$, where $M = P(G_{\mathcal{L}\mathcal{L}}^2 + G_{\mathcal{U}\mathcal{U}} QQ' G_{\mathcal{U}\mathcal{U}})P$.*

The choice of $H$ is flexible. For convenience, we focus on the class of $H$ that is an eligible community membership matrix, i.e., $H = \hat{\Pi}_{\mathcal{U}}$. Our theory can be easily extended to more general forms of $H$.

**Definition 1.** *For any $b_0 \in (0, 1)$, we say that $\hat{\Pi}_{\mathcal{U}}$ is $b_0$-correct if $\min_T \big( \sum_{i \in \mathcal{U}} \theta_i \cdot 1\{T\hat{\pi}_i \neq \pi_i\} \big) \leq b_0 \|\theta\|_1$, where the minimum is taken over all permutations of $K$ columns of $\hat{\Pi}_{\mathcal{U}}$.*

The next two theorems study $\psi_k(H)$ and $\hat{\psi}_k(H)$, respectively, for $H = \hat{\Pi}_{\mathcal{U}}$.

**Theorem 1.** *Consider the DCBM model where (8)-(9) hold. Let $k^*$ denote the true community label of the new node. Suppose $\hat{\Pi}_{\mathcal{U}}$ is $b_0$-correct, for a constant $b_0 \in (0, 1)$. When $b_0$ is properly small, there exists a constant $c_0 > 0$, which does not depend on $b_0$, such that $\psi_{k^*}(\hat{\Pi}_{\mathcal{U}}) = 0$ and $\min_{k \neq k^*} \{\psi_k(\hat{\Pi}_{\mathcal{U}})\} \geq c_0 \beta_n$.*

**Theorem 2.** *Consider the DCBM model where (8)-(9) hold. There exists constant $C > 0$, such that for any $\delta \in (0, 1/2)$, with probability $1 - \delta$, simultaneously for $1 \leq k \leq K$, $|\hat{\psi}_k(\hat{\Pi}_{\mathcal{U}}) - \psi_k(\hat{\Pi}_{\mathcal{U}})| \leq C \left( \sqrt{\frac{\log(1/\delta)}{\|\theta\|_1 \cdot \min\{\theta^*, \|\theta_{\mathcal{L}}^{(k)}\|_1\}}} + \frac{\|\theta_{\mathcal{L}}^{(k)}\|^2}{\|\theta_{\mathcal{L}}^{(k)}\|_1 \|\theta\|_1} \right)$.*

Write $\hat{\psi}_k = \hat{\psi}_k(\hat{\Pi}_{\mathcal{U}})$ and $\psi_k = \psi_k(\hat{\Pi}_{\mathcal{U}})$ for short. When $\max_k \{|\hat{\psi}_k - \psi_k|\} < (1/2) \min_{k \neq k^*} \{\psi_k\}$, the community label of the new node is correctly estimated. We can immediately translate the results in Theorems 1-2 to an upper bound for the misclassification probability.

**Corollary 1.** *Consider the DCBM model where (8)-(9) hold. Suppose for some constants $b_0 \in (0, 1)$ and $\epsilon \in (0, 1/2)$, $\hat{\Pi}_{\mathcal{U}}$ is $b_0$-correct with probability $1 - \epsilon$. When $b_0$ is properly small, there exist constants $C_0 > 0$ and $\bar{C} > 0$, which do not depend on $(b_0, \epsilon)$, such that $\mathbb{P}(\hat{y} \neq k^*) \leq \epsilon + \bar{C} \sum_{k=1}^K \exp\Big( -C_0 \beta_n^2 \|\theta\|_1 \cdot \min\{\theta^*, \|\theta_{\mathcal{L}}^{(k)}\|_1\} \Big)$.*

**Remark 4:** When $\min_k \|\theta_{\mathcal{L}}^{(k)}\|_1 \geq O(\theta^*)$, the stochastic noise in $X$ will dominate the error, and the misspecification probability in Corollary 1 will not improve with more label information. Typically, the error rate will be the same as in the ideal case that $\Pi_{\mathcal{U}}$ is known (except there is no $\epsilon$ in the ideal case). Hence, only little label information can make AngleMin+ perform almost as well as a fully supervised algorithm that possesses all the label information. We will formalize this in Section 3.

**Remark 5:** Notice that $\min_T \big( \sum_{i \in \mathcal{U}} \theta_i \cdot 1\{T\hat{\pi}_i \neq \pi_i\} \big) \leq \frac{1}{K!} \sum_T \big( \sum_{i \in \mathcal{U}} \theta_i \cdot 1\{T\hat{\pi}_i \neq \pi_i\} \big) \leq \frac{K-1}{K} \|\theta_{\mathcal{U}}\|_1$. Therefore, if $\|\theta_{\mathcal{L}}\|_1 \geq (1 - \frac{Kb_0}{K-1})\|\theta\|_1$, then $\min_T \big( \sum_{i \in \mathcal{U}} \theta_i \cdot 1\{T\hat{\pi}_i \neq \pi_i\} \big) \leq b_0 \|\theta\|_1$ is always true. In other words, as long as the information on the labels is strong enough, AngleMin+ would not require any assumption on the unsupervised community detection algorithm.

For AngleMin+ to be consistent, we need the bound in Corollary 1 to be $o(1)$. It then requires that for a small constant $b_0$, $\hat{\Pi}_{\mathcal{U}}$ is $b_0$-correct with probability $1 - o(1)$. This is a mild requirement and can be achieved by several unsupervised community detection algorithms. The next corollary studies the specific version of AngleMin+, when $\hat{\Pi}_{\mathcal{U}}$ is from SCORE+:

**Corollary 2.** *Consider the DCBM model where (8)-(9) hold. We apply SCORE+ to obtain $\hat{\Pi}_{\mathcal{U}}$ and plug it into AngleMin+. As $n \to \infty$, suppose for some constant $q_0 > 0$, $\min_{i \in \mathcal{U}} \theta_i \geq q_0 \max_{i \in \mathcal{U}} \theta_i$, $\beta_n \|\theta_{\mathcal{U}}\| \geq q_0 \sqrt{\log(n)}$, $\beta_n^2 \|\theta\|_1 \theta^* \to \infty$, and $\beta_n^2 \|\theta\|_1 \min_k \{\|\theta_{\mathcal{L}}^{(k)}\|_1\} \to \infty$. Then, $\mathbb{P}(\hat{y} \neq k^*) \to 0$, so the AngleMin+ estimate is consistent.*

**3.2 Comparison with an information theoretical lower bound**   We compare the performance of AngleMin+ with an ideal estimate that has access to all model parameters, except for the community label $k^*$ of the new node. For simplicity, we first consider the case of $K = 2$. For any label predictor $\tilde{y}$ for the new node, define $\text{Risk}(\tilde{y}) = \sum_{k^* \in [K]} \mathbb{P}(\tilde{y} \neq k^* | \pi^* = e_{k^*})$.

**Lemma 2.** *Consider a DCBM with $K = 2$ and $P = (1 - b)I_2 + b\mathbf{1}_2\mathbf{1}_2'$. Suppose $\theta^* = o(1)$, $\frac{\theta^*}{\min_k \|\theta_{\mathcal{L}}^{(k)}\|_1} = o(1)$, $1 - b = o(1)$, $\frac{\|\theta_{\mathcal{L}}^{(1)}\|_1}{\|\theta_{\mathcal{L}}^{(2)}\|_1} = \frac{\|\theta_{\mathcal{U}}^{(1)}\|_1}{\|\theta_{\mathcal{U}}^{(2)}\|_1} = 1$. There exists a constant $c_4 > 0$ such that $\inf_{\tilde{y}}\{\text{Risk}(\tilde{y})\} \geq c_4 \exp\left\{-2[1 + o(1)]\frac{(1-b)^2}{8} \cdot \theta^*(\|\theta_{\mathcal{L}}\|_1 + \|\theta_{\mathcal{U}}\|_1)\right\}$, where the infimum is taken over all measurable functions of $A$, $X$, and parameters $\Pi_{\mathcal{L}}$, $\Pi_{\mathcal{U}}$, $\Theta$, $P$, $\theta^*$. In AngleMin+, suppose the second part of condition 9 holds with $c_3 = o(1)$, $\hat{\Pi}_{\mathcal{U}}$ is $\tilde{b}_0$-correct with $\tilde{b}_0 \overset{a.s.}{\to} 0$. There is a constant $C_4 > 0$ such that, $\text{Risk}(\hat{y}) \leq C_4 \exp\left\{-[1 - o(1)]\frac{(1-b)^2}{8} \cdot \theta^* \frac{(\|\theta_{\mathcal{L}}\|_1^2 + \|\theta_{\mathcal{U}}\|_1^2)^2}{\|\theta_{\mathcal{L}}\|_1^3 + \|\theta_{\mathcal{U}}\|_1^3}\right\}$.*

Lemma 2 indicates that the classification error of AngleMin+ is almost the same as the information theoretical lower bound of an algorithm that knows all the parameters except $\pi^*$ apart from a mild difference of the exponents. This difference comes from two sources. The first is the extra "2" in the exponent of $\text{Risk}(\hat{y})$, which is largely an artifact of proof techniques, because we bound the total variation distance by the Hellinger distance (the total variation distance is hard to analyze directly). The second is the difference of $\|\theta_{\mathcal{L}}\|_1 + \|\theta_{\mathcal{U}}\|_1$ in $\inf_{\tilde{y}}\{\text{Risk}(\tilde{y})\}$ and $\frac{(\|\theta_{\mathcal{L}}\|_1^2 + \|\theta_{\mathcal{U}}\|_1^2)^2}{\|\theta_{\mathcal{L}}\|_1^3 + \|\theta_{\mathcal{U}}\|_1^3}$ in $\text{Risk}(\hat{y})$. Note that $\frac{(\|\theta_{\mathcal{L}}\|_1^2 + \|\theta_{\mathcal{U}}\|_1^2)^2}{\|\theta_{\mathcal{L}}\|_1^3 + \|\theta_{\mathcal{U}}\|_1^3} \leq \|\theta_{\mathcal{L}}\|_1 + \|\theta_{\mathcal{U}}\|_1 \leq 1.125 \frac{(\|\theta_{\mathcal{L}}\|_1^2 + \|\theta_{\mathcal{U}}\|_1^2)^2}{\|\theta_{\mathcal{L}}\|_1^3 + \|\theta_{\mathcal{U}}\|_1^3}$, so this difference is quite mild. It arises from the fact that AngleMin+ does not aggregate the information in labeled and unlabeled data by adding the first and last $K$ coordinates of $f(x; H)$ together. The reason we do not do this is that unsupervised community detection methods only provide class labels up to a permutation, and practically it is really hard to estimate this permutation, which will result in the algorithm being extremely unstable. To conclude, the difference of the error rate of our method and the information theoretical lower bound is mild, demonstrating that our algorithm is nearly optimal. For a general $K$, we have a similar conclusion:

**Theorem 3.** *Suppose the conditions of Corollary 1 hold, where $b_0$ is properly small , and suppose that $\hat{\Pi}_{\mathcal{U}}$ is $b_0$-correct. Furthermore, we assume for sufficiently large constant $C_3$, $\theta^* \leq \frac{1}{C_3}$, $\theta^* \leq \min_{k \in [K]} C_3 \|\theta_{\mathcal{L}}^{(k)}\|_1$, and for a constant $r_0 > 0$, $\min_{k \neq \ell}\{P_{k\ell}\} \geq r_0$. Then, there is a constant $\tilde{c}_2 = \tilde{c}_2(K, C_1, C_2, C_3, c_3, r_0) > 0$ such that $[-\log(\tilde{c}_2 \text{Risk}(\hat{y}))]/[-\log(\inf_{\tilde{y}}\{\text{Risk}(\tilde{y})\})] \geq \tilde{c}_2$.*

**3.3 In-sample Classification**   In this part, we briefly discuss the in-sample classification problem. Formally, our goal is to estimate $\pi_i$ for all $i \in \mathcal{U}$. As mentioned in section 1, an in-sample classification algorithm can be directly derived from AngleMin+: for each $i \in \mathcal{U}$, predict the label of $i$ as $\hat{y}_i(H) = \arg\min_{1 \leq k \leq K} \psi\big(f(A_{-i}^{(k)}; H_i), \ f(A_{-i,i}; H_i)\big)$, where $A_{-i}^{(k)}$ is the subvector of $A^{(k)}$ by removing the $i$th entry, $A_{-i,i}$ is the subvector of $A_i$ by removing the $i$th entry, and $H_i$ is a $(|\mathcal{U}| - 1) \times K$ projection matrix which may be different across distinct $i$. As discussed in subsection 2, the choices of $H_i$ are quite flexible. For purely theoretical convenience, we would focus on the case that $H_i = \hat{\Pi}_{\mathcal{U} \setminus \{i\}}$. For any in-sample classifier $\tilde{y} = (\tilde{y}_i)_{i \in \mathcal{U}} \in [K]^{|\mathcal{U}|}$, define the in-sample risk $\text{Risk}_{ins}(\tilde{y}) = \frac{1}{|\mathcal{U}|} \sum_{i \in \mathcal{U}} \sum_{k^* \in [K]} \mathbb{P}(\tilde{y}_i \neq k^* | \pi_i = e_{k^*})$. For the above in-sample classification algorithm, we have similar theoretical results as in section 3 on consistency and efficiency under some very mild conditions:

**Theorem 4.** *Consider the DCBM model where (8)-(9) hold. We apply SCORE+ to obtain $\hat{\Pi}_{\mathcal{U} \setminus \{i\}}$ and plug it into the above algorithm. As $n \to \infty$, suppose for some constant $q_0 > 0$, $\min_{i \in \mathcal{U}} \theta_i \geq q_0 \max_{i \in \mathcal{U}} \theta_i$, $\beta_n \|\theta_{\mathcal{U}}\| \geq q_0 \sqrt{\log(n)}$, $\beta_n^2 \|\theta\|_1 \min_{i \in \mathcal{U}} \theta_i \to \infty$, and $\beta_n^2 \|\theta\|_1 \min_k\{\|\theta_{\mathcal{L}}^{(k)}\|_1\} \to \infty$. Then, $\frac{1}{|\mathcal{U}|} \sum_{i \in \mathcal{U}} \mathbb{P}(\hat{y}_i \neq k_i) \to 0$, so the above in-sample classification algorithm is consistent.*

**Theorem 5.** *Suppose the conditions of Corollary 1 hold, where $b_0$ is properly small , and suppose that $\hat{\Pi}_{\mathcal{U} \setminus \{i\}}$ is $b_0$-correct for all $i \in \mathcal{U}$. Furthermore, we assume for sufficiently large constant $C_3$, $\max_{i \in \mathcal{U}} \theta_i \leq \frac{1}{C_3}$, $\max_{i \in \mathcal{U}} \theta_i \leq \min_{k \in [K]} C_3 \|\theta_{\mathcal{L}}^{(k)}\|_1$, $\log(|\mathcal{U}|) \leq C_3 \beta_n^2 \|\theta\|_1 \min_{i \in \mathcal{U}} \theta_i$, and for a constant $r_0 > 0$, $\min_{k \neq \ell}\{P_{k\ell}\} \geq r_0$. Then, there is a constant $\tilde{c}_{21} = \tilde{c}_{21}(K, C_1, C_2, C_3, c_3, r_0) > 0$ such that $[-\log(\tilde{c}_{21}\text{Risk}_{ins}(\hat{y}))]/[-\log(\inf_{\tilde{y}}\{\text{Risk}_{ins}(\tilde{y})\})] \geq \tilde{c}_{21}$, so the above in-sample classification algorithm is efficient.*

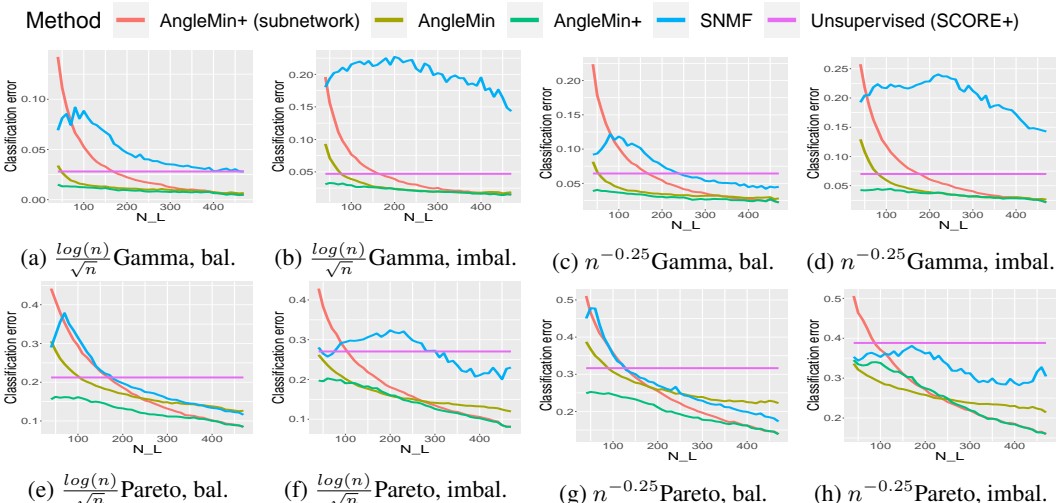

Figure 1: Simulations ($n = 500$, $K = 3$; data are generated from DCBM). In each plot, the x-axis is the number of labeled nodes, and the y-axis is the average misclassification rate over 100 repetitions.

## 4 EMPIRICAL STUDY

We study the performance of AngelMin+, where $\hat{\Pi}_\mathcal{U}$ is from SCORE+ (Jin et al., 2021). We compare our methods with SNMF (Yang et al., 2015) (a representative of semi-supervised approaches) and SCORE+ (a fully unsupervised approach). We also compare our algorithm to typical GNN methods (Kipf & Welling, 2016) in the real data part.

**Simulations**: To illustrate how information in $A_{\mathcal{U}\mathcal{U}}$ will improve the classification accuracy, we would consider AngleMin in (4) in simulations. Also, to cast light on how information on unlabeled data will ameliorate the classification accuracy, we consider a special version of AngleMin+ in simulations by feeding into the algorithm only $A_{\mathcal{L}\mathcal{L}}$ and $X_\mathcal{L}$. It ignores information on unlabeled data and only uses the subnetwork consisting of labeled nodes. We call it AngleMin+(subnetwork). This method is practically uninteresting, but it serves as a representative of the fully supervised approach that ignores unlabeled nodes. We simulate data from the DCBM with $(n, K) = (500, 3)$. To generate $P$, we draw its (off diagonal) entries from $\text{Uniform}(0, 1)$, and then symmetrize it. We generate the degree heterogeneity parameters $\theta_i$ i.i.d. from one of the 4 following distributions: $n^{-0.5}\sqrt{\log(n)}\text{Gamma}(3.5)$, $n^{-0.25}\text{Gamma}(3.5)$, $n^{-0.5}\sqrt{\log(n)}\text{Pareto}(3.5)$, $n^{-0.25}\text{Pareto}(3.5)$. They cover most scenarios: Gamma distributions have considerable mass near 0, so the network has severely low degree nodes; Pareto distributions have heavy tails, so the network has severely high degree nodes. The scaling $n^{-0.5}\sqrt{\log(n)}$ corresponds to the sparse regime, where the average node degree is $\asymp \log(n)^2$, and $n^{-0.25}$ corresponds to the dense regime, with average node degree $\asymp \sqrt{n}$. We consider two cases of $\Pi$: the balanced case (bal.) and the imbalanced case (inbal.). In the former, $\pi(i)$ are i.i.d. from $\text{Multinomial}(1/3, 1/3, 1/3)$, and in the latter, $\pi(i)$ are i.i.d. from $\text{Multinomial}(0.2, 0.2, 0.6)$. We repeat the simulation 100 times. Our results are presented in Figure 1, which shows the average classification error of each algorithm as the number of labeled nodes, $N_L$ increases. The plots indicate that AngleMin+ outperforms other methods in all the cases. Furthermore, though AngleMin is not so good as AngleMin+ when $N_L$ is small, it still surpasses all the other approaches except AngleMin+ in most scenarios. Compared to supervised and unsupervised methods which only use part of the data, we can see that AngleMin+ gains a great amount of accuracy by leveraging on both the labeled and unlabeled data.

**Real data**: We consider three benchmark datasets for community detection, Caltech (Traud et al., 2012) , Simmons (Traud et al., 2012) , and Polblogs (Adamic & Glance, 2005). For each data set, we separate nodes into 10 folds and treat each fold as the test data at a time, with the other 9 folds as training data. In the training network, we randomly choose $n_L$ nodes as labeled nodes. We then estimate the label of each node in the test data and report the misclassification error rate (averaged

over 10 folds). We consider $n_L/n \in \{0.3, 0.5, 0.7\}$, where $n$ is the number of nodes in training data. The results are shown in Table 1. In most cases, AngleMin+ significantly outperforms the other methods (unsupervised or semi-supervised). Additionally, we notice that in the Polblogs data, the standard deviation of the error of SCORE+ is quite large, indicating that its performance is unstable. Remarkably, even though AngleMin+ uses SCORE+ to initialize, the performance of AngleMin+ is nearly unaffected: It still achieves low means and standard deviations in misclassification error. This is consistent with our theory in Section 3. We also compare the running time of different methods (please see Section B of the appendix) and find that AngleMin+ is much faster than SNMF.

Table 1: Average misclassification error over 10 data splits, with standard deviation in the parentheses.

| Dataset | $n$ | $K$ | $n_L/n$ | SCORE+ | AngleMin+ | SNMF | GNN (cons.) | GNN (random) | GNN (adj.) | GNN (LP) | GNN (node2vec) | GNN ($A\Pi$) |
|---|---|---|---|---|---|---|---|---|---|---|---|---|
| Caltech | 590 | 8 | 0.3 | 0.237 (0.061) | **0.207** (0.059) | 0.312 (0.049) | 0.858 (0.038) | 0.859 (0.035) | 0.875 (0.038) | 0.839 (0.046) | 0.859 (0.055) | 0.880 (0.026) |
| | | | 0.5 | | **0.151** (0.040) | 0.310 (0.042) | 0.846 (0.054) | 0.895 (0.026) | 0.859 (0.037) | 0.861 (0.043) | 0.859 (0.039) | 0.856 (0.040) |
| | | | 0.7 | | **0.137** (0.046) | 0.264 (0.051) | 0.849 (0.043) | 0.861 (0.034) | 0.856 (0.031) | 0.859 (0.036) | 0.880 (0.027) | 0.842 (0.027) |
| Simmons | 1137 | 4 | 0.3 | 0.234 (0.084) | **0.128** (0.024) | 0.266 (0.041) | 0.691 (0.022) | 0.702 (0.039) | 0.702 (0.036) | 0.698 (0.026) | 0.706 (0.039) | 0.696 (0.028) |
| | | | 0.5 | | **0.096** (0.024) | 0.233 (0.033) | 0.691 (0.022) | 0.711 (0.034) | 0.685 (0.025) | 0.691 (0.022) | 0.710 (0.031) | 0.691 (0.022) |
| | | | 0.7 | | **0.092** (0.015) | 0.220 (0.037) | 0.691 (0.022) | 0.692 (0.022) | 0.691 (0.022) | 0.691 (0.022) | 0.707 (0.043) | 0.698 (0.026) |
| Polblogs | 1222 | 2 | 0.3 | 0.166 (0.165) | 0.074 (0.036) | **0.073** (0.019) | 0.499 (0.044) | 0.502 (0.038) | 0.439 (0.048) | 0.482 (0.037) | 0.502 (0.059) | 0.501 (0.044) |
| | | | 0.5 | | 0.092 (0.041) | **0.068** (0.033) | 0.517 (0.040) | 0.516 (0.038) | 0.453 (0.056) | 0.488 (0.044) | 0.499 (0.061) | 0.484 (0.041) |
| | | | 0.7 | | 0.066 (0.026) | **0.063** (0.028) | 0.485 (0.041) | 0.492 (0.043) | 0.430 (0.062) | 0.493 (0.041) | 0.492 (0.050) | 0.486 (0.039) |

GNN is a popular approach for attributed node clustering. Although it is not designed for the case of no node attributes, we are still interested in whether GNN can be easily adapted to our setting by self-created features. We take the GCN method in Kipf & Welling (2016) and consider 6 schemes of creating a feature vector for each node: i) a 50-dimensional constant vector of 1's, ii) a 50-dimensional randomly generated feature vector, iii) the $n$-dimensional adjacency vector, iv) the vector of landing probabilities (LP) (Li et al., 2019) (which contains network topology information), v) the embedding vector from node2vec (Grover & Leskovec, 2016), and vi) a practically infeasible vector $e_i' A\Pi \in \mathbb{R}^K$ (which uses the true $\Pi$). The results are in Table 1. GCN performs unsatisfactorily, regardless of how the features are created. For example, propagating messages with all-1 vectors seems to result in over-smoothing; and using adjacency vectors as node features means that the feature transformation linear layers' size changes with the number of nodes in a network, which could heavily overfit due to too many parameters. We conclude that it is not easy to adapt GNN to the case of no node attributes.

For a fairer comparison, we also consider a real network, Citeseer (Sen et al., 2008), that contains node features. We consider two state-of-the-art semi-supervised GNN algorithms, GCN (Kipf & Welling, 2016) and MasG (Jin et al., 2019). Our methods can also be generalized to accommodate node features. Using the "fusion" idea surveyed in Chunaev et al. (2019), we "fuse" the adjacency matrix $\bar{A}$ (on $n+1$ nodes) and node features into a weighted adjacency matrix $\bar{A}_{\text{fuse}}$ (see the appendix for details). We denote its top left block by $A_{\text{fuse}} \in \mathbb{R}^{n \times n}$ and its last column by $X_{\text{fuse}} \in \mathbb{R}^n$ and apply AngleMin+ by replacing $(A, X)$ by $(A_{\text{fuse}}, X_{\text{fuse}})$. The misclassification error averaged over 10 data splits is reported in Table 2. The error rates of GCN and MasG are quoted from those papers, which are based on 1 particular data split. We also re-run GCN on our 10 data splits.

Table 2: Error rates on Citeseer, where node attributes are available. If the error rate has *, it is quoted from literature and based on one particular data split; otherwise, it is averaged over 10 data splits.

| Dataset | $n$ | $K$ | $n_L/n$ | GCN | GCN* | MasG* | AngleMin+ |
|---|---|---|---|---|---|---|---|
| Citeseer | 3312 | 6 | 0.036 | 0.321 | 0.297 | 0.268 | 0.334 |

**Conclusion and discussions**: In this paper, we propose a fast semi-supervised community detection algorithm AngleMin+ based on the structural similarity metric of DCBM. Our method is able to address degree heterogeneity and non-assortative network, is computationally fast, and possesses favorable theoretical properties on consistency and efficiency. Also, our algorithm performs well on both simulations and real data, indicating its strong usage in practice.

There are possible extensions for our method. Our method does not directly deal with soft label (a.k.a mixed membership) where the available label information is the probability of a certain node being in each community. We are currently endeavoring to solve this by fitting our algorithm into the degree-corrected mixed membership model (DCMM), and developing sharp theories for it.

## ACKNOWLEDGMENTS

This work is partially supported by the NSF CAREER grant DMS-1943902.

## ETHICS STATEMENT

This paper proposes a novel semi-supervised community detection algorithm, AngleMin+, based on the structural similarity metric of DCBM. Our method may be maliciously manipulated to identify certain group of people such as dissenters. This is a common drawback of all the community detection algorithms, and we think that this can be solved by replacing the network data by their differential private counterpart. All the real data we use come from public datasets which we have clearly cited, and we do not think that they will raise any privacy issues or other potential problems.

## REPRODUCIBILITY STATEMENT

We provide detailed theory on our algorithm AngleMin+. we derive explicit bounds for the misclassification probability of our method under DCBM, and show that it is consistent. We also study the efficiency of our method by comparing its misclassification probability with that of an ideal classifier having access to the community labels of all nodes. Additionally, we provide clear explanations and insights of our theory. All the proofs, together with some generalization of our theory, are available in the appendix. Also, we perform empirical study on our proposed algorithms under both simulations and real data settings, and we consider a large number of scenarios in both cases. All the codes are available in the supplementary materials.

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

APPENDIX

## A PSEUDO CODE OF THE ALGORITHM

Below are the pseudo code of AngleMin+ which is deferred to the appendix due to the page limit.

---

**Algorithm 1:** AngleMin+

---

**Input:** Number of communities $K$, adjacency matrix $A \in \mathbb{R}^{n \times n}$, community labels $y_i$ for nodes in $i \in \mathcal{L}$, and the vector of edges between a new node and the existing nodes $X \in \mathbb{R}^n$ .
**Output:** Estimated community label $\hat{y}$ of the new node.

1. Unsupervised community detection: Apply a community detection algorithm (e.g., SCORE+ in Section 2) on $A_{\mathcal{U}\mathcal{U}}$, and let $\hat{\Pi}_{\mathcal{U}} = [\hat{\pi}_i]_{i \in \mathcal{U}}$ store the estimated community labels, where $\hat{\pi}_i = e_k$ if and only if node $k$ is clustered to community $k$, $1 \le k \le K$.

2. Assigning the community label to a new node: Let $\Pi_{\mathcal{L}} = [\pi_i]_{i \in \mathcal{L}}$ contain the community memberships of labeled nodes, where $\pi_i = e_k$ if and only if $y_i = k$, $1 \le k \le K$. Let $H = \hat{\Pi}_{\mathcal{U}}$. Compute

$$x = \left[ X'_{\mathcal{L}} \Pi_{\mathcal{L}}, \ X'_{\mathcal{U}} H \right]', \qquad v_k = \left[ e'_k \Pi'_{\mathcal{L}} A_{\mathcal{L}\mathcal{L}} \Pi_{\mathcal{L}}, \ e'_k \Pi'_{\mathcal{L}} A_{\mathcal{L}\mathcal{U}} H \right]', \quad 1 \le k \le K.$$

Suppose $k^*$ minimizes the angle between $v_k$ and $x$, among $1 \le k \le K$ (if there is a tie, pick the smaller $k$). Output $\hat{y} = k^*$.

---

## B RUNNING TIME

Table 3 exhibits the running time of all the algorithms considered in Table 1. It can be seen from the result that our algorithm AngleMin+ is much faster than all the other algorithms. This is one of the merits of our method.

Table 3: Running time on Caltech, Simons, and Polblogs networks. The quantities outside and inside the parentheses are the means and standard deviations of the running time, respectively.

| Dataset | $n$ | $K$ | $n_L/n$ | SCORE+ | AngleMin+ | SNMF | GNN (cons.) | GNN (random) | GNN (adj.) | GNN (LP) | GNN (node2vec) | GNN ($A\Pi$) |
|---|---|---|---|---|---|---|---|---|---|---|---|---|
| Caltech | 590 | 8 | 0.3 | 0.083 (0.009) | **0.068** (0.064) | 0.178 (0.017) | 0.277 (0.154) | 0.249 (0.049) | 0.311 (0.100) | 0.296 (0.044) | 0.498 (0.053) | 0.396 (0.097) |
| | | | 0.5 | | **0.034** (0.003) | 0.211 (0.069) | 0.575 (0.133) | 0.535 (0.061) | 0.620 (0.133) | 0.609 (0.067) | 0.836 (0.080) | 0.649 (0.045) |
| | | | 0.7 | | **0.022** (0.003) | 0.211 (0.054) | 0.861 (0.099) | 0.892 (0.116) | 1.068 (0.213) | 0.949 (0.049) | 1.204 (0.186) | 0.998 (0.068) |
| Simmons | 1137 | 4 | 0.3 | 0.157 (0.008) | **0.075** (0.008) | 0.515 (0.036) | 0.334 (0.086) | 0.344 (0.102) | 0.564 (0.273) | 0.421 (0.094) | 1.045 (0.680) | 0.455 (0.087) |
| | | | 0.5 | | **0.054** (0.011) | 0.577 (0.090) | 0.691 (0.199) | 0.692 (0.084) | 1.245 (0.691) | 0.642 (0.032) | 1.106 (0.151) | 0.685 (0.059) |
| | | | 0.7 | | **0.031** (0.003) | 0.541 (0.073) | 0.988 (0.139) | 0.897 (0.056) | 1.208 (0.454) | 0.958 (0.057) | 1.977 (0.775) | 1.046 (0.069) |
| Polblogs | 1222 | 2 | 0.3 | 0.093 (0.014) | **0.054** (0.006) | 0.356 (0.034) | 0.402 (0.127) | 0.353 (0.093) | 0.444 (0.160) | 0.311 (0.055) | 0.810 (0.261) | 0.343 (0.031) |
| | | | 0.5 | | **0.031** (0.004) | 0.431 (0.098) | 0.780 (0.147) | 0.700 (0.181) | 0.965 (0.179) | 0.649 (0.054) | 1.031 (0.190) | 0.644 (0.044) |
| | | | 0.7 | | **0.022** (0.004) | 0.351 (0.037) | 1.135 (0.118) | 1.152 (0.314) | 1.430 (0.169) | 0.986 (0.149) | 1.408 (0.210) | 0.999 (0.060) |

## C COMPARISON WITH LOCAL REFINEMENT ALGORITHM

We would first illustrate why local refinement may not work with an example and then explain our insight behind it.

Consider a network with $n = 4m$ nodes and $K = 2$ communities. Suppose that there are $2m$ labeled nodes, $m$ of them are in community $\mathcal{C}_1$ and have degree heterogeneity $\theta = 0.8$, and the other $m$ of them are in community $\mathcal{C}_2$ and have degree heterogeneity $\theta = 0.5$. There are $2m$ unlabeled nodes, $m$ of them are in community $\mathcal{C}_1$ and have degree heterogeneity $\theta = 0.6$, and the other $m$ of them are in community $\mathcal{C}_2$ and have degree heterogeneity $\theta = 0.7$. The $P$ matrix is defined as follows:

$$P = \begin{pmatrix} 1 & 0.9 \\ 0.9 & 1 \end{pmatrix}$$

Under this setting, all the assumptions in our paper are satisfied.

On the other hand, recall that the prototypical refinement algorithm, Algorithm 2 of Gao et al. (2018) is defined as follows:

$$\hat{y}_i = arg \max_{u \in [K]} \frac{1}{|\{j : \hat{y}^0(j) = u\}|} \sum_{\{j : \hat{y}^0(j) = u\}} A_{ij}$$

where $\hat{y}^0$ is a vector of community label and $\hat{y}$ is the refined community label.

For semi-supervised setting, one may consider the following modification of local refinement algorithm:

(i) Apply local refinement algorithm, with known labels to assign nodes in $\mathcal{U}$.

(ii) With the labels of all nodes, one updates the labels of every node by applying the same refinement procedure.

Under the setting of our toy example, for step (i), all the unlabeled nodes which are actually in community $\mathcal{C}_2$ will be assigned to community $\mathcal{C}_1$ with probability converging to 1 as $n \rightarrow \infty$. The reason is that for any unlabeled node $i$ which is actually in community $\mathcal{C}_2$, when $u = 1$, $\{A_{ij} : j \in \mathcal{L}, y_j = u\}$ are iid $\sim Bern(\theta_i \theta_j P_{21}) = Bern(\theta_i \cdot 0.8 \cdot 0.9) = Bern(0.72\theta_i)$; when $u = 2$, $\{A_{ij} : j \in \mathcal{L}, y_j = u\}$ are iid $\sim Bern(\theta_i \theta_j P_{22}) = Bern(\theta_i \cdot 0.5 \cdot 1) = Bern(0.5\theta_i)$. Hence, by law of large numbers,

$$\frac{1}{\{A_{ij} : j \in \mathcal{L}, y_j = u\}} \sum_{\{A_{ij} : j \in \mathcal{L}, y_j = u\}} A_{ij} \overset{a.s.}{\rightarrow} \begin{cases} 0.72\theta_i, & u = 1 \\ 0.5\theta_i, & u = 2 \end{cases}$$

Consequently, the prototypical refinement algorithm will incorrectly assign all the unlabeled nodes which are actually in the community $\mathcal{C}_2$ to $\mathcal{C}_1$ with probability converging to 1 as $n \rightarrow \infty$. This will cause a classification error of at least $50\%$.

Based on the huge classification error in step (i), step (ii) will also perform poorly. Similar to the reasoning above, by law of large numbers, it can be shown that after step (ii). the algorithm will still assign all the unlabeled nodes which are actually in the community $\mathcal{C}_2$ to $\mathcal{C}_1$ with probability converging to 1 as $n \rightarrow \infty$. In other words, even if the local refinement algorithm is applied to the whole network, a classification error of at least $50\%$ will always remain.

Even if all the labels of the nodes are known, applying the local refinement algorithm still can cause severe errors. Still consider our toy example. Suppose now that we know the label of all the nodes, and we perform the local refinement algorithm on these known labels in an attempt to purify them. By the law of large numbers, however, it is not hard to show that for any node $i$ which is actually in community $\mathcal{C}_2$,

$$\frac{1}{\{A_{ij} : j \in \mathcal{L}, y_j = u\}} \sum_{\{A_{ij} : y_j = u\}} A_{ij} \overset{a.s.}{\rightarrow} \begin{cases} 0.63\theta_i, & u = 1 \\ 0.6\theta_i, & u = 2 \end{cases}$$

Consequently, similar to the previous cases, the local refinement algorithm will incorrectly assign all the unlabeled nodes which are actually in the community $\mathcal{C}_2$ to $\mathcal{C}_1$ with probability converging to 1 as $n \rightarrow \infty$. This will cause a classification error of at least $50\%$, even though the input of the algorithm is actually the true label vector.

To conclude, in general, the local refinement algorithms may not work under the broad settings of our paper. Intrinsically, label refinement is quite challenging when there is moderate degree heterogeneity, not to mention the scenarios where non-assortative networks occur. Local refinement algorithm works theoretically because strong assumptions on degree heterogeneity are imposed. For instance, it is required that the mean of the degree heterogeneity parameter in each community is $1 + o(1)$, which means that the network is extremely dense and that the degree heterogeneity parameters across communities are strongly balanced. Both of these two assumptions are hardly true in the real world, where most of the networks are sparse and imbalanced. Gao et al. (2018) is a very good paper, but we think that local refinement algorithm or similar algorithms might not be good choices for our problem.

## D  GENERALIZATION OF LEMMA 2

In the main paper, for the smoothness and comprehensibility of the text, we do not present the most general form of Lemma 2. We have a more general version of the lemma by relaxing the condition $\frac{\|\theta_{\mathcal{L}}^{(1)}\|_1}{\|\theta_{\mathcal{L}}^{(2)}\|_1} = \frac{\|\theta_{\mathcal{U}}^{(1)}\|_1}{\|\theta_{\mathcal{U}}^{(2)}\|_1} = 1$. Please see Section J for more details.

## E  PRELIMINARIES

For any positive integer $N$, Define $[N] = \{1, 2, ..., N\}$.

For a matrix $D$ and two index sets $S_1, S_2$, define $D_{S_1 S_2}$ to be the submatrix $(D_{ij})_{i \in S_1, j \in S_2}$, $D_{S_1 \cdot}$ to be the submatrix $(D_{ij})_{i \in S_1, j \in \mathcal{L} \cup \mathcal{U}}$, and $D_{\cdot S_2}$ to be the submatrix $(D_{ij})_{i \in \mathcal{L} \cup \mathcal{U}, | \in \mathcal{S}_\in}$.

The two main assumptions (8), (9) in the main paper are presented below for convenience.

$$\|P\|_{\max} \leq C_1, \qquad |\lambda_{\min}(P)| \geq \beta_n. \tag{8}$$

$$\frac{\max_{1 \leq k \leq K}\{\|\theta^{(k)}\|_1\}}{\min_{1 \leq k \leq K}\{\|\theta^{(k)}\|_1\}} \leq C_2, \qquad \max_{1 \leq k \leq K}\left\{\frac{\|\theta_{\mathcal{L}}^{(k)}\|^2}{\|\theta_{\mathcal{L}}^{(k)}\|_1 \|\theta\|_1}\right\} \leq c_3 \beta_n. \tag{9}$$

, where constant $c_3$ is properly small. We would specify this precisely in our proofs.

A number of lemmas used in our proofs will be presented as follows.

The following lemma shows that $\sin x$ and $x$ have the same order.

**Lemma 3.** *Let $x \in \mathbb{R}$. When $x \geq 0$, $\sin x \leq x$; when $x \in [0, \frac{\pi}{2}]$, $\sin x \geq \frac{2}{\pi}x$.*

Lemma 3 is quite obvious, but for the completeness of our work, we provide a proof for it.

*Proof.* Let $g_1(x) = \sin x - x$.

Then

$$\frac{d}{dx}g_1(x) = \cos x - 1 \leq 0$$

Hence $g_1(x)$ is monotonously decreasing on $\mathbb{R}$. As a result, when $x \geq 0$, $g_1(x) \geq g_1(0) = 0$. Therefore, when $x \geq 0$, $\sin x \leq x$.

Let $g_2(x) = \sin x - \frac{2}{\pi}x$. Then

$$\frac{d}{dx}g_2(x) = \cos x - \frac{2}{\pi}$$

Since $\cos x$ is monotonously decreasing on $[0, \frac{\pi}{2}]$, $\frac{d}{dx}g_2(x) \geq 0$ when $x \in [0, \arccos\frac{2}{\pi}]$ and $\frac{d}{dx}g_2(x) \leq 0$ when $x \in [\arccos\frac{2}{\pi}, \frac{\pi}{2}]$. Hence, $g_2(x)$ is monotonously increasing on $[0, \arccos\frac{2}{\pi}]$ and is monotonously decreasing on $[\arccos\frac{2}{\pi}, \frac{\pi}{2}]$. As a result, when $x \in [0, \frac{\pi}{2}]$,

$$g_2(x) \geq \min\{g_2(0), g_2(\frac{2}{\pi})\} = 0$$

Therefore, when $x \in [0, \frac{\pi}{2}]$, $\sin x \geq \frac{2}{\pi}x$.

$\square$

The following lemma demonstrates that the angle $\psi(u, v)$ in Definition 1 satisfies the triangle inequality, so that it can be regarded as a sort of "metric".

**Lemma 4** (Angle Inequality). *Let $x, y, z$ be three real vectors. Then,*

$$\psi(x, z) \leq \psi(x, y) + \psi(y, z)$$

The proof of Lemma 4 can be seen in Gustafson & Rao (1997), pg 56. Also, for completeness of our work, we provide a proof of Lemma 4.

*Proof.* Let $\tilde{x} = \frac{x}{\|x\|}$, $\tilde{y} = \frac{y}{\|y\|}$, $\tilde{z} = \frac{z}{\|z\|}$, then $\cos\psi(x,y) = \langle \tilde{x}, \tilde{y} \rangle$, $\cos\psi(y,z) = \langle \tilde{y}, \tilde{z} \rangle$, $\cos\psi(x,z) = \langle \tilde{x}, \tilde{z} \rangle$. Consider the following matrix

$$G = \begin{pmatrix} 1 & \cos\psi(x,y) & \cos\psi(x,z) \\ \cos\psi(x,y) & 1 & \cos\psi(y,z) \\ \cos\psi(x,z) & \cos\psi(y,z) & 1 \end{pmatrix} = \begin{pmatrix} \langle \tilde{x}, \tilde{x} \rangle & \langle \tilde{x}, \tilde{y} \rangle & \langle \tilde{x}, \tilde{z} \rangle \\ \langle \tilde{y}, \tilde{x} \rangle & \langle \tilde{y}, \tilde{y} \rangle & \langle \tilde{y}, \tilde{z} \rangle \\ \langle \tilde{z}, \tilde{x} \rangle & \langle \tilde{z}, \tilde{y} \rangle & \langle \tilde{z}, \tilde{z} \rangle \end{pmatrix}$$

For any vector $c = (c_1, c_2, c_3)^T \in \mathbb{R}^3$,

$$c^T G c = \langle c_1\tilde{x} + c_2\tilde{y} + c_3\tilde{z}, c_1\tilde{x} + c_2\tilde{y} + c_3\tilde{z} \rangle \geq 0$$

Also, $G$ is symmetric. Therefore, $G$ is positive semi-definite. As a result, $det(G) \geq 0$. In other word,

$$1 - \cos^2\psi(x,y) - \cos^2\psi(y,z) - \cos^2\psi(x,z) + 2\cos\psi(x,y)\cos\psi(y,z)\cos\psi(x,z) \geq 0$$

The above inequality can be rewritten as

$$(1 - \cos^2\psi(x,y))(1 - \cos^2\psi(y,z)) \geq (\cos\psi(x,y)\cos\psi(y,z) - \cos\psi(x,z))^2$$

or

$$(\sin\psi(x,y)\sin\psi(y,z))^2 \geq (\cos\psi(x,y)\cos\psi(y,z) - \cos\psi(x,z))^2$$

By definition of arccos, $\psi(x,y), \psi(y,z) \in [0,\pi]$, so $\sin\psi(x,y)\sin\psi(y,z) \geq 0$. Therefore,

$$-\sin\psi(x,y)\sin\psi(y,z) \leq \cos\psi(x,y)\cos\psi(y,z) - \cos\psi(x,z) \leq \sin\psi(x,y)\sin\psi(y,z)$$

$$\cos\psi(x,z) \geq \cos\psi(x,y)\cos\psi(y,z) - \sin\psi(x,y)\sin\psi(y,z)$$
$$\cos\psi(x,z) \geq \cos(\psi(x,y) + \psi(y,z))$$

If $\psi(x,y) + \psi(y,z) > \pi$, because by definition of arccos, $\psi(x,z) \in [0,\pi]$, it is immediate that

$$\psi(x,z) \leq \psi(x,y) + \psi(y,z)$$

If $\psi(x,y) + \psi(y,z) \leq \pi$, recall that $\psi(x,y), \psi(y,z) \in [0,\pi]$, hence $\psi(x,y) + \psi(y,z) \in [0,\pi]$. Also, $\psi(x,z) \in [0,\pi]$. Since cos is monotone decreasing on $[0,\pi]$, we obtain

$$\psi(x,z) \leq \psi(x,y) + \psi(y,z)$$

In all,

$$\psi(x,z) \leq \psi(x,y) + \psi(y,z)$$

$\square$

The following lemma relates angle to Euclidean distance.

**Lemma 5.** *Suppose that $x, y \in \mathbb{R}^m$, $\|y\| < \|x\|$. Then,*

$$\psi(x, x+y) \leq \arcsin\left(\frac{\|y\|}{\|x\|}\right)$$

*The equality holds if and only if $\langle y, x+y \rangle = 0$*

*Proof.* Let $\rho = \frac{\|y\|}{\|x\|}$, $\psi_0 = \psi(x, y)$. Then $\langle x, y \rangle = \|x\| \|y\| \cos \psi_0$. Notice that

$$\rho^2 (\rho + \cos \psi_0)^2 \geq 0$$

This can be rewritten as

$$(1 + \rho \cos \psi_0)^2 \geq (1 + \rho^2 + 2\rho \cos \psi_0)(1 - \rho^2)$$

Since $\|y\| < \|x\|$, so $\rho < 1$, $1 + \rho \cos \psi_0 > 0$. Hence

$$\frac{1 + \rho \cos \psi_0}{\sqrt{1 + \rho^2 + 2\rho \cos \psi_0}} \geq \sqrt{1 - \rho^2}$$

Plugging in $\rho = \frac{\|y\|}{\|x\|}$, we have

$$\frac{\|x\|^2 + \|x\| \|y\| \cos \psi_0}{\sqrt{\|x\|^2 + \|y\|^2 + 2\|x\| \|y\| \cos \psi_0}} \geq \sqrt{1 - \frac{\|y\|^2}{\|x\|^2}}$$

Since $\langle x, y \rangle = \|x\| \|y\| \cos \psi_0$,

$$\frac{\|x\|^2 + \langle x, y \rangle}{\|x\| \sqrt{\|x\|^2 + \|y\|^2 + 2\langle x, y \rangle}} \geq \sqrt{1 - \frac{\|y\|^2}{\|x\|^2}}$$

$$\frac{\langle x, x + y \rangle}{\|x\| \sqrt{\langle x + y, x + y \rangle}} \geq \sqrt{1 - \frac{\|y\|^2}{\|x\|^2}}$$

In other words,

$$\cos \psi(x, x + y) \geq \sqrt{1 - \frac{\|y\|^2}{\|x\|^2}} = \cos \arcsin \left( \frac{\|y\|}{\|x\|} \right)$$

Since $\frac{\|y\|}{\|x\|} \geq 0$, $\arcsin \left( \frac{\|y\|}{\|x\|} \right) \in [0, \frac{\pi}{2}]$. Therefore, by monotonicity of $\cos$ on $[0, \frac{\pi}{2}]$,

$$\psi(x, x + y) \leq \arcsin \left( \frac{\|y\|}{\|x\|} \right)$$

The equality holds if and only if $\rho^2 (\rho + \cos \psi_0)^2 \geq 0$, or equivalently,

$$(\|y\|^2 + \|x\| \|y\| \cos \psi_0)^2 = 0$$

This can be reduced to

$$\langle y, x + y \rangle = 0$$

$\square$

## F    PROOF OF LEMMA 1

**Lemma 1.** *Consider the DCBM model where (8)-(9) are satisfied. We define three $K \times K$ matrices: $G_{\mathcal{LL}} = \Pi'_{\mathcal{L}} \Theta_{\mathcal{LL}} \Pi_{\mathcal{L}}$, $G_{\mathcal{UU}} = \Pi'_{\mathcal{U}} \Theta_{\mathcal{UU}} \Pi_{\mathcal{U}}$, and $Q = G_{\mathcal{UU}}^{-1} \Pi'_{\mathcal{U}} \Theta_{\mathcal{UU}} H$. For $1 \leq k \leq K$,*

$$\psi_k(H) = \arccos \left( \frac{M_{kk^*}}{\sqrt{M_{kk}} \sqrt{M_{k^*k^*}}} \right), \qquad where \ M = P \left( G_{\mathcal{LL}}^2 + G_{\mathcal{UU}} Q Q' G_{\mathcal{UU}} \right) P.$$

*Proof.* Recall that

$$\psi_k(H) = \psi\Big(f(\Omega^{(k)};H),\ f(\mathbb{E}X;H)\Big), \qquad \text{for } 1 \le k \le K. \tag{11}$$

where

$$f(x;H) = \Big[x'_{\mathcal{L}}\mathbf{1}_{(1)},\ \dots,\ x'_{\mathcal{L}}\mathbf{1}_{(k)},\ x'_{\mathcal{U}}h_1,\dots,x'_{\mathcal{U}}h_K\Big]' = [x'_{\mathcal{L}}\Pi_{\mathcal{L}}, x'_{\mathcal{U}}H]'. \tag{12}$$

and

$$\Omega_j^{(k)} = \sum_{i \in \mathcal{L} \cap \mathcal{C}_k} \Omega_{ij} = e'_k \Pi'_{\mathcal{L}} \Omega_{\mathcal{L}\cdot} e_j$$

which indicates

$$\Omega^{(k)} = (e'_k \Pi'_{\mathcal{L}} \Omega_{\mathcal{L}\cdot})'$$

Hence

$$\begin{aligned}
f(\Omega^{(k)};H) &= f((e'_k\Pi'_{\mathcal{L}}\Omega_{\mathcal{L}\cdot})';H) \\
&= [e'_k\Pi'_{\mathcal{L}}\Omega_{\mathcal{L}\mathcal{L}}\Pi_{\mathcal{L}}, e'_k\Pi'_{\mathcal{L}}\Omega_{\mathcal{L}\mathcal{U}}H]' \\
&= [e'_k\Pi'_{\mathcal{L}}\Theta_{\mathcal{L}\mathcal{L}}\Pi_{\mathcal{L}}P\Pi^T_{\mathcal{L}}\Theta_{\mathcal{L}\mathcal{L}}\Pi_{\mathcal{L}}, e'_k\Pi'_{\mathcal{L}}\Theta_{\mathcal{L}\mathcal{L}}\Pi_{\mathcal{L}}P\Pi^T_{\mathcal{U}}\Theta_{\mathcal{U}\mathcal{U}}H]' \\
&= [\Pi'_{\mathcal{L}}\Theta_{\mathcal{L}\mathcal{L}}\Pi_{\mathcal{L}}, H'\Pi'_{\mathcal{U}}\Theta_{\mathcal{U}\mathcal{U}}]'P\Pi'_{\mathcal{L}}\Theta_{\mathcal{L}\mathcal{L}}\Pi_{\mathcal{L}}e_k \\
&= [G_{\mathcal{L}\mathcal{L}}, Q'G_{\mathcal{U}\mathcal{U}}]'P\Pi'_{\mathcal{L}}\Theta_{\mathcal{L}\mathcal{L}}\Pi_{\mathcal{L}}e_k \tag{13}
\end{aligned}$$

Notice that

$$(\Pi'_{\mathcal{L}}\Theta_{\mathcal{L}\mathcal{L}}\Pi_{\mathcal{L}})_{kl} = \mathbf{1}_{(k)}\Theta_{\mathcal{L}\mathcal{L}}\mathbf{1}_{(l)} = \begin{cases} \|\theta_{\mathcal{L}}^{(k)}\|_1, & k = l \\ 0, & k \neq l \end{cases}$$

In other words,

$$\Pi'_{\mathcal{L}}\Theta_{\mathcal{L}\mathcal{L}}\Pi_{\mathcal{L}} = diag\left(\|\theta_{\mathcal{L}}^{(1)}\|_1, ..., \|\theta_{\mathcal{L}}^{(K)}\|_1\right)$$

Hence

$$f(\Omega^{(k)};H) = \|\theta_{\mathcal{L}}^{(k)}\|_1[G_{\mathcal{L}\mathcal{L}}, Q'G_{\mathcal{U}\mathcal{U}}]'Pe_k$$

Similarly,

$$f(\mathbb{E}X;H) = \theta^*[G_{\mathcal{L}\mathcal{L}}, Q'G_{\mathcal{U}\mathcal{U}}]'Pe_{k^*}$$

Therefore,

$$\begin{aligned}
\langle f(\Omega^{(k)};H), f(\mathbb{E}X;H)\rangle &= (\|\theta_{\mathcal{L}}^{(k)}\|_1[G_{\mathcal{L}\mathcal{L}}, Q'G_{\mathcal{U}\mathcal{U}}]'Pe_k)'\theta^*[G_{\mathcal{L}\mathcal{L}}, Q'G_{\mathcal{U}\mathcal{U}}]'Pe_{k^*} \\
&= \theta^*\|\theta_{\mathcal{L}}^{(k)}\|_1 e'_k P[G_{\mathcal{L}\mathcal{L}}, G_{\mathcal{U}\mathcal{U}}Q][G_{\mathcal{L}\mathcal{L}}, Q'G_{\mathcal{U}\mathcal{U}}]'Pe_{k^*} \\
&= \theta^*\|\theta_{\mathcal{L}}^{(k)}\|_1 e'_k P\Big(G^2_{\mathcal{L}\mathcal{L}} + G_{\mathcal{U}\mathcal{U}}QQ'G_{\mathcal{U}\mathcal{U}}\Big)Pe_{k^*} \\
&= \theta^*\|\theta_{\mathcal{L}}^{(k)}\|_1 e'_k Me_{k^*} \\
&= \theta^*\|\theta_{\mathcal{L}}^{(k)}\|_1 M_{kk^*} \tag{14}
\end{aligned}$$

Similarly,

$$\|f(\Omega^{(k)};H)\| = \sqrt{\langle f(\Omega^{(k)};H), f(\Omega^{(k)};H)\rangle} = \|\theta_{\mathcal{L}}^{(k)}\|_1\sqrt{M_{kk}} \tag{15}$$

$$\|f(\mathbb{E}X;H)\| = \sqrt{\langle f(\mathbb{E}X;H), f(\mathbb{E}X;H)\rangle} = \theta^*\sqrt{M_{k^*k^*}} \tag{16}$$

Hence,

$$
\begin{aligned}
\psi_k(H) &= \psi\Big( f(\Omega^{(k)}; H),\ f(\mathbb{E}X; H) \Big) \\
&= \arccos\left( \frac{\langle f(\Omega^{(k)}; H), f(\mathbb{E}X; H) \rangle}{\|f(\Omega^{(k)}; H)\| \|f(\mathbb{E}X; H)\|} \right) \\
&= \arccos\left( \frac{\theta^* \|\theta_{\mathcal{L}}^{(k)}\|_1 M_{kk^*}}{\|\theta_{\mathcal{L}}^{(k)}\|_1 \sqrt{M_{kk}} \theta^* \sqrt{M_{k^*k^*}}} \right) \\
&= \arccos\left( \frac{M_{kk^*}}{\sqrt{M_{kk}} \sqrt{M_{k^*k^*}}} \right)
\end{aligned}
\tag{17}
$$

$\square$

## G  PROOF OF THEOREM 1

**Theorem 1.** *Consider the DCBM model where (8)-(9) hold. Let $k^*$ denote the true community label of the new node. Suppose $\hat{\Pi}_{\mathcal{U}}$ is $b_0$-correct, for a constant $b_0 \in (0,1)$. When $b_0$ is properly small, there exists a constant $c_0 > 0$, which does not depend on $b_0$, such that $\psi_{k^*}(\hat{\Pi}_{\mathcal{U}}) = 0$ and $\min_{k \neq k^*}\{\psi_k(\hat{\Pi}_{\mathcal{U}})\} \geq c_0 \beta_n$.*

*Proof.* Define $G_{\mathcal{LL}} = \Pi_{\mathcal{L}}' \Theta_{\mathcal{LL}} \Pi_{\mathcal{L}}$, $G_{\mathcal{UU}} = \Pi_{\mathcal{U}}' \Theta_{\mathcal{UU}} \Pi_{\mathcal{U}}$, $Q = G_{\mathcal{UU}}^{-1} \Pi_{\mathcal{U}}' \Theta_{\mathcal{UU}} \hat{\Pi}_{\mathcal{U}}$, and $M = P\Big( G_{\mathcal{LL}}^2 + G_{\mathcal{UU}} Q Q' G_{\mathcal{UU}} \Big) P$ as in Lemma 1. According to Lemma 1,

$$
\psi_k(\hat{\Pi}_{\mathcal{U}}) = \arccos\left( \frac{M_{kk^*}}{\sqrt{M_{kk}} \sqrt{M_{k^*k^*}}} \right)
$$

Hence,

$$
\psi_{k^*}(\hat{\Pi}_{\mathcal{U}}) = \arccos\left( \frac{M_{k^*k^*}}{\sqrt{M_{k^*k^*}} \sqrt{M_{k^*k^*}}} \right) = \arccos 1 = 0
$$

When $k \neq k^*$, according to Lemma 3,

$$
\begin{aligned}
\psi_k(\hat{\Pi}_{\mathcal{U}}) &= 2 \cdot \frac{1}{2} \psi_k(\hat{\Pi}_{\mathcal{U}}) \\
&\geq 2 \sin \frac{1}{2} \psi_k(\hat{\Pi}_{\mathcal{U}}) \\
&= 2\sqrt{ \frac{1 - \cos \psi_k(\hat{\Pi}_{\mathcal{U}})}{2} } \\
&= \sqrt{ 2\Big( 1 - \cos \arccos\Big( \frac{M_{kk^*}}{\sqrt{M_{kk}} \sqrt{M_{k^*k^*}}} \Big) \Big) } \\
&= \sqrt{ 2\Big( 1 - \frac{M_{kk^*}}{\sqrt{M_{kk}} \sqrt{M_{k^*k^*}}} \Big) }
\end{aligned}
\tag{18}
$$

Let $D_M = \mathrm{diag}(M_{11}, ..., M_{KK})$, $\tilde{M} = D_M^{-\frac{1}{2}} M D_M^{-\frac{1}{2}}$. Then

$$
\begin{aligned}
&(e_k - e_{k^*})' \tilde{M} (e_k - e_{k^*}) \\
&= \tilde{M}_{kk} + \tilde{M}_{k^*k^*} - \tilde{M}_{kk^*} - \tilde{M}_{k^*k} \\
&= \frac{M_{kk}}{\sqrt{M_{kk}} \sqrt{M_{kk}}} + \frac{M_{k^*k^*}}{\sqrt{M_{k^*k^*}} \sqrt{M_{k^*k^*}}} - \frac{M_{kk^*}}{\sqrt{M_{kk}} \sqrt{M_{k^*k^*}}} - \frac{M_{k^*k}}{\sqrt{M_{k^*k^*}} \sqrt{M_{kk}}} \\
&= 2\Big( 1 - \frac{M_{kk^*}}{\sqrt{M_{kk}} \sqrt{M_{k^*k^*}}} \Big)
\end{aligned}
\tag{19}
$$

Hence,

$$\psi_k(\hat{\Pi}_{\mathcal{U}}) \geq \sqrt{(e_k - e_{k^*})'\tilde{M}(e_k - e_{k^*})} \tag{20}$$

$\tilde{M}$ is affected by $\hat{\Pi}_{\mathcal{U}}$ and is complicated to evaluate directly. Hence, we would first evaluate its oracle version and then reduce the noisy version to the oracle version.

Define the oracle version of $M$ as follow, where $\hat{\Pi}_{\mathcal{U}}$ is replaced by $\Pi_{\mathcal{U}}$

$$M^{(0)} = P\Big(G_{\mathcal{LL}}^2 + G_{\mathcal{UU}}^2\Big)P$$

Similarly, define the oracle version of $D_M$, $D_{M^{(0)}} = \text{diag}(M_{11}^{(0)}, ..., M_{KK}^{(0)})$, and the oracle version of $\tilde{M}$, $\tilde{M}^{(0)} = D_{M^{(0)}}^{-\frac{1}{2}} M^{(0)} D_{M^{(0)}}^{-\frac{1}{2}}$

**Oracle Case**  We first study the oracle case $|\alpha'\tilde{M}^{(0)}\alpha|$.

Since $G_{\mathcal{LL}} = \Pi_{\mathcal{L}}'\Theta_{\mathcal{LL}}\Pi_{\mathcal{L}} = \text{diag}\Big(\|\theta_{\mathcal{L}}^{(1)}\|_1, ..., \|\theta_{\mathcal{L}}^{(K)}\|_1\Big)$, $G_{\mathcal{UU}} = \Pi_{\mathcal{U}}'\Theta_{\mathcal{UU}}\Pi_{\mathcal{U}} = \text{diag}\Big(\|\theta_{\mathcal{U}}^{(1)}\|_1, ..., \|\theta_{\mathcal{U}}^{(K)}\|_1\Big)$, which indicates that $G_{\mathcal{LL}}^2 + G_{\mathcal{UU}}^2 = \text{diag}\Big(\Big(\|\theta_{\mathcal{L}}^{(k)}\|_1^2 + \|\theta_{\mathcal{U}}^{(k)}\|_1^2\Big)_{k=1}^K\Big)$, for any vector $\alpha \in \mathbb{R}^k$,

$$
\begin{aligned}
|\alpha'\tilde{M}^{(0)}\alpha| &= |\alpha' D_{M^{(0)}}^{-\frac{1}{2}} M^{(0)} D_{M^{(0)}}^{-\frac{1}{2}} \alpha| \\
&= |\alpha' D_{M^{(0)}}^{-\frac{1}{2}} P\Big(G_{\mathcal{LL}}^2 + G_{\mathcal{UU}}^2\Big) P D_{M^{(0)}}^{-\frac{1}{2}} \alpha| \\
&\geq \|P D_{M^{(0)}}^{-\frac{1}{2}} \alpha\|^2 \min_k (\|\theta_{\mathcal{L}}^{(k)}\|_1^2 + \|\theta_{\mathcal{U}}^{(k)}\|_1^2) \\
\text{(Cauchy-Schwartz Inequality)} \quad &\geq |\alpha' D_{M^{(0)}}^{-\frac{1}{2}} P P' D_{M^{(0)}}^{-\frac{1}{2}} \alpha| \min_k \frac{1}{2} (\|\theta_{\mathcal{L}}^{(k)}\|_1 + \|\theta_{\mathcal{U}}^{(k)}\|_1)^2 \\
&\geq \frac{1}{2}\lambda_{\min}(P)^2 \|D_{M^{(0)}}^{-\frac{1}{2}} \alpha\|^2 \min_k (\|\theta^{(k)}\|_1)^2 \\
\text{(Condition (8))} \quad &\geq \frac{1}{2}\beta_n^2 |\alpha D_{M^{(0)}}^{-1} \alpha| \min_k (\|\theta^{(k)}\|_1)^2 \\
&\geq \frac{1}{2}\beta_n^2 \|\alpha\|^2 \min_k (\|\theta_{\mathcal{L}}^{(k)}\|_1^2 + \|\theta_{\mathcal{U}}^{(k)}\|_1^2)^{-1} \min_k (\|\theta^{(k)}\|_1)^2 \\
&\geq \frac{1}{2}\beta_n^2 \|\alpha\|^2 \min_k \Big[(\|\theta_{\mathcal{L}}^{(k)}\|_1 + \|\theta_{\mathcal{U}}^{(k)}\|_1)^2\Big]^{-1} (\min_k \|\theta^{(k)}\|_1)^2 \\
&= \frac{1}{2}\beta_n^2 \|\alpha\|^2 \left(\frac{\min_k \|\theta^{(k)}\|_1}{\max_k \|\theta^{(k)}\|_1}\right)^2 \\
\text{(Condition (9))} \quad &\geq \frac{\beta_n^2 \|\alpha\|^2}{2C_2^2} \tag{21}
\end{aligned}
$$

It remains to study the noisy case. We reduce the noisy case to the oracle case through the following lemma.

**Lemma 6.** *Denote*

$$C_5 = 8K^2\sqrt{K}C_2^2 b_0 \frac{\|\theta_{\mathcal{U}}\|_1}{\|\theta\|_1} \tag{22}$$

*Suppose that $C_5 \leq \frac{1}{4}$. Then, for any vector $\alpha \in \mathbb{R}^k$,*

$$|\alpha'\tilde{M}\alpha| \geq \frac{1 - 3C_5}{1 - C_5}|\alpha'\tilde{M}^{(0)}\alpha| \geq \frac{1}{3}|\alpha'\tilde{M}^{(0)}\alpha| \tag{23}$$

The proof of Lemma 6 is quite tedious and we would defer it to the end of this section.

Set $b_0 \leq \frac{1}{32K^2\sqrt{K}C_2^2}$, then $C_5 \leq \frac{\|\theta_\mathcal{U}\|_1}{4\|\theta\|_1} \leq \frac{1}{4}$. As a result, combining (21) with Lemma 6, we have for any vector $\alpha \in \mathbb{R}^k$,

$$|\alpha'\tilde{M}\alpha| \geq \frac{1}{3}|\alpha'\tilde{M}^{(0)}\alpha| \geq \frac{\beta_n^2\|\alpha\|^2}{6C_2^2} \tag{24}$$

Hence, take $\alpha = e_k - e_{k^*}$ in (24) and combine it with (20), we obtain that for any $k \in [K]$

$$\psi_k(\hat{\Pi}_\mathcal{U}) \geq \sqrt{(e_k - e_{k^*})'\tilde{M}(e_k - e_{k^*})} \geq \sqrt{\frac{1}{6C_2^2}\beta_n^2\|e_k - e_{k^*}\|^2} = \sqrt{\frac{1}{3C_2^2}}\beta_n \tag{25}$$

Therefore, set $c_0 = \sqrt{\frac{1}{3C_2^2}}$, we have

$$\min_{k \neq k^*}\{\psi_k(\hat{\Pi}_\mathcal{U})\} \geq c_0\beta_n \tag{26}$$

In all, when $b_0$ is properly small such that $b_0 \leq \frac{1}{32K^2\sqrt{K}C_2^2}$, there exists constant $c_0 = \sqrt{\frac{1}{3C_2^2}} > 0$ not depending on $b_0$ such that $\psi_{k^*}(\hat{\Pi}_\mathcal{U}) = 0$ and $\min_{k \neq k^*}\{\psi_k(\hat{\Pi}_\mathcal{U})\} \geq c_0\beta_n$. $\qquad\square$

### G.1 PROOF OF LEMMA 6

*Proof.* For any vector $\alpha \in \mathbb{R}^k$,

$$\begin{aligned}
|\alpha'\tilde{M}\alpha - \alpha'\tilde{M}^{(0)}\alpha| &= |\alpha'D_M^{-\frac{1}{2}}MD_M^{-\frac{1}{2}}\alpha - \alpha'D_{M^{(0)}}^{-\frac{1}{2}}M^{(0)}D_{M^{(0)}}^{-\frac{1}{2}}\alpha| \\
&= |\alpha'D_M^{-\frac{1}{2}}(M - M^{(0)})D_M^{-\frac{1}{2}}\alpha \\
&\quad + \alpha'D_M^{-\frac{1}{2}}M^{(0)}D_M^{-\frac{1}{2}}\alpha - \alpha'D_{M^{(0)}}^{-\frac{1}{2}}M^{(0)}D_{M^{(0)}}^{-\frac{1}{2}}\alpha| \\
&\leq |\alpha'D_M^{-\frac{1}{2}}(M - M^{(0)})D_M^{-\frac{1}{2}}\alpha| \\
&\quad + |\alpha'\left(D_M^{-\frac{1}{2}}M^{(0)}D_M^{-\frac{1}{2}} - D_{M^{(0)}}^{-\frac{1}{2}}M^{(0)}D_{M^{(0)}}^{-\frac{1}{2}}\right)\alpha|
\end{aligned} \tag{27}$$

The first part on the RHS of (27),

$$\begin{aligned}
|\alpha'D_M^{-\frac{1}{2}}(M - M^{(0)})D_M^{-\frac{1}{2}}\alpha| &= |\alpha'D_M^{-\frac{1}{2}}P\Pi_\mathcal{U}'\Theta_{\mathcal{U}\mathcal{U}}(\hat{\Pi}_\mathcal{U}\hat{\Pi}_\mathcal{U}' - \Pi_\mathcal{U}\Pi_\mathcal{U}')\Theta_{\mathcal{U}\mathcal{U}}\Pi_\mathcal{U}PD_M^{-\frac{1}{2}}\alpha| \\
&\leq \|PD_M^{-\frac{1}{2}}\alpha\|^2\|\Pi_\mathcal{U}'\Theta_{\mathcal{U}\mathcal{U}}(\hat{\Pi}_\mathcal{U}\hat{\Pi}_\mathcal{U}' - \Pi_\mathcal{U}\Pi_\mathcal{U}')\Theta_{\mathcal{U}\mathcal{U}}\Pi_\mathcal{U}\|_2 \\
&\leq \sqrt{K}\|PD_M^{-\frac{1}{2}}\alpha\|^2\|\Pi_\mathcal{U}'\Theta_{\mathcal{U}\mathcal{U}}(\hat{\Pi}_\mathcal{U}\hat{\Pi}_\mathcal{U}' - \Pi_\mathcal{U}\Pi_\mathcal{U}')\Theta_{\mathcal{U}\mathcal{U}}\Pi_\mathcal{U}\|_\infty
\end{aligned} \tag{28}$$

Denote $G^{(d)} = \Pi_\mathcal{U}'\Theta_{\mathcal{U}\mathcal{U}}(\hat{\Pi}_\mathcal{U}\hat{\Pi}_\mathcal{U}' - \Pi_\mathcal{U}\Pi_\mathcal{U}')\Theta_{\mathcal{U}\mathcal{U}}\Pi_\mathcal{U}$. Define $\eta_{l\tilde{l}} = \sum_{i \in \mathcal{U}, \pi_i = e_l, \hat{\pi}_i = e_{\tilde{l}}}\theta_i$. In other words, $\eta_{l\tilde{l}}$ is the sum of the degree heterogeneity parameters of all the nodes in $\mathcal{U}$ with true label $l$ and estimated label $\tilde{l}$.

Then,

$$(\Pi_\mathcal{U}'\Theta_{\mathcal{U}\mathcal{U}}\hat{\Pi}_\mathcal{U})_{l\tilde{l}} = \eta_{l\tilde{l}}$$

$$(\Pi_\mathcal{U}'\Theta_{\mathcal{U}\mathcal{U}}\Pi_\mathcal{U})_{l\tilde{l}} = \sum_{i \in \mathcal{U}, \pi_i = e_l, \pi_i = e_{\tilde{l}}}\theta_i = I_{l=\tilde{l}}\sum_{s \in [K]}\eta_{ls}$$

where $I_{l=\tilde{l}}$ is the indicator function of event $\{l = \tilde{l}\}$.

Hence,

$$G_{l\tilde{l}}^{(d)} = (\Pi'_{\mathcal{U}} \Theta_{\mathcal{U}\mathcal{U}} (\hat{\Pi}_{\mathcal{U}} \hat{\Pi}'_{\mathcal{U}} - \Pi_{\mathcal{U}} \Pi'_{\mathcal{U}}) \Theta_{\mathcal{U}\mathcal{U}} \Pi_{\mathcal{U}})_{l\tilde{l}}$$

$$= ((\Pi'_{\mathcal{U}} \Theta_{\mathcal{U}\mathcal{U}} \hat{\Pi}_{\mathcal{U}})(\Pi'_{\mathcal{U}} \Theta_{\mathcal{U}\mathcal{U}} \hat{\Pi}_{\mathcal{U}})')_{l\tilde{l}} - ((\Pi'_{\mathcal{U}} \Theta_{\mathcal{U}\mathcal{U}} \Pi_{\mathcal{U}})(\Pi'_{\mathcal{U}} \Theta_{\mathcal{U}\mathcal{U}} \Pi_{\mathcal{U}})')_{l\tilde{l}}$$

$$= \sum_{s \in [K]} \eta_{ls} \eta_{\tilde{l}s} - I_{l=\tilde{l}} (\sum_{s \in [K]} \eta_{ls})^2 \tag{29}$$

Since $\hat{\Pi}_{\mathcal{U}}$ is $b_0$ correct, there exists permutation $T$ of $K$ columns of $\hat{\Pi}_{\mathcal{U}}$ such that $\left( \sum_{i \in \mathcal{U}} \theta_i \cdot 1\{T\hat{\pi}_i \neq \pi_i\} \right) \leq b_0 \|\theta\|_1$.

Let $r = r(l)$ satisfies $e_r = T^{-1} e_l$.

When $l = \tilde{l}$, we have

$$|G_{l\tilde{l}}^{(d)}| = |\sum_{s \in [K]} \eta_{ls}^2 - (\sum_{s \in [K]} \eta_{ls})^2|$$

$$= (\sum_{s \in [K]} \eta_{ls})^2 - \sum_{s \in [K]} \eta_{ls}^2$$

$$\leq (\sum_{s \in [K]} \eta_{ls})^2 - \eta_{lr}^2$$

$$= (\sum_{s \neq r} \eta_{ls})(\eta_{lr} + \sum_{s \in [K]} \eta_{ls})$$

$$\leq 2(\sum_{s \neq r} \eta_{ls})(\sum_{s \in [K]} \eta_{ls}) \tag{30}$$

When $l \neq \tilde{l}$, we have

$$|G_{l\tilde{l}}^{(d)}| = |\sum_{s \in [K]} \eta_{ls} \eta_{\tilde{l}s}|$$

$$= \sum_{s \in [K]} \eta_{ls} \eta_{\tilde{l}s} \tag{31}$$

Therefore,

$$\|G^{(d)}\|_\infty \leq \sum_{l,\tilde{l} \in [K]} |G_{l\tilde{l}}^{(d)}|$$

$$= \sum_{l \in [K]} |G_{ll}^{(d)}| + \sum_{l \neq \tilde{l}} |G_{ll}^{(d)}|$$

$$\leq \sum_{l \in [K]} \left( 2(\sum_{s \neq r} \eta_{ls})(\sum_{s \in [K]} \eta_{ls}) \right) + \sum_{l \neq \tilde{l}} \sum_{s \in [K]} \eta_{ls} \eta_{\tilde{l}s}$$

$$\leq 2 \max_l \left( \sum_{s \in [K]} \eta_{ls} \right) \left( \sum_{l \in [K], s \neq r(l)} \eta_{ls} \right) + \sum_{s \in [K]} \sum_{l \neq \tilde{l}} \eta_{ls} \eta_{\tilde{l}s}$$

$$= 2 \max_l \left( \sum_{s \in [K]} \eta_{ls} \right) \left( \sum_{l \in [K], s \neq r(l)} \eta_{ls} \right) + \sum_{s \in [K]} \left[ (\sum_{l \in [k]} \eta_{ls})^2 - \sum_{l \in [k]} \eta_{ls}^2 \right]$$

$$\leq 2 \max_l \left( \sum_{s \in [K]} \eta_{ls} \right) \left( \sum_{l \in [K], s \neq r(l)} \eta_{ls} \right) + \sum_{s \in [K]} \left[ (\sum_{l \in [k]} \eta_{ls})^2 - \sum_{r(l)=s} \eta_{ls}^2 \right]$$

$$= 2 \max_l \left( \sum_{s \in [K]} \eta_{ls} \right) \left( \sum_{l \in [K], s \neq r(l)} \eta_{ls} \right) + \sum_{s \in [K]} \left[ (\sum_{r(l) \neq s} \eta_{ls})(\sum_{l \in [K]} \eta_{ls} + \sum_{r(l)=s} \eta_{ls}) \right]$$

$$\leq 2\max_l\Big(\sum_{s\in[K]}\eta_{ls}\Big)\Big(\sum_{l\in[K],s\neq r(l)}\eta_{ls}\Big)+2\sum_{s\in[K]}\Big[\Big(\sum_{r(l)\neq s}\eta_{ls}\Big)\Big(\sum_{l\in[K]}\eta_{ls}\Big)\Big]$$

$$\leq 2\max_l\Big(\sum_{s\in[K]}\eta_{ls}\Big)\Big(\sum_{l\in[K],s\neq r(l)}\eta_{ls}\Big)+2\max_s\Big(\sum_{l\in[K]}\eta_{ls}\Big)\sum_{s\in[K]}\sum_{r(l)\neq s}\eta_{ls}$$

$$=2\Big(\sum_{l\in[K],s\neq r(l)}\eta_{ls}\Big)\Big(\max_l\Big(\sum_{s\in[K]}\eta_{ls}\Big)+\max_s\Big(\sum_{l\in[K]}\eta_{ls}\Big)\Big)$$

$$\leq 2\Big(\sum_{l\in[K],s\neq r(l)}\eta_{ls}\Big)\Big(\sum_{l\in[K]}\sum_{s\in[K]}\eta_{ls}+\sum_{s\in[K]}\sum_{l\in[K]}\eta_{ls}\Big)$$

$$=4\|\theta_{\mathcal{U}}\|_1\Big(\sum_{l\in[K],s\neq r(l)}\eta_{ls}\Big) \tag{32}$$

Recall that $T$ satisfies $\big(\sum_{i\in\mathcal{U}}\theta_i\cdot 1\{T\hat{\pi}_i\neq\pi_i\}\big)\leq b_0\|\theta\|_1$, hence

$$\sum_{l\in[K],s\neq r(l)}\eta_{ls}=\sum_{\substack{l\in[K],s\neq r(l),\\ i\in\mathcal{U},\pi_i=e_l,\hat{\pi}_i=e_s}}\theta_i=\sum_{i\in\mathcal{U}}\theta_i\cdot 1\{T\hat{\pi}_i\neq\pi_i\}\leq b_0\|\theta\|_1$$

Therefore,

$$\|G^{(d)}\|_\infty\leq 4b_0\|\theta_{\mathcal{U}}\|_1\|\theta\|_1 \tag{33}$$

Plugging (33) into (28), we obtain

$$|\alpha'D_M^{-\frac{1}{2}}(M-M^{(0)})D_M^{-\frac{1}{2}}\alpha|\leq 4\sqrt{K}b_0\|\theta_{\mathcal{U}}\|_1\|\theta\|_1\|PD_M^{-\frac{1}{2}}\alpha\|^2 \tag{34}$$

On the other hand,

$$|\alpha'D_M^{-\frac{1}{2}}M^{(0)}D_M^{-\frac{1}{2}}\alpha|=|\alpha'D_M^{-\frac{1}{2}}P\Big(G_{\mathcal{LL}}^2+G_{\mathcal{UU}}^2\Big)PD_M^{-\frac{1}{2}}\alpha|$$

Since $G_{\mathcal{LL}}=\Pi'_{\mathcal{L}}\Theta_{\mathcal{LL}}\Pi_{\mathcal{L}}=\mathrm{diag}(\|\theta_{\mathcal{L}}^{(1)}\|_1,...,\|\theta_{\mathcal{L}}^{(K)}\|_1)$, $G_{\mathcal{UU}}=\Pi'_{\mathcal{U}}\Theta_{\mathcal{UU}}\Pi_{\mathcal{U}}=\mathrm{diag}(\|\theta_{\mathcal{U}}^{(1)}\|_1,...,\|\theta_{\mathcal{U}}^{(K)}\|_1)$,

$$|\alpha'D_M^{-\frac{1}{2}}M^{(0)}D_M^{-\frac{1}{2}}\alpha|\geq\|PD_M^{-\frac{1}{2}}\alpha\|^2\min_k(\|\theta_{\mathcal{L}}^{(k)}\|_1^2+\|\theta_{\mathcal{U}}^{(k)}\|_1^2)$$

$$\text{(Cauchy-Schwartz Inequality)}\geq\|PD_M^{-\frac{1}{2}}\alpha\|^2\min_k\frac{1}{2}(\|\theta_{\mathcal{L}}^{(k)}\|_1+\|\theta_{\mathcal{U}}^{(k)}\|_1)^2$$

$$=\frac{1}{2}\|PD_M^{-\frac{1}{2}}\alpha\|^2\min_k\|\theta^{(k)}\|_1^2$$

Recall condition (9) in the main paper,

$$\frac{\max_{1\leq k\leq K}\{\|\theta^{(k)}\|_1\}}{\min_{1\leq k\leq K}\{\|\theta^{(k)}\|_1\}}\leq C_2$$

Hence

$$|\alpha'D_M^{-\frac{1}{2}}M^{(0)}D_M^{-\frac{1}{2}}\alpha|\geq\frac{1}{2}\|PD_M^{-\frac{1}{2}}\alpha\|^2(\frac{1}{C_2}\max_k\|\theta^{(k)}\|_1)^2$$

$$\geq\frac{1}{2}\|PD_M^{-\frac{1}{2}}\alpha\|^2(\frac{1}{KC_2}\|\theta\|_1)^2$$

$$=\frac{1}{2K^2C_2^2}\|PD_M^{-\frac{1}{2}}\alpha\|^2\|\theta\|_1^2 \tag{35}$$

Comparing (35) with (34), we obtain

$$|\alpha'D_M^{-\frac{1}{2}}(M-M^{(0)})D_M^{-\frac{1}{2}}\alpha|$$

$$
\leq 8K^2\sqrt{K}C_2^2 b_0 \frac{\|\theta_{\mathcal{U}}\|_1}{\|\theta\|_1} |\alpha' D_M^{-\frac{1}{2}} M^{(0)} D_M^{-\frac{1}{2}} \alpha|
$$

$$
\leq C_5 |\alpha' D_{M^{(0)}}^{-\frac{1}{2}} M^{(0)} D_{M^{(0)}}^{-\frac{1}{2}} \alpha|
$$

$$
+ C_5 |\alpha' \Big( D_M^{-\frac{1}{2}} M^{(0)} D_M^{-\frac{1}{2}} - D_{M^{(0)}}^{-\frac{1}{2}} M^{(0)} D_{M^{(0)}}^{-\frac{1}{2}} \Big) \alpha| \tag{36}
$$

Consequently, we bound the first part of (27) by the second part of (27). It remains to bound the second part of (27).

Since $D_M$, $D_{M^{(0)}}$, $M^{(0)}$ are all diagonal matrices, we can rewrite the second part on the LHS of (27) as follows:

$$
|\alpha' \Big( D_M^{-\frac{1}{2}} M^{(0)} D_M^{-\frac{1}{2}} - D_{M^{(0)}}^{-\frac{1}{2}} M^{(0)} D_{M^{(0)}}^{-\frac{1}{2}} \Big) \alpha|
$$

$$
= |\alpha' D_{M^{(0)}}^{-\frac{1}{2}} (M^{(0)})^{\frac{1}{2}} \Big( (M^{(0)})^{-\frac{1}{2}} D_{M^{(0)}}^{\frac{1}{2}} D_M^{-\frac{1}{2}} M^{(0)} D_M^{-\frac{1}{2}} D_{M^{(0)}}^{\frac{1}{2}} (M^{(0)})^{-\frac{1}{2}} - 1 \Big) (M^{(0)})^{\frac{1}{2}} D_{M^{(0)}}^{-\frac{1}{2}} \alpha|
$$

$$
= |\alpha' D_{M^{(0)}}^{-\frac{1}{2}} (M^{(0)})^{\frac{1}{2}} \Big( D_{M^{(0)}} D_M^{-1} - 1 \Big) (M^{(0)})^{\frac{1}{2}} D_{M^{(0)}}^{-\frac{1}{2}} \alpha|
$$

$$
\leq \lambda_{\max} \Big( D_{M^{(0)}} D_M^{-1} - 1 \Big) \|(M^{(0)})^{\frac{1}{2}} D_{M^{(0)}}^{-\frac{1}{2}} \alpha\|^2
$$

$$
= \max_{k \in [K]} \left| \frac{M_{kk}^{(0)}}{M_{kk}} - 1 \right| \cdot \|(M^{(0)})^{\frac{1}{2}} D_{M^{(0)}}^{-\frac{1}{2}} \alpha\|^2
$$

$$
= \max_{k \in [K]} \left| \frac{1}{\frac{M_{kk}^{(0)}}{(M^{(0)} - M)_{kk}} - 1} \right| \cdot |\alpha' D_{M^{(0)}}^{-\frac{1}{2}} M^{(0)} D_{M^{(0)}}^{-\frac{1}{2}} \alpha| \tag{37}
$$

Notice that for any $k \in [K]$

$$
\left| \frac{M_{kk}^{(0)}}{(M^{(0)} - M)_{kk}} \right| = \frac{\left| e_k' M^{(0)} e_k \right|}{\left| e_k' (M^{(0)} - M) e_k \right|}
$$

$$
= \frac{\left| e_k' P \Big( G_{\mathcal{L}\mathcal{L}}^2 + G_{\mathcal{U}\mathcal{U}}^2 \Big) P e_k \right|}{\left| e_k' P G^{(d)} P e_k \right|}
$$

$$
\geq \frac{\|P e_k\|^2 \min_k(\|\theta_{\mathcal{L}}^{(k)}\|_1^2 + \|\theta_{\mathcal{U}}^{(k)}\|_1^2)}{\|P e_k\|^2 \|G^{(d)}\|_2}
$$

$$
\text{(Cauchy-Schwartz Inequality)} \geq \frac{\min_k \frac{1}{2}(\|\theta_{\mathcal{L}}^{(k)}\|_1 + \|\theta_{\mathcal{U}}^{(k)}\|_1)^2}{\sqrt{K}\|G^{(d)}\|_\infty}
$$

$$
\text{(Plugging in (33))} \geq \frac{\min_k(\|\theta^{(k)}\|_1)^2}{8\sqrt{K}b_0\|\theta_{\mathcal{U}}\|_1\|\theta\|_1}
$$

$$
\text{(Condition (9))} \geq \frac{(\max_k \frac{1}{C_2}\|\theta^{(k)}\|_1)^2}{8\sqrt{K}b_0\|\theta_{\mathcal{U}}\|_1\|\theta\|_1}
$$

$$
\geq \frac{(\frac{1}{C_2 K}\|\theta\|_1)^2}{8\sqrt{K}b_0\|\theta_{\mathcal{U}}\|_1\|\theta\|_1}
$$

$$
= \frac{1}{C_5}
$$

$$
\geq 4 > 1 \tag{38}
$$

Plugging (38) into (37), we have

$$
|\alpha' \Big( D_M^{-\frac{1}{2}} M^{(0)} D_M^{-\frac{1}{2}} - D_{M^{(0)}}^{-\frac{1}{2}} M^{(0)} D_{M^{(0)}}^{-\frac{1}{2}} \Big) \alpha|
$$

$$
\begin{aligned}
&\leq \max_{k\in[K]} \frac{1}{\left|\frac{M_{kk}^{(0)}}{(M^{(0)}-M)_{kk}}\right|-1} \cdot |\alpha' D_{M^{(0)}}^{-\frac{1}{2}} M^{(0)} D_{M^{(0)}}^{-\frac{1}{2}}\alpha| \\
&\leq \max_{k\in[K]} \frac{1}{\frac{1}{C_5}-1} \cdot |\alpha' D_{M^{(0)}}^{-\frac{1}{2}} M^{(0)} D_{M^{(0)}}^{-\frac{1}{2}}\alpha| \\
&= \frac{C_5}{1-C_5} |\alpha' D_{M^{(0)}}^{-\frac{1}{2}} M^{(0)} D_{M^{(0)}}^{-\frac{1}{2}}\alpha|
\end{aligned} \tag{39}
$$

Combining (27), (36), and (39), we have

$$
\begin{aligned}
|\alpha'\tilde{M}\alpha - \alpha'\tilde{M}^{(0)}\alpha| &\leq |\alpha' D_M^{-\frac{1}{2}}(M - M^{(0)}) D_M^{-\frac{1}{2}}\alpha| \\
&\quad + |\alpha'\Big(D_M^{-\frac{1}{2}} M^{(0)} D_M^{-\frac{1}{2}} - D_{M^{(0)}}^{-\frac{1}{2}} M^{(0)} D_{M^{(0)}}^{-\frac{1}{2}}\Big)\alpha| \\
&\leq C_5 |\alpha' D_{M^{(0)}}^{-\frac{1}{2}} M^{(0)} D_{M^{(0)}}^{-\frac{1}{2}}\alpha| \\
&\quad + C_5 |\alpha'\Big(D_M^{-\frac{1}{2}} M^{(0)} D_M^{-\frac{1}{2}} - D_{M^{(0)}}^{-\frac{1}{2}} M^{(0)} D_{M^{(0)}}^{-\frac{1}{2}}\Big)\alpha| \\
&\quad + |\alpha'\Big(D_M^{-\frac{1}{2}} M^{(0)} D_M^{-\frac{1}{2}} - D_{M^{(0)}}^{-\frac{1}{2}} M^{(0)} D_{M^{(0)}}^{-\frac{1}{2}}\Big)\alpha| \\
&= C_5 |\alpha' D_{M^{(0)}}^{-\frac{1}{2}} M^{(0)} D_{M^{(0)}}^{-\frac{1}{2}}\alpha| \\
&\quad + (C_5 + 1)|\alpha'\Big(D_M^{-\frac{1}{2}} M^{(0)} D_M^{-\frac{1}{2}} - D_{M^{(0)}}^{-\frac{1}{2}} M^{(0)} D_{M^{(0)}}^{-\frac{1}{2}}\Big)\alpha| \\
&\leq C_5 |\alpha' D_{M^{(0)}}^{-\frac{1}{2}} M^{(0)} D_{M^{(0)}}^{-\frac{1}{2}}\alpha| \\
&\quad + (C_5 + 1)\frac{C_5}{1-C_5} |\alpha' D_{M^{(0)}}^{-\frac{1}{2}} M^{(0)} D_{M^{(0)}}^{-\frac{1}{2}}\alpha| \\
&= \frac{2C_5}{1-C_5} |\alpha'\tilde{M}^{(0)}\alpha|
\end{aligned} \tag{40}
$$

Hence for any vector $\alpha \in \mathbb{R}^k$,

$$
|\alpha'\tilde{M}\alpha| \geq |\alpha'\tilde{M}^{(0)}\alpha| - |\alpha'\tilde{M}\alpha - \alpha'\tilde{M}^{(0)}\alpha| \geq \frac{1-3C_5}{1-C_5} |\alpha'\tilde{M}^{(0)}\alpha| \tag{41}
$$

To conclude, in this subsection, we successfully reduce the noisy case $|\alpha'\tilde{M}\alpha|$ to the oracle case $|\alpha'\tilde{M}^{(0)}\alpha|$. Result (41) will also be used in the proof of other claims.

$\square$

## H  PROOF OF THEOREM 2

**Theorem 2.** *Consider the DCBM model where (8)-(9) hold. There exists constant $C > 0$, such that for any $\delta \in (0, 1/2)$, with probability $1 - \delta$, simultaneously for $1 \leq k \leq K$,*

$$
|\hat{\psi}_k(\hat{\Pi}_{\mathcal{U}}) - \psi_k(\hat{\Pi}_{\mathcal{U}})| \leq C \left( \sqrt{\frac{\log(1/\delta)}{\|\theta\|_1 \cdot \min\{\theta^*, \|\theta_{\mathcal{L}}^{(k)}\|_1\}}} + \frac{\|\theta_{\mathcal{L}}^{(k)}\|^2}{\|\theta_{\mathcal{L}}^{(k)}\|_1 \|\theta\|_1} \right).
$$

To prove Theorem 2, we need a famous concentration inequality, Bernstein inequality:

**Lemma 7** (Bernstein inequality)**.** *Suppose $X_1, ..., X_n$ are independent random variables such that $\mathbb{E}X_i = 0$, $|X_i| \leq b$ and $Var(X_i) \leq \sigma_i^2$ for all $i$. Let $\sigma^2 = n^{-1}\sum_{i=1}^n \sigma_i^2$. Then, for any $t > 0$,*

$$
\mathbb{P}\Big(n^{-1}|\sum_{i=1}^n X_i| \geq t\Big) \leq 2\exp\Big(-\frac{nt^2/2}{\sigma^2 + bt/3}\Big)
$$

The proof of Lemma 7, Bernstein inequality, can be seen in most probability textbooks such as Uspensky (1937).

*Proof.* Recall that for $k \in [K]$,

$$\hat{\psi}_k(\hat{\Pi}_{\mathcal{U}}) = \psi(f(A^{(k)}; \hat{\Pi}_{\mathcal{U}}), f(X; H)),$$

$$\psi_k(\hat{\Pi}_{\mathcal{U}}) = \psi\Big(f(\Omega^{(k)}; \hat{\Pi}_{\mathcal{U}}), \ f(\mathbb{E}X; \hat{\Pi}_{\mathcal{U}})\Big).$$

Denote $v_k = f(A^{(k)}; \hat{\Pi}_{\mathcal{U}})$, $v^* = f(X; \hat{\Pi}_{\mathcal{U}})$, $\tilde{v}_k = f(\Omega^{(k)}; \hat{\Pi}_{\mathcal{U}})$, $\tilde{v}^* = f(EX; \hat{\Pi}_{\mathcal{U}})$, $k \in [K]$. Then, by Lemma 4,

$$\begin{aligned}
\hat{\psi}_k(\hat{\Pi}_{\mathcal{U}}) = \psi(v_k, v^*) &\leq \psi(v_k, \tilde{v}_k) + \psi(\tilde{v}_k, v^*) \\
&\leq \psi(v_k, \tilde{v}_k) + \psi(\tilde{v}_k, \tilde{v}^*) + \psi(\tilde{v}^*, v^*) \\
&= \psi_k(\hat{\Pi}_{\mathcal{U}}) + \psi(v_k, \tilde{v}_k) + \psi(\tilde{v}^*, v^*)
\end{aligned}$$

Similarly,

$$\psi_k(\hat{\Pi}_{\mathcal{U}}) \leq \hat{\psi}_k(\hat{\Pi}_{\mathcal{U}}) + \psi(v_k, \tilde{v}_k) + \psi(\tilde{v}^*, v^*)$$

Therefore,

$$|\hat{\psi}_k(\hat{\Pi}_{\mathcal{U}}) - \psi_k(\hat{\Pi}_{\mathcal{U}})| \leq \psi(v_k, \tilde{v}_k) + \psi(\tilde{v}^*, v^*). \tag{42}$$

For any $\phi_1, ..., \phi_K \geq 0$.

$$\begin{aligned}
&\mathbb{P}\Big(\forall k \in [K], |\hat{\psi}_k(\hat{\Pi}_{\mathcal{U}}) - \psi_k(\hat{\Pi}_{\mathcal{U}})| \leq \phi_k\Big) \\
&= 1 - \mathbb{P}\Big(\exists k \in [K], |\hat{\psi}_k(\hat{\Pi}_{\mathcal{U}}) - \psi_k(\hat{\Pi}_{\mathcal{U}})| > \phi_k\Big) \\
&\geq 1 - \sum_{k=1}^{K} \mathbb{P}\Big(|\hat{\psi}_k(\hat{\Pi}_{\mathcal{U}}) - \psi_k(\hat{\Pi}_{\mathcal{U}})| > \phi_k\Big)
\end{aligned} \tag{43}$$

By definition of $\psi$, $\hat{\psi}_k(\hat{\Pi}_{\mathcal{U}}), \psi_k(\hat{\Pi}_{\mathcal{U}}) \in [0, \pi]$. Hence, $|\hat{\psi}_k(\hat{\Pi}_{\mathcal{U}}) - \psi_k(\hat{\Pi}_{\mathcal{U}})| \in [0, \pi]$. As a result, when $\phi_k \geq \pi$,

$$\mathbb{P}\Big(|\hat{\psi}_k(\hat{\Pi}_{\mathcal{U}}) - \psi_k(\hat{\Pi}_{\mathcal{U}})| > \phi_k\Big) = 0$$

When $\phi_k < \pi$, by (42),

$$\begin{aligned}
\mathbb{P}\Big(|\hat{\psi}_k(\hat{\Pi}_{\mathcal{U}}) - \psi_k(\hat{\Pi}_{\mathcal{U}})| > \phi_k\Big) &\leq \mathbb{P}\Big(\psi(v_k, \tilde{v}_k) + \psi(\tilde{v}^*, v^*) > \phi_k\Big) \\
&\leq \mathbb{P}\Big(\psi(v_k, \tilde{v}_k) > \frac{1}{2}\phi_k \text{ or } \psi(\tilde{v}^*, v^*) > \frac{1}{2}\phi_k\Big) \\
&\leq \mathbb{P}\Big(\psi(v_k, \tilde{v}_k) > \frac{1}{2}\phi_k\Big) + \mathbb{P}\Big(\psi(\tilde{v}^*, v^*) > \frac{1}{2}\phi_k\Big)
\end{aligned} \tag{44}$$

By lemma 5, when $\|v_k - \tilde{v}_k\| < \|\tilde{v}_k\|$

$$\psi(v_k, \tilde{v}_k) \leq \arcsin \frac{\|v_k - \tilde{v}_k\|}{\|\tilde{v}_k\|}$$

Hence, for any $\phi \in [0, \frac{\pi}{2})$, $\|v_k - \tilde{v}_k\| \leq \sin(\phi)\|\tilde{v}_k\|$ implies $\psi(v_k, \tilde{v}_k) \leq \phi$.

As a result, for any $\phi \in [0, \frac{\pi}{2})$, $\psi(v_k, \tilde{v}_k) > \phi$ implies $\|v_k - \tilde{v}_k\| > \sin(\phi)\|\tilde{v}_k\|$.

Similarly, for any $\phi \in [0, \frac{\pi}{2})$, $\psi(v^*, \tilde{v}^*) > \phi$ implies $\|v^* - \tilde{v}^*\| \geq \sin(\phi)\|\tilde{v}^*\|$.

By definition of $\phi_k$, $\phi_k \geq 0$. Hence, when $\phi_k < \pi$, $\frac{1}{2}\phi_k \in [0, \frac{\pi}{2})$. Plugging the above results into (44), we have

$$\mathbb{P}\Big(|\hat{\psi}_k(\hat{\Pi}_{\mathcal{U}}) - \psi_k(\hat{\Pi}_{\mathcal{U}})| > \phi_k\Big)$$

$$\leq \mathbb{P}\Big(\psi(v_k, \tilde{v}_k) > \frac{1}{2}\phi_k\Big) + \mathbb{P}\Big(\psi(\tilde{v}^*, v^*) > \frac{1}{2}\phi_k\Big)$$

$$\leq \mathbb{P}\Big(\|v_k - \tilde{v}_k\| \geq \sin(\frac{1}{2}\phi_k)\|\tilde{v}_k\|\Big) + \mathbb{P}\Big(\|v^* - \tilde{v}^*\| \geq \sin(\frac{1}{2}\phi_k)\|\tilde{v}^*\|\Big)$$

$$\leq \mathbb{P}\Big(\exists l \in [2K], |(v_k - \tilde{v}_k)_l| \geq \frac{1}{\sqrt{K}}\sin(\frac{1}{2}\phi_k)\|\tilde{v}_k\|\Big)$$

$$+ \mathbb{P}\Big(\exists l \in [2K], |(v^* - \tilde{v}^*)_l| \geq \frac{1}{\sqrt{K}}\sin(\frac{1}{2}\phi_k)\|\tilde{v}^*\|\Big)$$

$$\leq \sum_{l=1}^{2K} \mathbb{P}\Big(|(v_k - \tilde{v}_k)_l| \geq \frac{1}{\sqrt{K}}\sin(\frac{1}{2}\phi_k)\|\tilde{v}_k\|\Big)$$

$$+ \sum_{l=1}^{2K} \mathbb{P}\Big(|(v^* - \tilde{v}^*)_l| \geq \frac{1}{\sqrt{K}}\sin(\frac{1}{2}\phi_k)\|\tilde{v}^*\|\Big)$$

$$\tag{45}$$

Since when $\phi_k < \pi$, $\frac{1}{2}\phi_k \in [0, \frac{\pi}{2}]$, by Lemma 3, $\sin(\frac{1}{2}\phi_k) \geq \frac{2}{\pi}\frac{1}{2}\phi_k = \frac{1}{\pi}\phi_k$. Plugging back to (45), we have when $\phi_k < \pi$,

$$\mathbb{P}\Big(|\hat{\psi}_k(\hat{\Pi}_{\mathcal{U}}) - \psi_k(\hat{\Pi}_{\mathcal{U}})| > \phi_k\Big)$$

$$\leq \sum_{l=1}^{2K} \mathbb{P}\Big(|(v_k - \tilde{v}_k)_l| \geq \frac{1}{\pi\sqrt{K}}\phi_k\|\tilde{v}_k\|\Big) + \sum_{l=1}^{2K} \mathbb{P}\Big(|(v^* - \tilde{v}^*)_l| \geq \frac{1}{\pi\sqrt{K}}\phi_k\|\tilde{v}^*\|\Big) \tag{46}$$

It remains to evaluate $\mathbb{P}\Big(|(v_k - \tilde{v}_k)_l| \geq \frac{1}{\pi\sqrt{K}}\phi_k\|\tilde{v}_k\|\Big)$ and $\mathbb{P}\Big(|(v^* - \tilde{v}^*)_l| \geq \frac{1}{\pi\sqrt{K}}\phi_k\|\tilde{v}^*\|\Big)$, which are illustrated in the following two lemmas.

**Lemma 8.** *Define* $C_6 = \frac{C^2}{16\sqrt{2}\pi^2 C_2 K^2(\sqrt{K}+\frac{1}{3})}$. *When* $\phi_k \geq 2\sqrt{2}\pi C_2 K^2 \frac{\|\theta_{\mathcal{L}}^{(k)}\|^2}{\|\theta_{\mathcal{L}}^{(k)}\|_1\|\theta\|_1}$,

$$\mathbb{P}\Big(|(v_k - \tilde{v}_k)_l| \geq \frac{1}{\pi\sqrt{K}}\phi_k\|\tilde{v}_k\|\Big) \leq 2\exp\left(-\frac{C_6}{C^2}\phi_k^2\|\theta_{\mathcal{L}}^{(k)}\|_1\|\theta\|_1\right)$$

**Lemma 9.** *Define* $C_7 = \frac{C^2}{2\sqrt{2}\pi^2 C_2 K^2(\sqrt{K}+\frac{1}{3})}$. *Then,*

$$\mathbb{P}\Big(|(v^* - \tilde{v}^*)_l| \geq \frac{1}{\pi\sqrt{K}}\phi_k\|\tilde{v}^*\|\Big) \leq 2\exp\left(-\frac{C_7}{C^2}\phi_k^2\theta^*\|\theta\|_1\right)$$

The proof of Lemma 8 and 9 are quite tedious. We would defer their proofs to the end of this section.

Choose

$$\phi_k = \phi_k(C, \delta) = C\left(\sqrt{\frac{\log(1/\delta)}{\|\theta\|_1 \cdot \min\{\theta^*, \|\theta_{\mathcal{L}}^{(k)}\|_1\}}} + \frac{\|\theta_{\mathcal{L}}^{(k)}\|^2}{\|\theta_{\mathcal{L}}^{(k)}\|_1\|\theta\|_1}\right).$$

Then leveraging on Lemma 8 and 9, we have when $C \geq 2\sqrt{2}\pi K^2$,

$$\mathbb{P}\Big(|(v_k - \tilde{v}_k)_l| \geq \frac{1}{\pi\sqrt{K}}\phi_k\|\tilde{v}_k\|\Big) \leq 2\exp\left(-\frac{C_6 C^2 \log(1/\delta)\|\theta_{\mathcal{L}}^{(k)}\|_1\|\theta\|_1}{C^2\|\theta\|_1 \cdot \min\{\theta^*, \|\theta_{\mathcal{L}}^{(k)}\|_1\}}\right)$$

$$\leq 2\exp\left(-C_6 \log(1/\delta)\right)$$

$$= 2\delta^{C_6} \tag{47}$$

$$\mathbb{P}\Big(|(v^* - \tilde{v}^*)_l| \geq \frac{1}{\pi\sqrt{K}}\phi_k\|\tilde{v}^*\|\Big) \leq 2\exp\left(-\frac{C_7 C^2 \log(1/\delta)\theta^*\|\theta\|_1}{C^2\|\theta\|_1 \cdot \min\{\theta^*, \|\theta_{\mathcal{L}}^{(k)}\|_1\}}\right)$$
$$\leq 2\exp\left(-C_7 \log(1/\delta)\right)$$
$$= 2\delta^{C_7} \tag{48}$$

Plugging (47) and (48) back to (46), leveraging on the fact that $\delta \leq \frac{1}{2} < 1$, we obtain when $\phi_k < \pi$, and $C \geq 2\sqrt{2}\pi K^2$,

$$\mathbb{P}\Big(|\hat{\psi}_k(\hat{\Pi}_{\mathcal{U}}) - \psi_k(\hat{\Pi}_{\mathcal{U}})| > \phi_k\Big) \leq 4K\delta^{C_6} + 4K\delta^{C_7} \leq 8K\delta^{C_6}$$

Recall that when $\phi_k \geq \pi$,

$$\mathbb{P}\Big(|\hat{\psi}_k(\hat{\Pi}_{\mathcal{U}}) - \psi_k(\hat{\Pi}_{\mathcal{U}})| > \phi_k\Big) = 0$$

.

In all, we have that when $C \geq 2\sqrt{2}\pi K^2$,

$$\mathbb{P}\Big(|\hat{\psi}_k(\hat{\Pi}_{\mathcal{U}}) - \psi_k(\hat{\Pi}_{\mathcal{U}})| > \phi_k\Big) \leq 8K\delta^{C_6} \tag{49}$$

Substituting (49) into (43), we obtain that when $C \geq 2\sqrt{2}\pi K^2$,

$$\mathbb{P}\Big(\forall k \in [K], |\hat{\psi}_k(\hat{\Pi}_{\mathcal{U}}) - \psi_k(\hat{\Pi}_{\mathcal{U}})| \leq \phi_k\Big) \geq 1 - 8K^2\delta^{C_6} \tag{50}$$

Hence, it suffices to make $8K^2\delta^{C_6} \leq \delta$. Choose

$$C = \max\{2\sqrt{2}\pi K^2, \sqrt{16\sqrt{2}\pi^2 C_2 K^2(\sqrt{K} + \frac{1}{3})(1 + \frac{\log(8K^2)}{\log 2})}\}, \tag{51}$$

Then $C_6 - 1 \geq \frac{\log(8K^2)}{\log 2} \geq 1$. Since $\delta \leq \frac{1}{2}$,

$$\delta^{C_6 - 1} \leq \left(\frac{1}{2}\right)^{C_6 - 1} \leq \frac{1}{2^{\frac{\log(8K^2)}{\log 2}}} = \frac{1}{8K^2}$$

As a result, $8K^2\delta^{C_6} \leq \delta$.

Hence, choose $C$ as in (51), then $C >)$, and for any $\delta \in (0, 1/2)$,

$$\mathbb{P}\Big(\forall k \in [K], |\hat{\psi}_k(\hat{\Pi}_{\mathcal{U}}) - \psi_k(\hat{\Pi}_{\mathcal{U}})| \leq \phi_k\Big) \geq 1 - \delta \tag{52}$$

To conclude, there exists constant $C > 0$, such that for any $\delta \in (0, 1/2)$, with probability $1 - \delta$, simultaneously for $1 \leq k \leq K$,

$$|\hat{\psi}_k(\hat{\Pi}_{\mathcal{U}}) - \psi_k(\hat{\Pi}_{\mathcal{U}})| \leq C\sqrt{\frac{\log(1/\delta)}{\|\theta\|_1 \cdot \min\{\theta^*, \|\theta_{\mathcal{L}}^{(k)}\|_1\}}}.$$

$\square$

## H.1 PROOF OF LEMMA 8

*Proof.* When $l \in [K]$,

$$(\tilde{v}_k)_l = (f(\Omega^{(k)}; \hat{\Pi}_{\mathcal{U}}))_l = \Omega^{(k)}\mathbf{1}_{(l)} = \sum_{i \in \mathcal{C}_k \cap \mathcal{L}} \sum_{j \in \mathcal{C}_l \cap \mathcal{L}} \Omega_{ij}$$

When $l \in \{K+1, ..., 2K\}$, define

$$\hat{\mathcal{C}}_l = \{i \in \mathcal{U} : \hat{\pi}_i = e_{l-K}\}$$

, then

$$(\tilde{v}_k)_l = (f(\Omega^{(k)}; \hat{\Pi}_{\mathcal{U}}))_l = \Omega^{(k)}\left(\hat{\Pi}_{\mathcal{U}}\right)_l = \sum_{i \in \mathcal{C}_k \cap \mathcal{L}} \sum_{j \in \hat{\mathcal{C}}_l} \Omega_{ij}$$

Hence

$$\|\tilde{v}_k\| = \sqrt{\sum_{l \in [2K]} (\tilde{v}_k)_l^2}$$

$$(\text{Cauchy-Schwartz}) \geq \sqrt{\frac{1}{2K}\left(\sum_{l \in [2K]} (\tilde{v}_k)_l\right)^2}$$

$$= \frac{1}{\sqrt{2K}}\left|\sum_{l \in [2K]} (\tilde{v}_k)_l\right|$$

$$= \frac{1}{\sqrt{2K}}\left|\sum_{l=1}^{K} \sum_{i \in \mathcal{C}_k \cap \mathcal{L}} \sum_{j \in \mathcal{C}_l \cap \mathcal{L}} \Omega_{ij} + \sum_{l=K+1}^{2K} \sum_{i \in \mathcal{C}_k \cap \mathcal{L}} \sum_{j \in \hat{\mathcal{C}}_l} \Omega_{ij}\right|$$

$$= \frac{1}{\sqrt{2K}}\left|\sum_{i \in \mathcal{C}_k \cap \mathcal{L}} \sum_{j \in \mathcal{L}} \Omega_{ij} + \sum_{i \in \mathcal{C}_k \cap \mathcal{L}} \sum_{j \in \hat{\mathcal{U}}} \Omega_{ij}\right|$$

$$= \frac{1}{\sqrt{2K}} \sum_{i \in \mathcal{C}_k \cap \mathcal{L}} \sum_{j \in [n]} \Omega_{ij}$$

$$\geq \frac{1}{\sqrt{2K}} \sum_{i \in \mathcal{C}_k \cap \mathcal{L}} \sum_{j \in \mathcal{C}_k} \Omega_{ij}$$

$$= \frac{1}{\sqrt{2K}} \sum_{i \in \mathcal{C}_k \cap \mathcal{L}} \sum_{j \in \mathcal{C}_k} \theta_i \theta_j P_{kk}$$

$$(\text{Identifiability condition}) = \frac{1}{\sqrt{2K}} \sum_{i \in \mathcal{C}_k \cap \mathcal{L}} \sum_{j \in \mathcal{C}_k} \theta_i \theta_j$$

$$= \frac{1}{\sqrt{2K}} \|\theta_{\mathcal{L}}^{(k)}\|_1 \|\theta^{(k)}\|_1$$

$$\geq \frac{1}{\sqrt{2K}} \|\theta_{\mathcal{L}}^{(k)}\|_1 \min_{l \in [K]} \|\theta^{(l)}\|_1$$

$$(\text{Condition (9)}) \geq \frac{1}{C_2\sqrt{2K}} \|\theta_{\mathcal{L}}^{(k)}\|_1 \max_{l \in [K]} \|\theta^{(l)}\|_1$$

$$\geq \frac{1}{C_2 K\sqrt{2K}} \|\theta_{\mathcal{L}}^{(k)}\|_1 \|\theta\|_1 \tag{53}$$

When $l \in [K]$,

$$(v_k)_l = (f(A^{(k)}; \hat{\Pi}_{\mathcal{U}}))_l = A^{(k)}\mathbf{1}_{(l)} = \sum_{i \in \mathcal{C}_k \cap \mathcal{L}} \sum_{j \in \mathcal{C}_l \cap \mathcal{L}} A_{ij}$$

Recall that,

$$(\tilde{v}_k)_l = (f(\Omega^{(k)}; \hat{\Pi}_{\mathcal{U}}))_l = \Omega^{(k)}\mathbf{1}_{(l)} = \sum_{i \in \mathcal{C}_k \cap \mathcal{L}} \sum_{j \in \mathcal{C}_l \cap \mathcal{L}} \Omega_{ij}$$

So

$$|(v_k - \tilde{v}_k)_l| = \sum_{i \in \mathcal{C}_k \cap \mathcal{L}} \sum_{j \in \mathcal{C}_l \cap \mathcal{L}} (A_{ij} - \Omega_{ij})$$

When $l \in [K]\backslash\{k\}$, since $\hat{\Pi}_{\mathcal{U}}$ only depends on $A_{\mathcal{U}\mathcal{U}}$, it is independent of $A_{\mathcal{L}\mathcal{L}}$. Hence, given $\hat{\Pi}_{\mathcal{U}}$, $\{A_{ij} - \Omega_{ij} : i \in \mathcal{C}_k \cap \mathcal{L}, j \in \mathcal{C}_l \cap \mathcal{L}\}$ are a collection of $|\mathcal{C}_k \cap \mathcal{L}||\mathcal{C}_l \cap \mathcal{L}|$ independent random variables. Furthermore, given $\hat{\Pi}_{\mathcal{U}}$, for any $i \in \mathcal{C}_k \cap \mathcal{L}, j \in \mathcal{C}_l \cap \mathcal{L}$,

$$\mathbb{E}\Big[A_{ij} - \Omega_{ij}|\hat{\Pi}_{\mathcal{U}}\Big] = \mathbb{E}\Big[A_{ij}|\hat{\Pi}_{\mathcal{U}}\Big] - \Omega_{ij} = \Omega_{ij} - \Omega_{ij} = 0$$

Also,

$$-1 \leq -\Omega_{ij} \leq A_{ij} - \Omega_{ij} \leq A_{ij} \leq 1$$

So $|A_{ij} - \Omega_{ij}| \leq 1$. Additionally,

$$var\Big(A_{ij} - \Omega_{ij}\Big) = var\Big(A_{ij}|\hat{\Pi}_{\mathcal{U}}\Big) = \Omega_{ij}(1 - \Omega_{ij}) \leq \Omega_{ij}$$

Therefore, denote $n_{kl} = |\mathcal{C}_k \cap \mathcal{L}||\mathcal{C}_l \cap \mathcal{L}|$, by Lemma 7,

$$\mathbb{P}\Big(|(v_k - \tilde{v}_k)_l| \geq \frac{1}{\pi\sqrt{K}}\phi_k\|\tilde{v}_k\|\Big)$$

$$= \mathbb{E}\left[\mathbb{P}\Big(|(v_k - \tilde{v}_k)_l| \geq \frac{1}{\pi\sqrt{K}}\phi_k\|\tilde{v}_k\|\Big)|\hat{\Pi}_{\mathcal{U}}\right]$$

$$= \mathbb{E}\left[\mathbb{P}\Big(\frac{1}{n_{kl}}|\sum_{i\in\mathcal{C}_k\cap\mathcal{L}}\sum_{j\in\mathcal{C}_l\cap\mathcal{L}}(A_{ij} - \Omega_{ij})| \geq \frac{1}{\pi\sqrt{K}n_{kl}}\phi_k\|\tilde{v}_k\|\Big)|\hat{\Pi}_{\mathcal{U}}\right]$$

$$\leq 2\mathbb{E}\exp\left(-\frac{\frac{1}{2}n_{kl}\Big(\frac{1}{\pi\sqrt{K}n_{kl}}\phi_k\|\tilde{v}_k\|\Big)^2}{\frac{1}{n_{kl}}\sum_{i\in\mathcal{C}_k\cap\mathcal{L}}\sum_{j\in\mathcal{C}_l\cap\mathcal{L}}\Omega_{ij} + \frac{1}{3}\frac{1}{\pi\sqrt{K}n_{kl}}\phi_k\|\tilde{v}_k\|}\right)$$

$$= 2\mathbb{E}\exp\left(-\frac{\phi_k^2}{2\pi\sqrt{K}}\frac{\|\tilde{v}_k\|^2}{\pi\sqrt{K}\sum_{i\in\mathcal{C}_k\cap\mathcal{L}}\sum_{j\in\mathcal{C}_l\cap\mathcal{L}}\Omega_{ij} + \frac{1}{3}\phi_k\|\tilde{v}_k\|}\right)$$

$$= 2\mathbb{E}\exp\left(-\frac{\phi_k^2}{2\pi\sqrt{K}}\frac{\|\tilde{v}_k\|^2}{\pi\sqrt{K}|(\tilde{v}_k)_l| + \frac{1}{3}\phi_k\|\tilde{v}_k\|}\right)$$

$$= 2\mathbb{E}\exp\left(-\frac{\phi_k^2\|\tilde{v}_k\|}{2\pi\sqrt{K}}\frac{1}{\pi\sqrt{K}\frac{|(\tilde{v}_k)_l|}{\|\tilde{v}_k\|} + \frac{1}{3}\phi_k}\right)$$

$$\leq 2\mathbb{E}\exp\left(-\frac{\phi_k^2\|\tilde{v}_k\|}{2\pi\sqrt{K}(\pi\sqrt{K} + \frac{\pi}{3})}\right) \tag{54}$$

When $l = k$, $\{A_{ij} - \Omega_{ij} : i, j \in \mathcal{C}_k \cap \mathcal{L}, i < j\}$ are a collection of $\frac{1}{2}|\mathcal{C}_k \cap \mathcal{L}|(|\mathcal{C}_k \cap \mathcal{L}| - 1)$ independent random variables. Furthermore, for any $i, j \in \mathcal{C}_k \cap \mathcal{L}, i < j$,

$$\mathbb{E}(A_{ij} - \Omega_{ij}) = \mathbb{E}A_{ij} - \Omega_{ij} = \Omega_{ij} - \Omega_{ij} = 0$$

Also,

$$-1 \leq -\Omega_{ij} \leq A_{ij} - \Omega_{ij} \leq A_{ij} \leq 1$$

So $|A_{ij} - \Omega_{ij}| \leq 1$.

Additionally,

$$var(A_{ij} - \Omega_{ij}) = var(A_{ij}) = \Omega_{ij}(1 - \Omega_{ij}) \leq \Omega_{ij}$$

Denote $n_{kk} = \frac{1}{2}|\mathcal{C}_k \cap \mathcal{L}|(|\mathcal{C}_k \cap \mathcal{L}| - 1)$, we have

$$\mathbb{P}\Big(|(v_k - \tilde{v}_k)_l| \geq \frac{1}{\pi\sqrt{K}}\phi_k\|\tilde{v}_k\|\Big)$$

$$= \mathbb{P}\Big(\frac{1}{n_{kk}}|\sum_{i\in\mathcal{C}_k\cap\mathcal{L}}\sum_{j\in\mathcal{C}_k\cap\mathcal{L}}(A_{ij} - \Omega_{ij})| \geq \frac{1}{\pi\sqrt{K}n_{kk}}\phi_k\|\tilde{v}_k\|\Big)$$

$$= \mathbb{P}\Big(\frac{1}{n_{kk}}|2\sum_{i<j\in\mathcal{C}_k\cap\mathcal{L}}(A_{ij}-\Omega_{ij})+\sum_{i\in\mathcal{C}_k\cap\mathcal{L}}\Omega_{ii}|\geq\frac{1}{\pi\sqrt{K}n_{kk}}\phi_k\|\tilde{v}_k\|\Big)$$

$$\leq \mathbb{P}\Big(\frac{1}{n_{kk}}|2\sum_{i<j\in\mathcal{C}_k\cap\mathcal{L}}(A_{ij}-\Omega_{ij})|\geq\frac{1}{\pi\sqrt{K}n_{kk}}\phi_k\|\tilde{v}_k\|-\frac{1}{n_{kk}}\sum_{i\in\mathcal{C}_k\cap\mathcal{L}}\Omega_{ii}\Big)$$

$$= \mathbb{P}\Big(\frac{1}{n_{kk}}|\sum_{i<j\in\mathcal{C}_k\cap\mathcal{L}}(A_{ij}-\Omega_{ij})|\geq\frac{1}{2\pi\sqrt{K}n_{kk}}\phi_k\|\tilde{v}_k\|-\frac{1}{2n_{kk}}\sum_{i\in\mathcal{C}_k\cap\mathcal{L}}\Omega_{ii}\Big)$$

$$= \mathbb{E}\left[\mathbb{P}\Big(\frac{1}{n_{kk}}|\sum_{i<j\in\mathcal{C}_k\cap\mathcal{L}}(A_{ij}-\Omega_{ij})|\geq\frac{1}{2\pi\sqrt{K}n_{kk}}\phi_k\|\tilde{v}_k\|-\frac{1}{2n_{kk}}\sum_{i\in\mathcal{C}_k\cap\mathcal{L}}\Omega_{ii}\Big)|\hat{\Pi}_{\mathcal{U}}\right] \quad (55)$$

Notice that

$$\phi_k \geq C\frac{\|\theta_{\mathcal{L}}^{(k)}\|^2}{\|\theta_{\mathcal{L}}^{(k)}\|_1\|\theta\|_1}.$$

Since $\phi_k \geq 2\sqrt{2}\pi C_2 K^2\frac{\|\theta_{\mathcal{L}}^{(k)}\|^2}{\|\theta_{\mathcal{L}}^{(k)}\|_1\|\theta\|_1}\|\tilde{v}_k\|$,

$$\frac{1}{2\pi\sqrt{K}n_{kk}}\phi_k\|\tilde{v}_k\| \geq \frac{2\sqrt{2}\pi K^2}{2\pi\sqrt{K}n_{kk}}\frac{\|\theta_{\mathcal{L}}^{(k)}\|^2}{\|\theta_{\mathcal{L}}^{(k)}\|_1\|\theta\|_1}\|\tilde{v}_k\|$$

$$((\text{By }(53))) \geq \frac{C_2 K\sqrt{2K}}{n_{kk}}\frac{\|\theta_{\mathcal{L}}^{(k)}\|^2}{\|\theta_{\mathcal{L}}^{(k)}\|_1\|\theta\|_1}\frac{1}{C_2 K\sqrt{2K}}\|\theta_{\mathcal{L}}^{(k)}\|_1\|\theta\|_1$$

$$= 2\|\frac{1}{2n_{kk}}\theta_{\mathcal{L}}^{(k)}\|^2$$

$$= 2\frac{1}{2n_{kk}}\sum_{i\in\mathcal{C}_k\cap\mathcal{L}}\theta_i^2$$

$$(\text{Identifiability condition}) = \frac{C}{\sqrt{2}\pi K^2}\frac{1}{2n_{kk}}\sum_{i\in\mathcal{C}_k\cap\mathcal{L}}\theta_i^2 P_{kk}$$

$$= 2\frac{1}{2n_{kk}}\sum_{i\in\mathcal{C}_k\cap\mathcal{L}}\Omega_{ii} \quad (56)$$

Therefore, by Lemma 7,

$$\mathbb{P}\Big(|(v_k-\tilde{v}_k)_l|\geq\frac{1}{\pi\sqrt{K}}\phi_k\|\tilde{v}_k\|\Big)$$

$$= \mathbb{E}\left[\mathbb{P}\Big(\frac{1}{n_{kk}}|\sum_{i<j\in\mathcal{C}_k\cap\mathcal{L}}(A_{ij}-\Omega_{ij})|\geq\frac{1}{2\pi\sqrt{K}n_{kk}}\phi_k\|\tilde{v}_k\|-\frac{1}{2}\frac{1}{2\pi\sqrt{K}n_{kk}}\phi_k\|\tilde{v}_k\|\Big)|\hat{\Pi}_{\mathcal{U}}\right]$$

$$\leq 2\mathbb{E}\exp\left(-\frac{\frac{1}{2}n_{kk}\Big(\frac{1}{4\pi\sqrt{K}n_{kk}}\phi_k\|\tilde{v}_k\|\Big)^2}{\frac{1}{n_{kk}}\sum_{i<j\in\mathcal{C}_k\cap\mathcal{L}}\Omega_{ij}+\frac{1}{3}\Big(\frac{1}{2\pi\sqrt{K}n_{kk}}\phi_k\|\tilde{v}_k\|-\frac{1}{2n_{kk}}\sum_{i\in\mathcal{C}_k\cap\mathcal{L}}\Omega_{ii}\Big)}\right)$$

$$\leq 2\mathbb{E}\exp\left(-\frac{1}{16\pi\sqrt{K}}\frac{\Big(\phi_k\|\tilde{v}_k\|\Big)^2}{\pi\sqrt{K}\cdot2\sum_{i<j\in\mathcal{C}_k\cap\mathcal{L}}\Omega_{ij}+\frac{1}{3}\phi_k\|\tilde{v}_k\|}\right)$$

$$\leq 2\mathbb{E}\exp\left(-\frac{1}{16\pi\sqrt{K}}\frac{\Big(\phi_k\|\tilde{v}_k\|\Big)^2}{\pi\sqrt{K}\sum_{i,j\in\mathcal{C}_k\cap\mathcal{L}}\Omega_{ij}+\frac{1}{3}\phi_k\|\tilde{v}_k\|}\right)$$

$$= 2\mathbb{E}\exp\left(-\frac{\phi_k^2}{16\pi\sqrt{K}}\frac{\|\tilde{v}_k\|^2}{\pi\sqrt{K}|(\tilde{v}_k)_k|+\frac{1}{3}\phi_k\|\tilde{v}_k\|}\right)$$

$$= 2\mathbb{E}\exp\left(-\frac{\phi_k^2\|\tilde{v}_k\|}{16\pi\sqrt{K}}\frac{1}{\pi\sqrt{K}\frac{|(\tilde{v}_k)_k|}{\|\tilde{v}_k\|} + \frac{1}{3}\phi_k}\right)$$

$$\le 2\mathbb{E}\exp\left(-\frac{\phi_k^2\|\tilde{v}_k\|}{16\pi\sqrt{K}(\pi\sqrt{K}+\frac{\pi}{3})}\right) \tag{57}$$

When $l \in \{K+1,...,2K\}$, recall

$$\hat{\mathcal{C}}_l = \{i \in \mathcal{U} : \hat{\pi}_i = e_{l-K}\}$$

So

$$(v_k)_l = (f(A^{(k)}; \hat{\Pi}_\mathcal{U}))_l = A^{(k)}\left(\hat{\Pi}_\mathcal{U}\right)_l = \sum_{i\in\mathcal{C}_k\cap\mathcal{L}}\sum_{j\in\hat{\mathcal{C}}_l} A_{ij}$$

Recall that

$$(\tilde{v}_k)_l = (f(\Omega^{(k)}; \hat{\Pi}_\mathcal{U}))_l = \Omega^{(k)}\left(\hat{\Pi}_\mathcal{U}\right)_l = \sum_{i\in\mathcal{C}_k\cap\mathcal{L}}\sum_{j\in\hat{\mathcal{C}}_l} \Omega_{ij}$$

So

$$|(v_k - \tilde{v}_k)_l| = \sum_{i\in\mathcal{C}_k\cap\mathcal{L}}\sum_{j\in\hat{\mathcal{C}}_l}(A_{ij} - \Omega_{ij})$$

Since $\hat{\Pi}_\mathcal{U}$ only depends on $A_{\mathcal{U}\mathcal{U}}$, it is independent of $A_{\mathcal{L}\mathcal{U}}$. Hence, given $\hat{\Pi}_\mathcal{U}$, $\{A_{ij} - \Omega_{ij} : i \in \mathcal{C}_k \cap \mathcal{L}, j \in \hat{\mathcal{C}}_l\}$ are a collection of $|\mathcal{C}_k \cap \mathcal{L}||\hat{\mathcal{C}}_l|$ independent random variables. Furthermore, given $\hat{\Pi}_\mathcal{U}$, for any $i \in \mathcal{C}_k \cap \mathcal{L}, j \in \hat{\mathcal{C}}_l$,

$$\mathbb{E}\left[A_{ij} - \Omega_{ij}|\hat{\Pi}_\mathcal{U}\right] = \mathbb{E}\left[A_{ij}|\hat{\Pi}_\mathcal{U}\right] - \Omega_{ij} = \Omega_{ij} - \Omega_{ij} = 0$$

Also,

$$-1 \le -\Omega_{ij} \le A_{ij} - \Omega_{ij} \le A_{ij} \le 1$$

So $|A_{ij} - \Omega_{ij}| \le 1$. Additionally,

$$var\left(A_{ij} - \Omega_{ij}\right) = var\left(A_{ij}|\hat{\Pi}_\mathcal{U}\right) = \Omega_{ij}(1 - \Omega_{ij}) \le \Omega_{ij}$$

Therefore, denote $\hat{n}_{kl} = |\mathcal{C}_k \cap \mathcal{L}||\hat{\mathcal{C}}_l|$, by Lemma 7,

$$\mathbb{P}\left(|(v_k - \tilde{v}_k)_l| \ge \frac{1}{\pi\sqrt{K}}\phi_k\|\tilde{v}_k\|\right)$$

$$= \mathbb{E}\left[\mathbb{P}\left(|(v_k - \tilde{v}_k)_l| \ge \frac{1}{\pi\sqrt{K}}\phi_k\|\tilde{v}_k\|\right)|\hat{\Pi}_\mathcal{U}\right]$$

$$= \mathbb{E}\left[\mathbb{P}\left(\frac{1}{\hat{n}_{kl}}|\sum_{i\in\mathcal{C}_k\cap\mathcal{L}}\sum_{j\in\hat{\mathcal{C}}_l}(A_{ij} - \Omega_{ij})| \ge \frac{1}{\pi\sqrt{K}\hat{n}_{kl}}\phi_k\|\tilde{v}_k\|\right)|\hat{\Pi}_\mathcal{U}\right]$$

$$\le 2\mathbb{E}\exp\left(-\frac{\frac{1}{2}\hat{n}_{kl}\left(\frac{1}{\pi\sqrt{K}\hat{n}_{kl}}\phi_k\|\tilde{v}_k\|\right)^2}{\frac{1}{\hat{n}_{kl}}\sum_{i\in\mathcal{C}_k\cap\mathcal{L}}\sum_{j\in\hat{\mathcal{C}}_l}\Omega_{ij} + \frac{1}{3}\frac{1}{\pi\sqrt{K}\hat{n}_{kl}}\phi_k\|\tilde{v}_k\|}\right)$$

$$= 2\mathbb{E}\exp\left(-\frac{\phi_k^2}{2\pi\sqrt{K}}\frac{\|\tilde{v}_k\|^2}{\pi\sqrt{K}\sum_{i\in\mathcal{C}_k\cap\mathcal{L}}\sum_{j\in\hat{\mathcal{C}}_l}\Omega_{ij} + \frac{1}{3}\phi_k\|\tilde{v}_k\|}\right)$$

$$= 2\mathbb{E}\exp\left(-\frac{\phi_k^2}{2\pi\sqrt{K}}\frac{\|\tilde{v}_k\|^2}{\pi\sqrt{K}|(\tilde{v}_k)_l| + \frac{1}{3}\phi_k\|\tilde{v}_k\|}\right)$$

$$= 2\mathbb{E}\exp\left(-\frac{\phi_k^2\|\tilde{v}_k\|}{2\pi\sqrt{K}}\frac{1}{\pi\sqrt{K}\frac{|(\tilde{v}_k)_l|}{\|\tilde{v}_k\|} + \frac{1}{3}\phi_k}\right)$$

$$\leq 2\mathbb{E}\exp\left(-\frac{\phi_k^2\|\tilde{v}_k\|}{2\pi\sqrt{K}(\pi\sqrt{K}+\frac{\pi}{3})}\right) \tag{58}$$

In all, for any $l \in [2K]$,

$$\mathbb{P}\left(|(v_k-\tilde{v}_k)_l| \geq \frac{1}{\pi\sqrt{K}}\phi_k\|\tilde{v}_k\|\right) \leq 2\mathbb{E}\exp\left(-\frac{\phi_k^2\|\tilde{v}_k\|}{16\pi\sqrt{K}(\pi\sqrt{K}+\frac{\pi}{3})}\right) \tag{59}$$

Plugging (53) into (59), we obtain

$$\mathbb{P}\left(|(v_k-\tilde{v}_k)_l| \geq \frac{1}{\pi\sqrt{K}}\phi_k\|\tilde{v}_k\|\right) \leq 2\exp\left(-\frac{\phi_k^2\|\theta_{\mathcal{L}}^{(k)}\|_1\|\theta\|_1}{16\sqrt{2}\pi^2C_2K^2(\sqrt{K}+\frac{1}{3})}\right)$$

$$= 2\exp\left(-\frac{C_6}{C^2}\phi_k^2\|\theta_{\mathcal{L}}^{(k)}\|_1\|\theta\|_1\right) \tag{60}$$

That concludes the proof.

$\square$

## H.2 Proof of Lemma 9

The proof of Lemma 9 is nearly the same as Lemma 8. For the completeness of our paper, we will present a proof of Lemma 9 as follows.

*Proof.* When $l \in [K]$,
$$(v^*)_l = (f(X;\hat{\Pi}_{\mathcal{U}}))_l = X\mathbf{1}_{(l)} = \sum_{j\in\mathcal{C}_l\cap\mathcal{L}}X_j$$

Similarly,
$$(\tilde{v}^*)_l = (f(\mathbb{E}[X];\hat{\Pi}_{\mathcal{U}}))_l = \mathbb{E}[X]\mathbf{1}_{(l)} = \sum_{j\in\mathcal{C}_l\cap\mathcal{L}}\mathbb{E}[X_j]$$

So
$$|(v^*-\tilde{v}^*)_l| = \sum_{j\in\mathcal{C}_l\cap\mathcal{L}}(X_j-\mathbb{E}[X_j])$$

When $l \in [K]$, since $\hat{\Pi}_{\mathcal{U}}$ only depends on $A_{\mathcal{U}\mathcal{U}}$, it is independent of $X$. Hence, given $\hat{\Pi}_{\mathcal{U}}$, $\{X_j - \mathbb{E}[X_j] : j \in \mathcal{C}_l \cap \mathcal{L}\}$ are a collection of $|\mathcal{C}_l \cap \mathcal{L}|$ independent random variables. Furthermore, given $\hat{\Pi}_{\mathcal{U}}$, for any $j \in \mathcal{C}_l \cap \mathcal{L}$,

$$\mathbb{E}\left[X_j-\mathbb{E}[X_j]|\hat{\Pi}_{\mathcal{U}}\right] = \mathbb{E}\left[X_j|\hat{\Pi}_{\mathcal{U}}\right] - \mathbb{E}[X_j] = \mathbb{E}[X_j] - \mathbb{E}[X_j] = 0$$

Also,
$$-1 \leq -\mathbb{E}[X_j] \leq X_j - \mathbb{E}[X_j] \leq X_j \leq 1$$

So $|X_j - \mathbb{E}[X_j]| \leq 1$. Additionally,

$$var\left(X_j-\mathbb{E}[X_j]\right) = var\left(X_j|\hat{\Pi}_{\mathcal{U}}\right) = \mathbb{E}[X_j](1-\mathbb{E}[X_j]) \leq \mathbb{E}[X_j]$$

Therefore, denote $n_l = |\mathcal{C}_l \cap \mathcal{L}|$, by Lemma 7,

$$\mathbb{P}\left(|(v^*-\tilde{v}^*)_l| \geq \frac{1}{\pi\sqrt{K}}\phi_k\|\tilde{v}^*\|\right)$$

$$= \mathbb{E}\left[\mathbb{P}\left(|(v^*-\tilde{v}^*)_l| \geq \frac{1}{\pi\sqrt{K}}\phi_k\|\tilde{v}^*\|\right)|\hat{\Pi}_{\mathcal{U}}\right]$$

$$= \mathbb{E}\left[\mathbb{P}\Big(\frac{1}{n_l}|\sum_{j \in \mathcal{C}_l \cap \mathcal{L}}(X_j - \mathbb{E}[X_j])| \geq \frac{1}{\pi\sqrt{K}n_l}\phi_k\|\tilde{v}^*\|\Big)|\hat{\Pi}_{\mathcal{U}}\right]$$

$$\leq 2\mathbb{E}\exp\left(-\frac{\frac{1}{2}n_l\left(\frac{1}{\pi\sqrt{K}n_l}\phi_k\|\tilde{v}^*\|\right)^2}{\frac{1}{n_l}\sum_{j \in \mathcal{C}_l \cap \mathcal{L}}\mathbb{E}[X_j] + \frac{1}{3}\frac{1}{\pi\sqrt{K}n_l}\phi_k\|\tilde{v}^*\|}\right)$$

$$= 2\mathbb{E}\exp\left(-\frac{\phi_k^2}{2\pi\sqrt{K}}\frac{\|\tilde{v}^*\|^2}{\pi\sqrt{K}\sum_{j \in \mathcal{C}_l \cap \mathcal{L}}\mathbb{E}[X_j] + \frac{1}{3}\phi_k\|\tilde{v}^*\|}\right)$$

$$= 2\mathbb{E}\exp\left(-\frac{\phi_k^2}{2\pi\sqrt{K}}\frac{\|\tilde{v}^*\|^2}{\pi\sqrt{K}|(\tilde{v}^*)_l| + \frac{1}{3}\phi_k\|\tilde{v}^*\|}\right)$$

$$= 2\mathbb{E}\exp\left(-\frac{\phi_k^2\|\tilde{v}^*\|}{2\pi\sqrt{K}}\frac{1}{\pi\sqrt{K}\frac{|(\tilde{v}^*)_l|}{\|\tilde{v}^*\|} + \frac{1}{3}\phi_k}\right)$$

$$\leq 2\mathbb{E}\exp\left(-\frac{\phi_k^2\|\tilde{v}^*\|}{2\pi\sqrt{K}(\pi\sqrt{K} + \frac{\pi}{3})}\right) \tag{61}$$

When $l \in \{K+1, ..., 2K\}$, define

$$\hat{\mathcal{C}}_l = \{i \in \mathcal{U} : \hat{\pi}_i = e_{l-K}\}$$

Then

$$(v^*)_l = (f(X; \hat{\Pi}_{\mathcal{U}}))_l = X\mathbf{1}_{(l)} = \sum_{j \in \hat{\mathcal{C}}_l} X_j$$

Similarly,

$$(\tilde{v}^*)_l = (f(\mathbb{E}[X]; \hat{\Pi}_{\mathcal{U}}))_l = \mathbb{E}[X]\mathbf{1}_{(l)} = \sum_{j \in \hat{\mathcal{C}}_l} \mathbb{E}[X_j]$$

So

$$|(v^* - \tilde{v}^*)_l| = \sum_{j \in \hat{\mathcal{C}}_l}(X_j - \mathbb{E}[X_j])$$

When $l \in [K]$, since $\hat{\Pi}_{\mathcal{U}}$ only depends on $A_{\mathcal{U}\mathcal{U}}$, it is independent of $X$. Hence, given $\hat{\Pi}_{\mathcal{U}}$, $\{X_j - \mathbb{E}[X_j] : j \in \hat{\mathcal{C}}_l\}$ are a collection of $|\mathcal{C}_l \cap \mathcal{L}|$ independent random variables. Furthermore, given $\hat{\Pi}_{\mathcal{U}}$, for any $j \in \hat{\mathcal{C}}_l$,

$$\mathbb{E}\Big[X_j - \mathbb{E}[X_j]|\hat{\Pi}_{\mathcal{U}}\Big] = \mathbb{E}\Big[X_j|\hat{\Pi}_{\mathcal{U}}\Big] - \mathbb{E}[X_j] = \mathbb{E}[X_j] - \mathbb{E}[X_j] = 0$$

Also,

$$-1 \leq -\mathbb{E}[X_j] \leq X_j - \mathbb{E}[X_j] \leq X_j \leq 1$$

So $|X_j - \mathbb{E}[X_j]| \leq 1$. Additionally,

$$var\Big(X_j - \mathbb{E}[X_j]\Big) = var\Big(X_j|\hat{\Pi}_{\mathcal{U}}\Big) = \mathbb{E}[X_j](1 - \mathbb{E}[X_j]) \leq \mathbb{E}[X_j]$$

Therefore, denote $\hat{n}_l = |\hat{\mathcal{C}}_l|$, by Lemma 7,

$$\mathbb{P}\Big(|(v^* - \tilde{v}^*)_l| \geq \frac{1}{\pi\sqrt{K}}\phi_k\|\tilde{v}^*\|\Big)$$

$$= \mathbb{E}\left[\mathbb{P}\Big(|(v^* - \tilde{v}^*)_l| \geq \frac{1}{\pi\sqrt{K}}\phi_k\|\tilde{v}^*\|\Big)|\hat{\Pi}_{\mathcal{U}}\right]$$

$$= \mathbb{E}\left[\mathbb{P}\Big(\frac{1}{\hat{n}_l}|\sum_{j \in \hat{\mathcal{C}}_l}(X_j - \mathbb{E}[X_j])| \geq \frac{1}{\pi\sqrt{K}\hat{n}_l}\phi_k\|\tilde{v}^*\|\Big)|\hat{\Pi}_{\mathcal{U}}\right]$$

$$\leq 2\mathbb{E}\exp\left(-\frac{\frac{1}{2}\hat{n}_l\left(\frac{1}{\pi\sqrt{K}\hat{n}_l}\phi_k\|\tilde{v}^*\|\right)^2}{\frac{1}{\hat{n}_l}\sum_{j\in\hat{\mathcal{C}}_l}\mathbb{E}[X_j] + \frac{1}{3}\frac{1}{\pi\sqrt{K}\hat{n}_l}\phi_k\|\tilde{v}^*\|}\right)$$

$$= 2\mathbb{E}\exp\left(-\frac{\phi_k^2}{2\pi\sqrt{K}}\frac{\|\tilde{v}^*\|^2}{\pi\sqrt{K}\sum_{j\in\hat{\mathcal{C}}_l}\mathbb{E}[X_j] + \frac{1}{3}\phi_k\|\tilde{v}^*\|}\right)$$

$$= 2\mathbb{E}\exp\left(-\frac{\phi_k^2}{2\pi\sqrt{K}}\frac{\|\tilde{v}^*\|^2}{\pi\sqrt{K}|(\tilde{v}^*)_l| + \frac{1}{3}\phi_k\|\tilde{v}^*\|}\right)$$

$$= 2\mathbb{E}\exp\left(-\frac{\phi_k^2\|\tilde{v}^*\|}{2\pi\sqrt{K}}\frac{1}{\pi\sqrt{K}\frac{|(\tilde{v}^*)_l|}{\|\tilde{v}^*\|} + \frac{1}{3}\phi_k}\right)$$

$$\leq 2\mathbb{E}\exp\left(-\frac{\phi_k^2\|\tilde{v}^*\|}{2\pi\sqrt{K}(\pi\sqrt{K} + \frac{\pi}{3})}\right) \tag{62}$$

In all, for any $l\in[2K]$,

$$\mathbb{P}\left(|(v^* - \tilde{v}^*)_l| \geq \frac{1}{\pi\sqrt{K}}\phi_k\|\tilde{v}^*\|\right) \leq 2\mathbb{E}\exp\left(-\frac{\phi_k^2\|\tilde{v}^*\|}{2\pi\sqrt{K}(\pi\sqrt{K} + \frac{\pi}{3})}\right) \tag{63}$$

Notice that

$$\|\tilde{v}^*\| = \sqrt{\sum_{l\in[2K]}(\tilde{v}^*)_l^2}$$

$$\text{(Cauchy-Schwartz)} \geq \sqrt{\frac{1}{2K}(\sum_{l\in[2K]}(\tilde{v}^*)_l)^2}$$

$$= \frac{1}{\sqrt{2K}}|\sum_{l\in[2K]}(\tilde{v}^*)_l|$$

$$= \frac{1}{\sqrt{2K}}|\sum_{l=1}^{K}\sum_{j\in\hat{\mathcal{C}}_l}\mathbb{E}[X_j] + \sum_{l=K+1}^{2K}\sum_{j\in\hat{\mathcal{C}}_l}\mathbb{E}[X_j]|$$

$$= \frac{1}{\sqrt{2K}}|\sum_{j\in\hat{\mathcal{L}}}\mathbb{E}[X_j] + \sum_{j\in\hat{\mathcal{U}}}\mathbb{E}[X_j]|$$

$$= \frac{1}{\sqrt{2K}}\sum_{j\in[n]}\mathbb{E}[X_j]$$

$$\geq \frac{1}{\sqrt{2K}}\sum_{j\in\mathcal{C}_{k^*}}\mathbb{E}[X_j]$$

$$= \frac{1}{\sqrt{2K}}\sum_{j\in\mathcal{C}_k}\theta^*\theta_j P_{k^*k^*}$$

$$\text{(Identifiability condition)} = \frac{1}{\sqrt{2K}}\sum_{j\in\mathcal{C}_k}\theta^*\theta_j$$

$$= \frac{1}{\sqrt{2K}}\theta^*\|\theta^{(k)}\|_1$$

$$\geq \frac{1}{\sqrt{2K}}\theta^*\min_{l\in[K]}\|\theta^{(l)}\|_1$$

$$\text{(Condition (9))} \geq \frac{1}{C_2\sqrt{2K}}\theta^*\max_{l\in[K]}\|\theta^{(l)}\|_1$$

$$\geq \frac{1}{C_2 K\sqrt{2K}}\theta^*\|\theta\|_1 \tag{64}$$

Plugging (64) into (63), we obtain

$$\mathbb{P}\Big(|(v^* - \tilde{v}^*)_l| \geq \frac{1}{\pi\sqrt{K}}\phi_k\|\tilde{v}^*\|\Big) \leq 2\exp\left(-\frac{\phi_k^2\theta^*\|\theta\|_1}{2\sqrt{2}\pi^2 C_2 K^2(\sqrt{K} + \frac{1}{3})}\right)$$

$$= 2\exp\left(-\frac{C_7}{C^2}\phi_k^2\theta^*\|\theta\|_1\right) \tag{65}$$

That concludes the proof.

$\square$

# I PROOF OF COROLLARY 1, 2

## I.1 PROOF OF COROLLARY 1

**Corollary 1.** *Consider the DCBM model where (8)-(9) hold. Suppose for some constants $b_0 \in (0, 1)$ and $\epsilon \in (0, 1/2)$, $\hat{\Pi}_{\mathcal{U}}$ is $b_0$-correct with probability $1 - \epsilon$. When $b_0$ is properly small, there exist constants $C_0 > 0$ and $\bar{C} > 0$, which do not depend on $(b_0, \epsilon)$, such that $\mathbb{P}(\hat{y} \neq k^*) \leq \epsilon + \bar{C}\sum_{k=1}^K \exp\Big(-C_0\beta_n^2\|\theta\|_1 \cdot \min\{\theta^*, \|\theta_{\mathcal{L}}^{(k)}\|_1\}\Big)$.*

*Proof.* Let $B_0$ be the event that $\hat{\Pi}_{\mathcal{U}}$ is $b_0$-correct. Then,

$$\mathbb{P}(\hat{y} \neq k^*)$$
$$= \mathbb{P}(\hat{y} \neq k^*, B_0^C) + \mathbb{P}\Big(\hat{y} \neq k^*, B_0\Big)$$
$$\leq \mathbb{P}(B_0^C) + \mathbb{P}\Big(\Big\{\exists k \neq k^*, \hat{\psi}_k(\hat{\Pi}_{\mathcal{U}}) \leq \hat{\psi}_{k^*}(\hat{\Pi}_{\mathcal{U}})\Big\}, B_0\Big)$$
$$\leq \epsilon + \mathbb{P}\Big(\Big\{\exists k \neq k^*, \Big(\psi_k(\hat{\Pi}_{\mathcal{U}}) - \hat{\psi}_k(\hat{\Pi}_{\mathcal{U}})\Big)$$
$$\qquad\qquad + \Big(\hat{\psi}_{k^*}(\hat{\Pi}_{\mathcal{U}}) - \psi_{k^*}(\hat{\Pi}_{\mathcal{U}})\Big) \geq \Big(\psi_k(\hat{\Pi}_{\mathcal{U}}) - \psi_{k^*}(\hat{\Pi}_{\mathcal{U}})\Big)\Big\}, B_0\Big)$$
$$\leq \epsilon + \mathbb{P}\Big(\Big\{\exists k \neq k^*, \Big|\psi_k(\hat{\Pi}_{\mathcal{U}}) - \hat{\psi}_k(\hat{\Pi}_{\mathcal{U}})\Big|$$
$$\qquad\qquad + \Big|\hat{\psi}_{k^*}(\hat{\Pi}_{\mathcal{U}}) - \psi_{k^*}(\hat{\Pi}_{\mathcal{U}})\Big| \geq \Big|\psi_k(\hat{\Pi}_{\mathcal{U}}) - \psi_{k^*}(\hat{\Pi}_{\mathcal{U}})\Big|\Big\}, B_0\Big) \tag{66}$$

By Theorem 1, when $b_0$ is properly small, $B_0$ implies that there exists a constant $c_0 > 0$, which does not depend on $b_0$, such that $\psi_{k^*}(\hat{\Pi}_{\mathcal{U}}) = 0$ and $\min_{k \neq k^*}\{\psi_k(\hat{\Pi}_{\mathcal{U}})\} \geq c_0\beta_n$.

Substituting this result into (66), we have

$$\mathbb{P}(\hat{y} \neq k^*)$$
$$\leq \epsilon + \mathbb{P}\Big(\Big\{\exists k \neq k^*, \Big|\psi_k(\hat{\Pi}_{\mathcal{U}}) - \hat{\psi}_k(\hat{\Pi}_{\mathcal{U}})\Big| + \Big|\hat{\psi}_{k^*}(\hat{\Pi}_{\mathcal{U}}) - \psi_{k^*}(\hat{\Pi}_{\mathcal{U}})\Big| \geq c_0\beta_n\Big\}, B_0\Big)$$
$$\leq \epsilon + \mathbb{P}\Big(\exists k \neq k^*, \Big|\psi_k(\hat{\Pi}_{\mathcal{U}}) - \hat{\psi}_k(\hat{\Pi}_{\mathcal{U}})\Big| + \Big|\hat{\psi}_{k^*}(\hat{\Pi}_{\mathcal{U}}) - \psi_{k^*}(\hat{\Pi}_{\mathcal{U}})\Big| \geq c_0\beta_n\Big)$$
$$\leq \epsilon + \mathbb{P}\Big(\exists k \in [K], \Big|\psi_k(\hat{\Pi}_{\mathcal{U}}) - \hat{\psi}_k(\hat{\Pi}_{\mathcal{U}})\Big| \geq \frac{1}{2}c_0\beta_n\Big)$$
$$\leq \epsilon + \mathbb{P}\Big(\exists k \in [K], \Big|\psi_k(\hat{\Pi}_{\mathcal{U}}) - \hat{\psi}_k(\hat{\Pi}_{\mathcal{U}})\Big| > \frac{1}{3}c_0\beta_n\Big) \tag{67}$$

According to Theorem 2, there exists a constant $C > 0$, such that for any $\delta \in (0, 1/2)$, with probability $1 - \delta$, simultaneously for $1 \leq k \leq K$,

$$|\hat{\psi}_k(\hat{\Pi}_{\mathcal{U}}) - \psi_k(\hat{\Pi}_{\mathcal{U}})| \leq C\left(\sqrt{\frac{\log(1/\delta)}{\|\theta\|_1 \cdot \min\{\theta^*, \|\theta_{\mathcal{L}}^{(k)}\|_1\}}} + \frac{\|\theta_{\mathcal{L}}^{(k)}\|^2}{\|\theta_{\mathcal{L}}^{(k)}\|_1\|\theta\|_1}\right).$$

Take $C_0 = \frac{c_0^2}{36C^2}$,

$$\delta = \exp\left(-C_0\beta_n^2\|\theta\|_1 \cdot \min\left\{\theta^*, \min_{k\in[K]} \|\theta_{\mathcal{L}}^{(k)}\|_1\right\}\right)$$

Then

$$C\sqrt{\frac{\log(1/\delta)}{\|\theta\|_1 \cdot \min\{\theta^*, \|\theta_{\mathcal{L}}^{(k)}\|_1\}}} = C\sqrt{\frac{\log\left(1/\exp\left(-C_0\beta_n^2\|\theta\|_1 \cdot \min\left\{\theta^*, \min_{k\in[K]} \|\theta_{\mathcal{L}}^{(k)}\|_1\right\}\right)\right)}{\|\theta\|_1 \cdot \min\{\theta^*, \|\theta_{\mathcal{L}}^{(k)}\|_1\}}}$$

$$= C\sqrt{\frac{C_0\beta_n^2\|\theta\|_1 \cdot \min\left\{\theta^*, \min_{k\in[K]} \|\theta_{\mathcal{L}}^{(k)}\|_1\right\}}{\|\theta\|_1 \cdot \min\{\theta^*, \|\theta_{\mathcal{L}}^{(k)}\|_1\}}}$$

$$\leq \frac{1}{6}c_0\beta_n \tag{68}$$

On the other hand, take $c_3$ in condition (9) properly small such that $c_3 \leq \frac{c_0}{6C}$, then according to condition (9),

$$C\frac{\|\theta_{\mathcal{L}}^{(k)}\|^2}{\|\theta_{\mathcal{L}}^{(k)}\|_1\|\theta\|_1} \leq C \cdot c_3\beta_n \leq \frac{1}{6}c_0\beta_n \tag{69}$$

Combining (68) and (69), we have

$$C\left(\sqrt{\frac{\log(1/\delta)}{\|\theta\|_1 \cdot \min\{\theta^*, \|\theta_{\mathcal{L}}^{(k)}\|_1\}}} + \frac{\|\theta_{\mathcal{L}}^{(k)}\|^2}{\|\theta_{\mathcal{L}}^{(k)}\|_1\|\theta\|_1}\right) \leq \frac{1}{3}c_0\beta_n$$

Therefore, when $\delta < \frac{1}{2}$, by Theorem 2, with probability $1 - \delta$, simultaneously for $1 \leq k \leq K$,

$$|\hat{\psi}_k(\hat{\Pi}_{\mathcal{U}}) - \psi_k(\hat{\Pi}_{\mathcal{U}})| \leq \frac{1}{3}c_0\beta_n.$$

As a result, when $\delta < \frac{1}{2}$,

$$\mathbb{P}\left(\exists k \in [K], \left|\psi_k(\hat{\Pi}_{\mathcal{U}}) - \hat{\psi}_k(\hat{\Pi}_{\mathcal{U}})\right| > \frac{1}{3}c_0\beta_n\right) \leq \delta$$

When $\delta \geq \frac{1}{2}$,

$$\mathbb{P}\left(\exists k \in [K], \left|\psi_k(\hat{\Pi}_{\mathcal{U}}) - \hat{\psi}_k(\hat{\Pi}_{\mathcal{U}})\right| > \frac{1}{3}c_0\beta_n\right) \leq 1 \leq 2\delta$$

Hence in total, we have

$$\mathbb{P}\left(\exists k \in [K], \left|\psi_k(\hat{\Pi}_{\mathcal{U}}) - \hat{\psi}_k(\hat{\Pi}_{\mathcal{U}})\right| > \frac{1}{3}c_0\beta_n\right) \leq 2\delta \tag{70}$$

Plugging (70) into (67), we obtain

$$\mathbb{P}(\hat{y} \neq k^*) \leq \epsilon + 2\delta$$

$$= \epsilon + 2\exp\left(-C_0\beta_n^2\|\theta\|_1 \cdot \min\left\{\theta^*, \min_{k\in[K]} \|\theta_{\mathcal{L}}^{(k)}\|_1\right\}\right)$$

$$\leq \epsilon + 2\sum_{k=1}^{K} \exp\left(-C_0\beta_n^2\|\theta\|_1 \cdot \min\{\theta^*, \|\theta_{\mathcal{L}}^{(k)}\|_1\}\right) \tag{71}$$

Choose $\bar{C} = 2$, we obtain

$$\mathbb{P}(\hat{y} \neq k^*) \leq \epsilon + \bar{C}\sum_{k=1}^{K} \exp\left(-C_0\beta_n^2\|\theta\|_1 \cdot \min\{\theta^*, \|\theta_{\mathcal{L}}^{(k)}\|_1\}\right)$$

To conclude, when $b_0$ is properly small, there exist constants $C_0 = \frac{c_0^2}{36C^2} > 0$ and $\bar{C} = 2 > 0$, which do not depend on $(b_0, \epsilon)$, such that $\mathbb{P}(\hat{y} \neq k^*) \leq \epsilon + \bar{C} \sum_{k=1}^K \exp\left(-C_0 \beta_n^2 \|\theta\|_1 \cdot \min\{\theta^*, \|\theta_{\mathcal{L}}^{(k)}\|_1\}\right)$.

$\square$

## I.2 PROOF OF COROLLARY 2

**Corollary 2.** *Consider the DCBM model where (8)-(9) hold. We apply SCORE+ to obtain $\hat{\Pi}_{\mathcal{U}}$ and plug it into AngleMin+. As $n \to \infty$, suppose for some constant $q_0 > 0$, $\min_{i \in \mathcal{U}} \theta_i \geq q_0 \max_{i \in \mathcal{U}} \theta_i$, $\beta_n \|\theta_{\mathcal{U}}\| \geq q_0 \sqrt{\log(n)}$, $\beta_n^2 \|\theta\|_1 \theta^* \to \infty$, and $\beta_n^2 \|\theta\|_1 \min_k\{\|\theta_{\mathcal{L}}^{(k)}\|_1\} \to \infty$. Then, $\mathbb{P}(\hat{y} \neq k^*) \to 0$, so the AngleMin+ estimate is consistent.*

*Proof.* By Corollary 2, let $\epsilon$ be the probability that $\hat{\Pi}_{\mathcal{U}}$ obtained through SCORE+ is not $b_0$-correct, then

$$\mathbb{P}(\hat{y} \neq k^*) \leq \epsilon + \bar{C} \sum_{k=1}^K \exp\left(-C_0 \beta_n^2 \|\theta\|_1 \cdot \min\{\theta^*, \|\theta_{\mathcal{L}}^{(k)}\|_1\}\right)$$

Since $\beta_n^2 \|\theta\|_1 \theta^* \to \infty$, and $\beta_n^2 \|\theta\|_1 \min_k\{\|\theta_{\mathcal{L}}^{(k)}\|_1\} \to \infty$,

$$\bar{C} \sum_{k=1}^K \exp\left(-C_0 \beta_n^2 \|\theta\|_1 \cdot \min\{\theta^*, \|\theta_{\mathcal{L}}^{(k)}\|_1\}\right) \to 0$$

By Theorem 2.2 in Jin et al. (2021), when $q_0$ is sufficiently large, $\min_{i \in \mathcal{U}} \theta_i \geq q_0 \max_{i \in \mathcal{U}} \theta_i$ and $\beta_n \|\theta_{\mathcal{U}}\| \geq q_0 \sqrt{\log(n)}$ imply that $\epsilon \to 0$.

Hence, in all, we have $\mathbb{P}(\hat{y} \neq k^*) \to 0$, so the AngleMin+ estimate is consistent. $\square$

## J PROOF OF LEMMA 2

As mentioned in section D, in the main paper, for the smoothness and comprehensibility of the text, we do not present the most general form of Lemma 2. Here, we present both the original version, Lemma 2, and the generalized version, Lemma 2' below, where we relax the assumption that $\frac{\|\theta_{\mathcal{L}}^{(1)}\|_1}{\|\theta_{\mathcal{L}}^{(2)}\|_1} = \frac{\|\theta_{\mathcal{U}}^{(1)}\|_1}{\|\theta_{\mathcal{U}}^{(2)}\|_1} = 1$ to the much weaker assumption: the first part of condition (9) in the main text, which only assumes that $\|\theta^{(1)}\|_1$ and $\|\theta^{(2)}\|_1$ are of the same order.

**Lemma 2.** *Consider a DCBM with $K = 2$ and $P = (1 - b)I_2 + b\mathbf{1}_2\mathbf{1}_2'$. Suppose $\theta^* = o(1)$, $\frac{\theta^*}{\min_k \|\theta_{\mathcal{L}}^{(k)}\|_1} = o(1)$, $1 - b = o(1)$, $\frac{\|\theta_{\mathcal{L}}^{(1)}\|_1}{\|\theta_{\mathcal{L}}^{(2)}\|_1} = \frac{\|\theta_{\mathcal{U}}^{(1)}\|_1}{\|\theta_{\mathcal{U}}^{(2)}\|_1} = 1$. There exists a constant $c_4 > 0$ such that*

$$\inf_{\tilde{y}}\{\text{Risk}(\tilde{y})\} \geq c_4 \exp\left\{-2[1 + o(1)]\frac{(1-b)^2}{8} \cdot \theta^*(\|\theta_{\mathcal{L}}\|_1 + \|\theta_{\mathcal{U}}\|_1)\right\}, \quad (12)$$

*where the infimum is taken over all measurable functions of $A$, $X$, and parameters $\Pi_{\mathcal{L}}$, $\Pi_{\mathcal{U}}$, $\Theta$, $P$, $\theta^*$. In AngleMin+, suppose the second part of condition 9 holds with $c_3 = o(1)$, $\hat{\Pi}_{\mathcal{U}}$ is $\tilde{b}_0$-correct with $\tilde{b}_0 \overset{a.s.}{\to} 0$. There is a constant $C_4 > 0$ such that,*

$$\text{Risk}(\hat{y}) \leq C_4 \exp\left\{-[1 - o(1)]\frac{(1-b)^2}{8} \cdot \theta^* \frac{(\|\theta_{\mathcal{L}}\|_1^2 + \|\theta_{\mathcal{U}}\|_1^2)^2}{\|\theta_{\mathcal{L}}\|_1^3 + \|\theta_{\mathcal{U}}\|_1^3}\right\}. \quad (13)$$

**Lemma 2'.** *Consider a DCBM with $K = 2$ and $P = (1 - b)I_2 + b\mathbf{1}_2\mathbf{1}_2'$. Suppose $1 - b = o(1)$. There exists a constant $c_4 > 0$ such that*

$$\inf_{\tilde{y}}\{\text{Risk}(\tilde{y})\} \geq c_4 \exp\left\{-2[1 + o(1)]\frac{(1-b)^2}{8} \cdot \theta^*(\|\theta_{\mathcal{L}}\|_1 + \|\theta_{\mathcal{U}}\|_1)\right\}, \quad (75)$$

*where the infimum is taken over all measurable functions of $A$, $X$, and parameters $\Pi_{\mathcal{L}}$, $\Pi_{\mathcal{U}}$, $\Theta$, $P$, $\theta^*$. In AngleMin+, suppose condition 9 holds with $c_3 = o(1)$, $\theta^* = o(1)$, $\frac{\theta^*}{\min_k \|\theta_{\mathcal{L}}^{(k)}\|_1} = o(1)$, $\hat{\Pi}_{\mathcal{U}}$ is*

$\tilde{b}_0$-correct with $\tilde{b}_0 \overset{a.s.}{\to} 0$. There is a constant $C_4 > 0$ such that,

$$\text{Risk}(\hat{y}) \leq C_4 \exp\left(-[1 - o(1)]\frac{(1 - b)^2}{8} \cdot \theta^* \frac{4}{\frac{\|\theta_{\mathcal{L}}^{(1)}\|_1^3 + \|\theta_{\mathcal{U}}^{(1)}\|_1^3}{(\|\theta_{\mathcal{L}}^{(1)}\|_1^2 + \|\theta_{\mathcal{U}}^{(1)}\|_1^2)^2} + \frac{\|\theta_{\mathcal{L}}^{(2)}\|_1^3 + \|\theta_{\mathcal{U}}^{(2)}\|_1^3}{(\|\theta_{\mathcal{L}}^{(2)}\|_1^2 + \|\theta_{\mathcal{U}}^{(2)}\|_1^2)^2}}\right). \tag{76'}$$

When conditions of Lemma 2 hold, conditions of Lemma 2' hold. Also, with $\frac{\|\theta_{\mathcal{L}}^{(1)}\|_1}{\|\theta_{\mathcal{L}}^{(2)}\|_1} = \frac{\|\theta_{\mathcal{U}}^{(1)}\|_1}{\|\theta_{\mathcal{U}}^{(2)}\|_1} = 1$ assumed in Lemma 2, the results of Lemma 2' imply the results of 2. Therefore, it suffices to prove the generalized version, Lemma 2'.

We prove the lower bound (75) and upper bound (76') separately.

## J.1 PROOF OF LOWER BOUND (75)

*Proof.* Let $\mathbb{P}^{(1)}$ and $\mathbb{P}^{(2)}$ be the joint distribution of $A$ and $X$ given $\pi^* = e_1$ and $\pi^* = e_2$, respectively. For a random variable or vector or matrix $Y$, let $\mathbb{P}_Y^{(1)}$ and $\mathbb{P}_Y^{(2)}$ be the distribution of $Y$ given $\pi^* = e_1$ and $\pi^* = e_2$, respectively.

According to Theorem 2.2 in Section 2.4.2 of Tsybakov (2009),

$$\inf_{\tilde{y}}\{\text{Risk}(\tilde{y})\} \geq 2 \cdot \frac{1}{2}(1 - \sqrt{H^2(\mathbb{P}^{(1)}, \mathbb{P}^{(2)})(1 - H^2(\mathbb{P}^{(1)}, \mathbb{P}^{(2)})/4))}$$

$$= 1 - \sqrt{1 - \left(1 - \frac{1}{2}H^2(\mathbb{P}^{(1)}, \mathbb{P}^{(2)})\right)^2}$$

$$\geq 1 - \left(1 - \frac{1}{2}\left(1 - \frac{1}{2}H^2(\mathbb{P}^{(1)}, \mathbb{P}^{(2)})\right)^2\right)$$

$$= \frac{1}{2}\left(1 - \frac{1}{2}H^2(\mathbb{P}^{(1)}, \mathbb{P}^{(2)})\right)^2 \tag{76}$$

where

$$H^2(\mathbb{P}^{(1)}, \mathbb{P}^{(2)}) = \int \left(\sqrt{d\mathbb{P}^{(1)}} - \sqrt{d\mathbb{P}^{(2)}}\right)^2$$

is the Hellinger distance between $\mathbb{P}^{(1)}$ and $\mathbb{P}^{(2)}$.

As in Section 2.4 of Tsybakov (2009), one key property of Hellinger distance is that if $\mathbb{Q}^{(1)}$ and $\mathbb{Q}^{(2)}$ are product measures, $\mathbb{Q}^{(1)} = \otimes_{i=1}^N \mathbb{Q}_i^{(1)}$, $\mathbb{Q}^{(2)} = \otimes_{i=1}^N \mathbb{Q}_i^{(2)}$, then

$$H^2(\mathbb{Q}^{(1)}, \mathbb{Q}^{(2)}) = 2\left(1 - \prod_{i=1}^N\left(1 - \frac{H^2(\mathbb{Q}_i^{(1)}, \mathbb{Q}_i^{(2)})}{2}\right)\right) \tag{77}$$

Notice that for $k = 1, 2$, since according to DCBM, $A, X_1, ..., X_n$ are independent,

$$\mathbb{P}^{(k)} = \mathbb{P}_A^{(k)} \times \otimes_{i=1}^n \mathbb{P}_{X_i}^{(k)} \tag{78}$$

Combining (77) and (78), we obtain

$$H^2(\mathbb{P}^{(1)}, \mathbb{P}^{(2)}) = 2\left(1 - \left(1 - \frac{H^2(\mathbb{P}_A^{(1)}, \mathbb{P}_A^{(2)})}{2}\right)\prod_{i=1}^n\left(1 - \frac{H^2(\mathbb{P}_{X_i}^{(1)}, \mathbb{P}_{X_i}^{(2)})}{2}\right)\right) \tag{79}$$

Given $\pi^* = e_1$ and $\pi^* = e_2$, according to DCBM, the distribution of $A$ remains the same. As a result, $H^2(\mathbb{P}_A^{(1)}, \mathbb{P}_A^{(2)}) = 0$.

On the other hand, for $k = 1, 2$ and $i \in [n]$, according to DCBM model

$$\mathbb{P}_{X_i}^{(k)} \sim Bern(\theta^* \theta_i P_{kk_i})$$

where $k_i$ is the true label of node $i$.

As a result,

$$
\begin{aligned}
&1 - \frac{H^2(\mathbb{P}_{X_i}^{(1)}, \mathbb{P}_{X_i}^{(2)})}{2} \\
&= 1 - \frac{1}{2}\Big[\Big(\sqrt{\theta^*\theta_i P_{1k_i}} - \sqrt{\theta^*\theta_i P_{2k_i}}\Big)^2 + \Big(\sqrt{1 - \theta^*\theta_i P_{1k_i}} - \sqrt{1 - \theta^*\theta_i P_{2k_i}}\Big)^2\Big] \\
&= \sqrt{\theta^*\theta_i P_{1k_i}\theta^*\theta_i P_{2k_i}} + \sqrt{(1 - \theta^*\theta_i P_{1k_i})(1 - \theta^*\theta_i P_{2k_i})}
\end{aligned}
\tag{80}
$$

For $x, y \in \mathbb{R}$, denote $g(x, y) = \log(xy + \sqrt{(1 - x^2)(1 - y^2)})$.

Then, $1 - \frac{H^2(\mathbb{P}_{X_i}^{(1)}, \mathbb{P}_{X_i}^{(2)})}{2} = \exp g(\sqrt{\theta^*\theta_i P_{1k_i}}, \sqrt{\theta^*\theta_i P_{2k_i}})$

Hence, by (79)

$$
\begin{aligned}
1 - \frac{1}{2}H^2(\mathbb{P}^{(1)}, \mathbb{P}^{(2)}) &= \Big(1 - \frac{H^2(\mathbb{P}_A^{(1)}, \mathbb{P}_A^{(2)})}{2}\Big)\prod_{i=1}^{n}\Big(1 - \frac{H^2(\mathbb{P}_{X_i}^{(1)}, \mathbb{P}_{X_i}^{(2)})}{2}\Big) \\
&= \exp \sum_{i=1}^{n} g(\sqrt{\theta^*\theta_i P_{1k_i}}, \sqrt{\theta^*\theta_i P_{2k_i}})
\end{aligned}
\tag{81}
$$

Substituting (81) into (76), we obtain

$$
\inf_{\tilde{y}}\{\text{Risk}(\tilde{y})\} \geq \frac{1}{2}\exp\Big(2\sum_{i=1}^{n} g(\sqrt{\theta^*\theta_i P_{1k_i}}, \sqrt{\theta^*\theta_i P_{2k_i}})\Big)
\tag{82}
$$

To reduce the RHS of (82), we need to evaluate $g$. The following lemma shows that $g(x, y) \approx -\frac{1}{2}(x - y)^2$.

**Lemma 10.** *Suppose that $0 \leq x, y \leq a < 1$. Then,*

$$
g(x, y) \geq -\frac{1}{2(1 - a^2)^{\frac{3}{2}}}(x - y)^2
$$

*Proof.* We first prove a short inequality on log. For $z > 0$, define $g_3(z) = \log(z) - \frac{z-1}{z}$. Then

$$
\frac{d}{dz}g_3(z) = \frac{1}{z} - \frac{1}{z^2} = \frac{z - 1}{z^2}
$$

Therefore, when $z \leq 1$, $\frac{d}{dz}g_3(z) \leq 0$, $g_3(z)$ is monotonously deceasing, hence $g_3(z) \geq g_3(1) = 0$; when $z \geq 1$, $\frac{d}{dz}g_3(z) \geq 0$, $g_3(z)$ is monotonously increasing, hence $g_3(z) \geq g_3(1) = 0$.

In all, $g_3(z) \geq 0$, so

$$
log(z) \geq \frac{z - 1}{z}.
\tag{83}
$$

Since $x, y \in [0, 1]$, we could define $\phi_x = \arcsin x \in [0, \frac{\pi}{2}]$, $\phi_y = \arcsin y \in [0, \frac{\pi}{2}]$. Then,

$$
\begin{aligned}
g(x, y) &= g(\sin \phi_x, \sin \phi_y) \\
&= \log\Big(\sin \phi_x \sin \phi_y + \sqrt{(1 - \sin^2 \phi_x)(1 - \sin^2 \phi_y)}\Big) \\
&= \log(\sin \phi_x \sin \phi_y + \cos \phi_x \cos \phi_y) \\
&= \log(\cos(\phi_x - \phi_y))
\end{aligned}
$$

$$\text{(By (83))} \geq \frac{\cos(\phi_x - \phi_y) - 1}{\cos(\phi_x - \phi_y)}$$

$$= \frac{-2\sin^2 \frac{(\phi_x - \phi_y)}{2}}{\cos(\phi_x - \phi_y)}$$

$$= -\frac{1}{2} \frac{4\sin^2 \frac{(\phi_x - \phi_y)}{2}}{(\sin\phi_x - \sin\phi_y)^2} \frac{1}{\cos(\phi_x - \phi_y)} (\sin\phi_x - \sin\phi_y)^2$$

$$= -\frac{1}{2} \frac{4\sin^2 \frac{(\phi_x - \phi_y)}{2}}{(2\sin\frac{(\phi_x - \phi_y)}{2}\cos\frac{(\phi_x + \phi_y)}{2})^2} \frac{1}{\cos(\phi_x - \phi_y)} (x - y)^2$$

$$= -\frac{1}{2}(x - y)^2 \frac{1}{\cos(|\phi_x - \phi_y|)\cos^2 \frac{(\phi_x + \phi_y)}{2}} \tag{84}$$

Since $x, y \in [0, a]$, $\phi_x, \phi_y \in [0, \arcsin a]$. As a result, $|\phi_x - \phi_y|, \frac{(\phi_x + \phi_y)}{2} \in [0, \arcsin a]$. Plugging this result back into (84), we have

$$g(x, y) \geq -\frac{1}{2}(x - y)^2 \frac{1}{\cos(\arcsin a)\cos^2 \arcsin a} = -\frac{1}{2(1 - a^2)^{\frac{3}{2}}}(x - y)^2 \tag{85}$$

This concludes the proof. $\qquad\square$

Back to the proof of lower bound (75). Define $\theta^a = \sqrt{\theta^*\left(\max_{i\in[n]}\theta_i\right)(\max\{1, b\})}$, then for any $k \in \{1, 2\}$, $i \in [n]$, $\sqrt{\theta^*\theta_i P_{kk_i}} \leq \theta^a$. Therefore, applying Lemma 10 in (82), we have when $\theta^a < 1$,

$$\inf_{\tilde{y}}\{\text{Risk}(\tilde{y})\} \geq \frac{1}{2}\exp\left(2\sum_{i=1}^{n} -\frac{1}{2(1 - (\theta^a)^2)^{\frac{3}{2}}}\left(\sqrt{\theta^*\theta_i P_{1k_i}} - \sqrt{\theta^*\theta_i P_{2k_i}}\right)^2\right)$$

$$= \frac{1}{2}\exp\left(-\frac{1}{(1 - (\theta^a)^2)^{\frac{3}{2}}}\sum_{i=1}^{n}\theta^*\theta_i\left(\sqrt{P_{1k_i}} - \sqrt{P_{2k_i}}\right)^2\right)$$

$$= \frac{1}{2}\exp\left(-\frac{1}{(1 - (\theta^a)^2)^{\frac{3}{2}}}\sum_{i=1}^{n}\theta^*\theta_i(1 - \sqrt{b})^2\right)$$

$$= \frac{1}{2}\exp\left(-\frac{1}{(1 - (\theta^a)^2)^{\frac{3}{2}}}\theta^*\|\theta\|_1\frac{(1 - b)^2}{(1 + \sqrt{b})^2}\right)$$

$$= \frac{1}{2}\exp\left(-2\frac{4}{(1 + \sqrt{b})^2(1 - (\theta^a)^2)^{\frac{3}{2}}}\frac{(1 - b)^2}{8}\cdot\theta^*(\|\theta_{\mathcal{L}}\|_1 + \|\theta_{\mathcal{U}}\|_1)\right) \tag{86}$$

Since $\theta^* = o(1)$, $b = 1 - o(1)$, and by DCBM model, $\max_{i\in[n]}\theta_i \leq$, we have $\theta^a = \sqrt{\theta^*\left(\max_{i\in[n]}\theta_i\right)(\max\{1, b\})} = o(1)$. Since $b = 1 - o(1)$, $(1 + \sqrt{b})^2 \to 4$. Therefore, $\frac{4}{(1 + \sqrt{b})^2(1 - (\theta^a)^2)^{\frac{3}{2}}} = 1 - o(1)$. Substituting these results into (86), we obtain

$$\inf_{\tilde{y}}\{\text{Risk}(\tilde{y})\} \geq \frac{1}{2}\exp\left\{-2[1 + o(1)]\frac{(1 - b)^2}{8}\cdot\theta^*(\|\theta_{\mathcal{L}}\|_1 + \|\theta_{\mathcal{U}}\|_1)\right\}, \tag{87}$$

This concludes our proof of lower bound (75), with $c_4 = \frac{1}{2}$.

$\qquad\square$

## J.2 PROOF OF UPPER BOUND (76')

*Proof.* When $K = 2$, $\text{Risk}(\hat{y}) = \mathbb{P}(\hat{y} = 2|\pi^* = e_1) + \mathbb{P}(\hat{y} = 1|\pi^* = e_2)$. The evaluation of $\mathbb{P}(\hat{y} = 2|\pi^* = e_1)$ and $\mathbb{P}(\hat{y} = 1|\pi^* = e_2)$ are exactly the same. Without the loss of generosity, we would focus on $\mathbb{P}(\hat{y} = 2|\pi^* = e_1)$.

Recall that in the proof of 2, we define $v_k = f(A^{(k)}; \hat{\Pi}_{\mathcal{U}})$, $v^* = f(X; \hat{\Pi}_{\mathcal{U}})$, $\tilde{v}_k = f(\Omega^{(k)}; \hat{\Pi}_{\mathcal{U}})$, $\tilde{v}^* = f(EX; \hat{\Pi}_{\mathcal{U}})$, $k \in [K]$.

We have

$$
\begin{aligned}
\mathbb{P}(\hat{y} = 2 | \pi^* = e_1) &= \mathbb{P}(\psi(v_2, v^*) \geq \psi(v_1, v^*) | \pi^* = e_1) \\
&= \mathbb{P}\Big( \frac{\langle v^*, v_2 \rangle}{\|v^*\| \|v_2\|} \geq \frac{\langle v^*, v_1 \rangle}{\|v^*\| \|v_1\|} \Big| \pi^* = e_1 \Big) \\
&= \mathbb{P}\Big( \langle v^*, \Big( \frac{v_2}{\|v_2\|} - \frac{v_1}{\|v_1\|} \Big) \rangle \geq 0 \Big| \pi^* = e_1 \Big)
\end{aligned}
$$

$$(88)$$

Recall that in proof of Lemma 8 9, we define $\hat{\mathcal{C}}_k = \{i \in \mathcal{U} : \hat{\pi}_i = e_{k-K}\}, k = K+1, ..., 2K$. Let $w = \frac{v_2}{\|v_2\|} - \frac{v_1}{\|v_1\|}$. Since when $l \in [K]$,

$$
(v^*)_l = (f(X; \hat{\Pi}_{\mathcal{U}}))_l = X \mathbf{1}_{(l)} = \sum_{j \in \mathcal{C}_l \cap \mathcal{L}} X_j
$$

; when $l \in \{K+1, ..., 2K\}$,

$$
(v^*)_l = (f(X; \hat{\Pi}_{\mathcal{U}}))_l = X \mathbf{1}_{(l)} = \sum_{j \in \hat{\mathcal{C}}_l} X_j
$$

we have

$$
\begin{aligned}
\mathbb{P}(\hat{y} = 2 | \pi^* = e_1) &= \mathbb{P}\Big( \sum_{k=1}^{K} w_k \sum_{i \in \mathcal{C}_k \cap \mathcal{L}} X_i + \sum_{k=K+1}^{2K} w_k \sum_{i \in \hat{\mathcal{C}}_k} X_i \geq \Big| \pi^* = e_1 \Big) \\
&= \mathbb{P}\Big( \sum_{k=1}^{K} \sum_{i \in \mathcal{C}_k \cap \mathcal{L}} w_k(X_i - \mathbb{E}X_i) + \sum_{k=K+1}^{2K} \sum_{i \in \hat{\mathcal{C}}_k} w_k(X_i - \mathbb{E}X_i) \geq \\
&\quad - \sum_{k=1}^{K} \sum_{i \in \mathcal{C}_k \cap \mathcal{L}} w_k \mathbb{E}X_i - \sum_{k=K+1}^{2K} \sum_{i \in \hat{\mathcal{C}}_k} w_k \mathbb{E}X_i \Big| \pi^* = e_1 \Big) \\
&\leq \mathbb{P}\Big( \Big| \sum_{k=1}^{K} \sum_{i \in \mathcal{C}_k \cap \mathcal{L}} w_k(X_i - \mathbb{E}X_i) + \sum_{k=K+1}^{2K} \sum_{i \in \hat{\mathcal{C}}_k} w_k(X_i - \mathbb{E}X_i) \Big| \geq \\
&\quad \Big| \sum_{k=1}^{K} \sum_{i \in \mathcal{C}_k \cap \mathcal{L}} w_k \mathbb{E}X_i + \sum_{k=K+1}^{2K} \sum_{i \in \hat{\mathcal{C}}_k} w_k \mathbb{E}X_i \Big| \Big| \pi^* = e_1 \Big) \\
&= \mathbb{E}\Big[ \mathbb{P}\Big( \Big| \sum_{k=1}^{K} \sum_{i \in \mathcal{C}_k \cap \mathcal{L}} w_k(X_i - \mathbb{E}X_i) + \sum_{k=K+1}^{2K} \sum_{i \in \hat{\mathcal{C}}_k} w_k(X_i - \mathbb{E}X_i) \Big| \geq \\
&\quad \Big| \sum_{k=1}^{K} \sum_{i \in \mathcal{C}_k \cap \mathcal{L}} w_k \mathbb{E}X_i + \sum_{k=K+1}^{2K} \sum_{i \in \hat{\mathcal{C}}_k} w_k \mathbb{E}X_i \Big| \Big| A, \pi^* = e_1 \Big) \Big]
\end{aligned}
$$

$$(89)$$

Since $A$ and $X$ are independent and for $k \in [2K]$, $w_k$ is measurable with respect to $A$, given $A$, $\{w_k(X_i - \mathbb{E}X_i) : k \in [K], i \in \mathcal{C}_k \cap \mathcal{L}\} \cup \{w_k(X_i - \mathbb{E}X_i) : k \in [2K] \backslash [K], i \in \hat{\mathcal{C}}_k\}$ are a collection of independent random variables. Also, for any $k \in [2K]$, $i \in [n]$,

$$
\mathbb{E}[w_k(X_i - \mathbb{E}X_i) | A] = w_k(\mathbb{E}[X_i | A] - \mathbb{E}X_i) = w_k(\mathbb{E}X_i - \mathbb{E}X_i) = 0
$$

Furthermore,

$$
-1 \leq -\mathbb{E}X_i \leq X_i - \mathbb{E}X_i \leq X_i \leq 1
$$

So $|w_k(X_i - \mathbb{E}X_i)| \leq \max_{k \in [2K]} |w_k|$.

Additionally,

$$var(w_k(X_i - \mathbb{E}X_i)|A) = w_k^2 var(X_i) = w_k^2 \mathbb{E}X_i(1 - \mathbb{E}X_i) \le w_k^2 \mathbb{E}X_i$$

Let

$$t = \frac{1}{n}\Big|\sum_{k=1}^{K}\sum_{i \in \mathcal{C}_k \cap \mathcal{L}} w_k \mathbb{E}X_i + \sum_{k=K+1}^{2K}\sum_{i \in \hat{\mathcal{C}}_k} w_k \mathbb{E}X_i\Big| = \frac{1}{n}|\langle w, \tilde{v}^* \rangle|$$

$$\sigma^2 = \frac{1}{n}\Big(\sum_{k=1}^{K}\sum_{i \in \mathcal{C}_k \cap \mathcal{L}} w_k^2 \mathbb{E}X_i + \sum_{k=K+1}^{2K}\sum_{i \in \hat{\mathcal{C}}_k} w_k^2 \mathbb{E}X_i\Big) = \frac{1}{n}\langle w \circ w, \tilde{v}^* \rangle$$

where $w \circ w$ is defined as $(w_1^2, ..., w_{2K}^2)$.

Then by Lemma 7,

$$\mathbb{P}(\hat{y} = 2|\pi^* = e_1) \le \mathbb{E}\Big[\mathbb{P}\Big(\frac{1}{n}\Big|\sum_{k=1}^{K}\sum_{i \in \mathcal{C}_k \cap \mathcal{L}} w_k(X_i - \mathbb{E}X_i) + \sum_{k=K+1}^{2K}\sum_{i \in \hat{\mathcal{C}}_k} w_k(X_i - \mathbb{E}X_i)\Big| \ge$$

$$\frac{1}{n}\Big(\sum_{k=1}^{K}\sum_{i \in \mathcal{C}_k \cap \mathcal{L}} w_k \mathbb{E}X_i + \sum_{k=K+1}^{2K}\sum_{i \in \hat{\mathcal{C}}_k} w_k \mathbb{E}X_i\Big)\Big| A, \pi^* = e_1\Big)\Big]$$

$$\le 2\mathbb{E}\exp\left(-\frac{\frac{1}{2}nt^2}{\sigma^2 + \frac{1}{3}\max_{k \in [2K]}|w_k|t}\right)$$

$$= 2\mathbb{E}\exp\left(-\frac{\frac{1}{2}\langle w, \tilde{v}^* \rangle^2}{\langle w \circ w, \tilde{v}^* \rangle + \frac{1}{3}\max_{k \in [2K]}|w_k||\langle w, \tilde{v}^* \rangle|}\right) \tag{90}$$

By Lemma 8, when $\phi \ge 2\sqrt{2}\pi C_2 K^2 \frac{\|\theta_{\mathcal{L}}^{(k)}\|^2}{\|\theta_{\mathcal{L}}^{(k)}\|_1 \|\theta\|_1}$,

$$\mathbb{P}\Big(|(v_k - \tilde{v}_k)_l| \ge \frac{1}{\pi\sqrt{K}}\phi\|\tilde{v}_k\|\Big) \le 2\exp\left(-\frac{C_6}{C^2}\phi^2\|\theta_{\mathcal{L}}^{(k)}\|_1 \|\theta\|_1\right)$$

Take

$$\phi = \max\{2\sqrt{2}\pi C_2 K^2 \frac{\|\theta_{\mathcal{L}}^{(k)}\|^2}{\|\theta_{\mathcal{L}}^{(k)}\|_1 \|\theta\|_1}, |1 - b|\Big(\frac{\theta^*}{\min_{k \in [K]}\|\theta_{\mathcal{L}}^{(k)}\|_1}\Big)^{0.25}\}$$

Then because $\frac{\theta^*}{\min_{k \in [K]}\|\theta_{\mathcal{L}}^{(k)}\|_1} = o(1)$ and by condition 9, $\frac{\|\theta_{\mathcal{L}}^{(k)}\|^2}{\|\theta_{\mathcal{L}}^{(k)}\|_1 \|\theta\|_1} \le c_3 \beta_n = o(|1 - b|)$, we have $\phi = o(|1 - b|)$. Also,

$$\mathbb{P}\Big(\exists k \in [K], l \in [2K], |(v_k - \tilde{v}_k)_l| \ge \frac{1}{\pi\sqrt{K}}\phi\|\tilde{v}_k\|\Big)$$

$$\le \sum_{k \in [K]}\sum_{l \in [2K]} \mathbb{P}\Big(|(v_k - \tilde{v}_k)_l| \ge \frac{1}{\pi\sqrt{K}}\phi\|\tilde{v}_k\|\Big)$$

$$\le \sum_{k \in [K]}\sum_{l \in [2K]} 2\exp\left(-\frac{C_6}{C^2}\phi^2\|\theta_{\mathcal{L}}^{(k)}\|_1 \|\theta\|_1\right)$$

$$= \sum_{k \in [K]}\sum_{l \in [2K]} 2\exp\left(-\frac{C_6}{C^2}(1 - b)^2\Big(\frac{\theta^*}{\min_{k \in [K]}\|\theta_{\mathcal{L}}^{(k)}\|_1}\Big)^{-0.5}\frac{\|\theta_{\mathcal{L}}^{(k)}\|_1}{\min_{k \in [K]}\|\theta_{\mathcal{L}}^{(k)}\|_1}\theta^*\|\theta\|_1\right)$$

$$\le 4K^2 \exp\left(-\frac{1}{o(1)}(1 - b)^2\theta^*\|\theta\|_1\right)$$

$$\ll \inf_{\tilde{y}}\{\text{Risk}(\tilde{y})\} \tag{91}$$

Hence, we can focus on the case where for all $k \in [K]$, $l \in [2K]$, $|(v_k - \tilde{v}_k)_l| = o(|1-b|)\|\tilde{v}_k\|$. Until the end of the proof, we assume that we are under this case.

We first evaluate $\langle w, \tilde{v}^* \rangle$. Let $\tilde{w} = \frac{\tilde{v}_2}{\|\tilde{v}_2\|} - \frac{\tilde{v}_1}{\|\tilde{v}_1\|}$. Then,

$$
\begin{aligned}
|\langle w, \tilde{v}^* \rangle - \langle \tilde{w}, \tilde{v}^* \rangle| &= |\langle (\frac{v_2}{\|v_2\|} - \frac{v_1}{\|v_1\|}) - (\frac{\tilde{v}_2}{\|\tilde{v}_2\|} - \frac{\tilde{v}_1}{\|\tilde{v}_1\|}), \tilde{v}^* \rangle| \\
&= \|\tilde{v}^*\| \cdot |\langle (\frac{v_2}{\|v_2\|} - \frac{\tilde{v}_2}{\|\tilde{v}_2\|}) - (\frac{v_1}{\|v_1\|} - \frac{\tilde{v}_1}{\|\tilde{v}_1\|}), \frac{\tilde{v}^*}{\|\tilde{v}^*\|} \rangle| \\
&= \|\tilde{v}^*\| \cdot |(\cos\psi(v_2, \tilde{v}^*) - \cos\psi(\tilde{v}_2, \tilde{v}^*)) - (\cos\psi(v_1, \tilde{v}^*) - \cos\psi(\tilde{v}_1, \tilde{v}^*))| \\
&= \|\tilde{v}^*\| \cdot |-2\sin\frac{\psi(v_2, \tilde{v}^*) - \psi(\tilde{v}_2, \tilde{v}^*)}{2}\sin\frac{\psi(v_2, \tilde{v}^*) + \psi(\tilde{v}_2, \tilde{v}^*)}{2} \\
&\qquad + 2\sin\frac{\psi(v_1, \tilde{v}^*) - \psi(\tilde{v}_1, \tilde{v}^*)}{2}\sin\frac{\psi(v_1, \tilde{v}^*) + \psi(\tilde{v}_1, \tilde{v}^*)}{2}| \\
&\leq 2\|\tilde{v}^*\| \cdot \sin\frac{|\psi(v_2, \tilde{v}^*) - \psi(\tilde{v}_2, \tilde{v}^*)|}{2}\sin\frac{\psi(v_2, \tilde{v}^*) + \psi(\tilde{v}_2, \tilde{v}^*)}{2} \\
&\qquad + 2\|\tilde{v}^*\| \cdot \sin\frac{|\psi(v_1, \tilde{v}^*) - \psi(\tilde{v}_1, \tilde{v}^*)|}{2}\sin\frac{\psi(v_1, \tilde{v}^*) + \psi(\tilde{v}_1, \tilde{v}^*)}{2} \\
\text{(By Lemma 4)} &\leq 2\|\tilde{v}^*\| \cdot \sin\frac{\psi(v_2, \tilde{v}_2)}{2}\sin\frac{2\psi(\tilde{v}_2, \tilde{v}^*) + \psi(v_2, \tilde{v}_2)}{2} \\
&\qquad + 2\|\tilde{v}^*\| \cdot \sin\frac{\psi(v_1, \tilde{v}_1)}{2}\sin\frac{2\psi(\tilde{v}_1, \tilde{v}^*) + \psi(v_1, \tilde{v}_1)}{2}
\end{aligned}
\tag{92}
$$

Since $\pi^* = 1$, $\tilde{b}_0 \to 0$, by Theorem 1, $\psi(\tilde{v}_1, \tilde{v}^*) = \psi_1 = 0$, $\psi(\tilde{v}_2, \tilde{v}^*) = \psi_2 \geq c_0\beta_n = c_0|1-b|$. On the other hand, that for all $k \in [K]$, $l \in [2K]$, $|(v_k - \tilde{v}_k)_l| = o(|1-b|)\|\tilde{v}_k\|$ indicates that $\|v_k - \tilde{v}_k\| = o(|1-b|)\|\tilde{v}_k\|$. By lemma 5, this implies that $\psi(v_1, \tilde{v}_1) = o(|1-b|), k = 1, 2$. Therefore,

$$
\begin{aligned}
|\langle w, \tilde{v}^* \rangle - \langle \tilde{w}, \tilde{v}^* \rangle| &\leq o(1) \cdot 2\|\tilde{v}^*\| \sin\frac{\psi(\tilde{v}_2, \tilde{v}^*)}{2}\sin\psi(\tilde{v}_2, \tilde{v}^*) \\
&= o(1) \cdot 2\|\tilde{v}^*\| \sin\frac{\psi(\tilde{v}_2, \tilde{v}^*)}{2}2\sin\frac{\psi(\tilde{v}_2, \tilde{v}^*)}{2}\cos\frac{\psi(\tilde{v}_2, \tilde{v}^*)}{2} \\
&= o(1) \cdot \|\tilde{v}^*\| \sin^2\frac{\psi(\tilde{v}_2, \tilde{v}^*)}{2}\cos\frac{\psi(\tilde{v}_2, \tilde{v}^*)}{2} \\
&\leq o(1) \cdot \|\tilde{v}^*\| \sin^2\frac{\psi(\tilde{v}_2, \tilde{v}^*)}{2} \\
&\leq o(1) \cdot \|\tilde{v}^*\|(1 - \cos\psi(\tilde{v}_2, \tilde{v}^*)) \\
&= o(1) \cdot \|\tilde{v}^*\|(\cos\psi(\tilde{v}_1, \tilde{v}^*) - \cos\psi(\tilde{v}_2, \tilde{v}^*)) \\
&= o(1) \cdot \|\tilde{v}^*\|\langle \frac{\tilde{v}_1}{\|\tilde{v}_1\|} - \frac{\tilde{v}_2}{\|\tilde{v}_2\|}, \frac{\tilde{v}^*}{\|\tilde{v}^*\|} \rangle \\
&= o(1) \cdot (-\langle \tilde{w}, \tilde{v}^* \rangle) \\
&\leq o(1) \cdot |\langle \tilde{w}, \tilde{v}^* \rangle|
\end{aligned}
\tag{93}
$$

Therefore, $\langle w, \tilde{v}^* \rangle = (1 + o(1))\langle \tilde{w}, \tilde{v}^* \rangle$.

Let $\eta_{kl} = \sum_{\pi_i = e_k, \hat{\pi}_i = e_l} \theta_i$. Let $\mu_a^{(k)} = \|\theta_a^{(k)}\|_1, a \in \{\mathcal{L}, \mathcal{U}\}, k \in [K]$. Then,

$$
\begin{aligned}
\tilde{v}_1 &= \mu_{\mathcal{L}}^{(1)}(\mu_{\mathcal{L}}^{(1)}, b\mu_{\mathcal{L}}^{(2)}, \mu_{\mathcal{U}}^{(1)} - \eta_{12} + b\eta_{21}, b\mu_{\mathcal{U}}^{(2)} + \eta_{12} - b\eta_{21}) \\
\tilde{v}_2 &= \mu_{\mathcal{L}}^{(2)}(b\mu_{\mathcal{L}}^{(1)}, \mu_{\mathcal{L}}^{(2)}, b\mu_{\mathcal{U}}^{(1)} - b\eta_{12} + \eta_{21}, \mu_{\mathcal{U}}^{(2)} + b\eta_{12} - \eta_{21}) \\
\tilde{v}^* &= \theta^*(\mu_{\mathcal{L}}^{(1)}, b\mu_{\mathcal{L}}^{(2)}, \mu_{\mathcal{U}}^{(1)} - \eta_{12} + b\eta_{21}, b\mu_{\mathcal{U}}^{(2)} + \eta_{12} - b\eta_{21})
\end{aligned}
$$

Hence,

$$
-\langle \tilde{w}, \tilde{v}^* \rangle = \|\tilde{v}^*\|\Big( \frac{\langle \tilde{v}_1 \tilde{v}^* \rangle}{\|\tilde{v}_1\|\|\tilde{v}^*\|} - \frac{\langle \tilde{v}_2 \tilde{v}^* \rangle}{\|\tilde{v}_2\|\|\tilde{v}^*\|} \Big)
$$

$$= \|\tilde{v}^*\| \left(1 - \sqrt{1 - (1-b^2)^2 (\mu_{\mathcal{L}}^{(1)})^2 (\mu_{\mathcal{L}}^{(2)})^2 \frac{\mathcal{I}_1 \mathcal{I}_2 - 4\mathcal{I}_3^2}{\|\tilde{v}_1\|^2 \|\tilde{v}_2\|^2}}\right) \tag{94}$$

where

$$\mathcal{I}_1 = (\mu_{\mathcal{L}}^{(1)})^2 + (\mu_{\mathcal{U}}^{(1)} - \eta_{12})^2 + \eta_{21}^2$$

$$\mathcal{I}_2 = (\mu_{\mathcal{L}}^{(2)})^2 + (\mu_{\mathcal{U}}^{(2)} - \eta_{21})^2 + \eta_{12}^2$$

$$\mathcal{I}_3 = (\mu_{\mathcal{U}}^{(1)} - \eta_{12})\eta_{21} + (\mu_{\mathcal{U}}^{(2)} - \eta_{21})\eta_{12}$$

Notice that

$$
\begin{aligned}
&|\mathcal{I}_1 \mathcal{I}_2 - 4\mathcal{I}_3^2 - ((\mu_{\mathcal{L}}^{(1)})^2 + (\mu_{\mathcal{U}}^{(1)})^2)((\mu_{\mathcal{L}}^{(2)})^2 + (\mu_{\mathcal{U}}^{(2)})^2)| \\
&\leq ((\mu_{\mathcal{L}}^{(1)})^2 + (\mu_{\mathcal{U}}^{(1)})^2 + (\mu_{\mathcal{L}}^{(2)})^2 + (\mu_{\mathcal{U}}^{(2)})^2)(\eta_{12}^2 + \eta_{21}^2) \\
&\quad + 2\eta_{12}\mu_{\mathcal{U}}^{(1)}((\mu_{\mathcal{L}}^{(2)})^2 + (\mu_{\mathcal{U}}^{(2)})^2) + 2\eta_{21}\mu_{\mathcal{U}}^{(2)}((\mu_{\mathcal{L}}^{(1)})^2 + (\mu_{\mathcal{U}}^{(1)})^2) \\
&\quad + 4\mathcal{I}_3^2 \\
&\leq ((\mu_{\mathcal{L}}^{(1)})^2 + (\mu_{\mathcal{U}}^{(1)})^2 + (\mu_{\mathcal{L}}^{(2)})^2 + (\mu_{\mathcal{U}}^{(2)})^2)(\eta_{12} + \eta_{21})^2 \\
&\quad + 2\eta_{12}\mu_{\mathcal{U}}^{(1)}((\mu_{\mathcal{L}}^{(2)})^2 + (\mu_{\mathcal{U}}^{(2)})^2) + 2\eta_{21}\mu_{\mathcal{U}}^{(2)}((\mu_{\mathcal{L}}^{(1)})^2 + (\mu_{\mathcal{U}}^{(1)})^2) \\
&\quad + 8(\mu_{\mathcal{U}}^{(1)})^2 \eta_{21}^2 + 8(\mu_{\mathcal{U}}^{(2)})^2 \eta_{12}^2 \\
&\leq \|\theta\|_1^2 (\eta_{12} + \eta_{21})^2 + 2\|\theta\|_1^3 (\eta_{12} + \eta_{21}) + 8\|\theta\|_1^2 (\eta_{12} + \eta_{21})^2 \\
&\leq (9\tilde{b}_0^2 + 2\tilde{b}_0)\|\theta\|_1^4 \\
&= o(1) \cdot \|\theta\|_1^4
\end{aligned} \tag{95}
$$

On the other hand,

$$
\begin{aligned}
((\mu_{\mathcal{L}}^{(1)})^2 + (\mu_{\mathcal{U}}^{(1)})^2)((\mu_{\mathcal{L}}^{(2)})^2 + (\mu_{\mathcal{U}}^{(2)})^2) &\geq \frac{1}{4}(\mu_{\mathcal{L}}^{(1)} + \mu_{\mathcal{U}}^{(1)})^2 (\mu_{\mathcal{L}}^{(2)} + \mu_{\mathcal{U}}^{(2)})^2 \\
&= \frac{1}{4}\|\theta^{(1)}\|_1^2 \|\theta^{(2)}\|_1^2 \\
&= \frac{1}{4} \min_{k \in [K]} \|\theta^{(k)}\|_1^2 \max_{k \in [K]} \|\theta^{(k)}\|_1^2 \\
\text{(By condition (9))} &\geq \frac{1}{4C_2} \max_{k \in [K]} \|\theta^{(k)}\|_1^4 \\
&\geq \frac{1}{64C_2} \|\theta\|_1^4
\end{aligned} \tag{96}
$$

Therefore,

$$\mathcal{I}_1 \mathcal{I}_2 - 4\mathcal{I}_3^2 = (1 + o(1))((\mu_{\mathcal{L}}^{(1)})^2 + (\mu_{\mathcal{U}}^{(1)})^2)((\mu_{\mathcal{L}}^{(2)})^2 + (\mu_{\mathcal{U}}^{(2)})^2)$$

Similarly,

$$\frac{\|\tilde{v}_1\|^2 \|\tilde{v}_2\|^2}{(\mu_{\mathcal{L}}^{(1)})^2 (\mu_{\mathcal{L}}^{(2)})^2} = (1 + o(1))((\mu_{\mathcal{L}}^{(1)})^2 + (\mu_{\mathcal{U}}^{(1)})^2 + (\mu_{\mathcal{L}}^{(2)})^2 + (\mu_{\mathcal{U}}^{(2)})^2)^2$$

$$\|\tilde{v}^*\| = (1 + o(1))\theta^* \sqrt{(\mu_{\mathcal{L}}^{(1)})^2 + (\mu_{\mathcal{U}}^{(1)})^2 + (\mu_{\mathcal{L}}^{(2)})^2 + (\mu_{\mathcal{U}}^{(2)})^2}$$

Therefore,

$$
\begin{aligned}
-\langle \tilde{w}, \tilde{v}^* \rangle &= \|\tilde{v}^*\| \left( 1 - \sqrt{1 - (1+o(1))(1-b^2)^2 \frac{((\mu_{\mathcal{L}}^{(1)})^2 + (\mu_{\mathcal{U}}^{(1)})^2)((\mu_{\mathcal{L}}^{(2)})^2 + (\mu_{\mathcal{U}}^{(2)})^2)}{((\mu_{\mathcal{L}}^{(1)})^2 + (\mu_{\mathcal{U}}^{(1)})^2 + (\mu_{\mathcal{L}}^{(2)})^2 + (\mu_{\mathcal{U}}^{(2)})^2)^2}} \right) \\
&= \frac{1}{2}(1+o(1))\theta^*(1-b^2)^2 \frac{((\mu_{\mathcal{L}}^{(1)})^2 + (\mu_{\mathcal{U}}^{(1)})^2)((\mu_{\mathcal{L}}^{(2)})^2 + (\mu_{\mathcal{U}}^{(2)})^2)}{((\mu_{\mathcal{L}}^{(1)})^2 + (\mu_{\mathcal{U}}^{(1)})^2 + (\mu_{\mathcal{L}}^{(2)})^2 + (\mu_{\mathcal{U}}^{(2)})^2)^{1.5}} \\
&= 2(1+o(1))\theta^*(1-b)^2 \frac{((\mu_{\mathcal{L}}^{(1)})^2 + (\mu_{\mathcal{U}}^{(1)})^2)((\mu_{\mathcal{L}}^{(2)})^2 + (\mu_{\mathcal{U}}^{(2)})^2)}{((\mu_{\mathcal{L}}^{(1)})^2 + (\mu_{\mathcal{U}}^{(1)})^2 + (\mu_{\mathcal{L}}^{(2)})^2 + (\mu_{\mathcal{U}}^{(2)})^2)^{1.5}}
\end{aligned}
\tag{97}
$$

We turn to $\langle w \circ w, \tilde{v}^* \rangle$.

Denote that

$$
\tilde{\nu}_1 = \frac{1}{\mu_{\mathcal{L}}^{(1)}} \tilde{v}_1 = (\mu_{\mathcal{L}}^{(1)}, b\mu_{\mathcal{L}}^{(2)}, \mu_{\mathcal{U}}^{(1)} - \eta_{12} + b\eta_{21}, b\mu_{\mathcal{U}}^{(2)} + \eta_{12} - b\eta_{21})
$$

$$
\tilde{\nu}_2 = \frac{1}{\mu_{\mathcal{L}}^{(2)}} \tilde{v}_2 = (b\mu_{\mathcal{L}}^{(1)}, \mu_{\mathcal{L}}^{(2)}, b\mu_{\mathcal{U}}^{(1)} - b\eta_{12} + \eta_{21}, \mu_{\mathcal{U}}^{(2)} + b\eta_{12} - \eta_{21})
$$

One simple but useful fact is that $\|\tilde{\nu}_1\|, \|\tilde{\nu}_2\| = O(\|\theta\|_1)$. The reason is that for $k \in [K]$

$$
\|\tilde{\nu}_k\| = O(\|\tilde{\nu}_k\|_1) = O(\mu_{\mathcal{L}}^{(1)} + b\mu_{\mathcal{L}}^{(2)} + \mu_{\mathcal{U}}^{(1)} + b\mu_{\mathcal{U}}^{(2)}) = O(\|\theta\|_1)
$$

Notice that

$$
\begin{aligned}
\tilde{w}_3 &= \frac{b\mu_{\mathcal{U}}^{(1)} - b\eta_{12} + \eta_{21}}{\|\tilde{\nu}_2\|} - \frac{\mu_{\mathcal{U}}^{(1)} - \eta_{12} + b\eta_{21}}{\|\tilde{\nu}_1\|} \\
&= (b\mu_{\mathcal{U}}^{(1)} - b\eta_{12} + \eta_{21})\left(\frac{1}{\|\tilde{\nu}_2\|} - \frac{1}{\|\tilde{\nu}_1\|}\right) - \frac{1-b}{\|\tilde{\nu}_1\|}(\mu_{\mathcal{U}}^{(1)} - \eta_{12} - \eta_{21}) \\
&= -\frac{b\mu_{\mathcal{U}}^{(1)} - b\eta_{12} + \eta_{21}}{\|\tilde{\nu}_1\|}\left(1 - \frac{\|\tilde{\nu}_1\|}{\|\tilde{\nu}_2\|}\right) - \frac{1-b}{\|\tilde{\nu}_1\|}\mu_{\mathcal{U}}^{(1)} + o(1-b) \\
&= -\frac{b\mu_{\mathcal{U}}^{(1)} - b\eta_{12} + \eta_{21}}{\|\tilde{\nu}_1\|}\left(1 - \sqrt{1 - \frac{\|\tilde{\nu}_2\|^2 - \|\tilde{\nu}_1\|^2}{\|\tilde{\nu}_2\|^2}}\right) - \frac{1-b}{\|\tilde{v}_1\|}\mu_{\mathcal{U}}^{(1)} + o(1-b)
\end{aligned}
\tag{98}
$$

Since

$$
\begin{aligned}
\|\tilde{\nu}_2\|^2 - \|\tilde{\nu}_1\|^2 &= (1-b^2)((\mu_{\mathcal{L}}^{(2)})^2 + (\mu_{\mathcal{U}}^{(2)} - \eta_{21})^2 - (\mu_{\mathcal{L}}^{(1)})^2 - (\mu_{\mathcal{U}}^{(1)} - \eta_{12})^2) \\
&= (1-b^2)((\mu_{\mathcal{L}}^{(2)})^2 + (\mu_{\mathcal{U}}^{(2)})^2 - (\mu_{\mathcal{L}}^{(1)})^2 - (\mu_{\mathcal{U}}^{(1)})^2 + o(1) \cdot \|\theta\|_1^2)
\end{aligned}
\tag{99}
$$

We have

$$
\begin{aligned}
\tilde{w}_3 &= -\frac{b\mu_{\mathcal{U}}^{(1)} + o(1) \cdot \|\theta\|_1}{\|\tilde{\nu}_1\|} \frac{1}{2\|\tilde{\nu}_2\|^2}(1-b^2)((\mu_{\mathcal{L}}^{(2)})^2 + (\mu_{\mathcal{U}}^{(2)})^2 - (\mu_{\mathcal{L}}^{(1)})^2 - (\mu_{\mathcal{U}}^{(1)})^2 + o(1) \cdot \|\theta\|_1^2) \\
&\quad - \frac{1-b}{\|\tilde{\nu}_1\|}\mu_{\mathcal{U}}^{(1)} + o(1-b) \\
&= -\frac{1-b}{\|\tilde{\nu}_1\|}\mu_{\mathcal{U}}^{(1)}\left(1 + \frac{(\mu_{\mathcal{L}}^{(2)})^2 + (\mu_{\mathcal{U}}^{(2)})^2 - (\mu_{\mathcal{L}}^{(1)})^2 - (\mu_{\mathcal{U}}^{(1)})^2}{\|\tilde{\nu}_2\|^2}\right) + o(1-b)
\end{aligned}
\tag{100}
$$

Since for all $k \in [K]$, $l \in [2K]$, $|(v_k - \tilde{v}_k)_l| = o(|1-b|) \|\tilde{v}_k\|$, we have $|w_k - \tilde{w}_k| = o(|1-b|)$, $k = 1, 2, ..., 2K$. Hence, denote $\gamma = \frac{(\mu_{\mathcal{L}}^{(2)})^2 + (\mu_{\mathcal{U}}^{(2)})^2 - (\mu_{\mathcal{L}}^{(1)})^2 - (\mu_{\mathcal{U}}^{(1)})^2}{\|\tilde{\nu}_2\|^2}$, we obtain

$$w_3 = -\frac{1-b}{\|\tilde{\nu}_1\|} \mu_{\mathcal{U}}^{(1)}(1 + \gamma) + o(1-b) \tag{101}$$

Similarly, we can show that

$$w_1 = -\frac{1-b}{\|\tilde{\nu}_1\|} \mu_{\mathcal{L}}^{(1)}(1 + \gamma) + o(1-b) \tag{102}$$

$$w_2 = \frac{1-b}{\|\tilde{\nu}_2\|} \mu_{\mathcal{L}}^{(2)}(1 - \gamma) + o(1-b) \tag{103}$$

$$w_4 = \frac{1-b}{\|\tilde{\nu}_2\|} \mu_{\mathcal{U}}^{(2)}(1 - \gamma) + o(1-b) \tag{104}$$

As a result,

$$\langle w \circ w, \tilde{\nu}^* \rangle$$
$$= \sum_{a \in \{\mathcal{L},\mathcal{U}\}} \sum_{k=1}^{K} (\frac{1-b}{\|\tilde{\nu}_k\|} \mu_a^{(k)}(1 - (-1)^k \gamma) + o(1-b))^2 \theta^* \mu_a^{(k)}$$
$$= \theta^*(1-b)^2 \sum_{a \in \{\mathcal{L},\mathcal{U}\}} \sum_{k=1}^{K} \left( \frac{(\mu_a^{(k)})^3 (1 - (-1)^k \gamma)^2}{\|\tilde{\nu}_k\|^2} + 2o(1) \frac{(\mu_a^{(k)})^2 (1 - (-1)^k \gamma)}{\|\tilde{\nu}_k\|^2} + o(1)^2 \frac{\mu_a^{(k)}}{\|\tilde{\nu}_k\|^2} \right) \tag{105}$$

When $\|\theta\|_1 = o(1)$, the bounds (75) and (76') both become trivial, so we can focus on the case where $\|\theta\|_1 \geq O(1)$.

In this case, $\frac{(\mu_a^{(k)})^2 (1-(-1)^k\gamma)}{\|\tilde{\nu}_k\|^2} \leq O(1)$, $\frac{\mu_a^{(k)}}{\|\tilde{\nu}_k\|^2} \leq O(1)$. On the other hand, for $k \in [K]$,

$$\sum_{a \in \{\mathcal{L},\mathcal{U}\}} \frac{(\mu_a^{(k)})^3}{\|\tilde{\nu}_k\|^2} = \frac{(\mu_{\mathcal{L}}^{(k)})^3 + (\mu_{\mathcal{U}}^{(k)})^3}{\|\tilde{\nu}_k\|^2}$$

$$\text{(Holder Inequality)} \geq \frac{(\mu_{\mathcal{L}}^{(k)} + \mu_{\mathcal{U}}^{(k)})^3}{4\|\tilde{\nu}_k\|^2}$$

$$= \frac{\|\theta^{(k)}\|_1^3}{4\|\tilde{\nu}_k\|^2}$$

$$\geq \frac{\min_{k \in [K]} \|\theta^{(k)}\|_1^3}{4\|\tilde{\nu}_k\|^2}$$

$$\text{(By (9))} \geq \frac{\max_{k \in [K]} \|\theta^{(k)}\|_1^3}{4C_2^3 \|\tilde{\nu}_k\|^2}$$

$$\geq \frac{\|\theta\|_1^3}{4K^3 C_2^3 \|\tilde{\nu}_k\|^2}$$

$$\geq O(1) \tag{106}$$

Since $1 - \gamma$ and $1 + \gamma$ cannot be both $o(1)$, we have

$$\sum_{a \in \{\mathcal{L},\mathcal{U}\}} \sum_{k=1}^{K} \frac{(\mu_a^{(k)})^3 (1 - (-1)^k \gamma)^2}{\|\tilde{\nu}_k\|^2} \geq O(1)$$

Therefore,

$$
\begin{aligned}
\langle w \circ w, \tilde{\nu}^* \rangle &= (1 + o(1))\theta^*(1 - b)^2 \sum_{a \in \{\mathcal{L}, \mathcal{U}\}} \sum_{k=1}^{K} \frac{(\mu_a^{(k)})^3 (1 - (-1)^k \gamma)^2}{\|\tilde{\nu}_k\|^2} \\
&= (1 + o(1))4\theta^*(1 - b)^2 \\
&\quad \cdot \frac{((\mu_{\mathcal{L}}^{(1)})^3 + (\mu_{\mathcal{U}}^{(1)})^3)((\mu_{\mathcal{L}}^{(2)})^2 + (\mu_{\mathcal{U}}^{(2)})^2)^2 + ((\mu_{\mathcal{L}}^{(2)})^3 + (\mu_{\mathcal{U}}^{(2)})^3)((\mu_{\mathcal{L}}^{(1)})^2 + (\mu_{\mathcal{U}}^{(1)})^2)^2}{((\mu_{\mathcal{L}}^{(1)})^2 + (\mu_{\mathcal{U}}^{(1)})^2 + (\mu_{\mathcal{L}}^{(2)})^2 + (\mu_{\mathcal{U}}^{(2)})^2)^3}
\end{aligned}
\tag{107}
$$

By (101) (102) (103) (104), $\max_{k \in [2K]} |w_k| = O(1 - b) = o(1)$. Substituting this result together with (97) and (107) into (90), we obtain

$$
\begin{aligned}
&\mathbb{P}(\hat{y} = 2 | \pi^* = e_1) \\
&= 2 \exp\left( -(1 - o(1)) \frac{(1 - b)^2}{8} \cdot \theta^* \frac{4}{\frac{\|\theta_{\mathcal{L}}^{(1)}\|_1^3 + \|\theta_{\mathcal{U}}^{(1)}\|_1^3}{(\|\theta_{\mathcal{L}}^{(1)}\|_1^2 + \|\theta_{\mathcal{U}}^{(1)}\|_1^2)^2} + \frac{\|\theta_{\mathcal{L}}^{(2)}\|_1^3 + \|\theta_{\mathcal{U}}^{(2)}\|_1^3}{(\|\theta_{\mathcal{L}}^{(2)}\|_1^2 + \|\theta_{\mathcal{U}}^{(2)}\|_1^2)^2}} \right)
\end{aligned}
\tag{108}
$$

Similarly,

$$
\begin{aligned}
&\mathbb{P}(\hat{y} = 1 | \pi^* = e_2) \\
&\leq 2 \exp\left( -(1 - o(1)) \frac{(1 - b)^2}{8} \cdot \theta^* \frac{4}{\frac{\|\theta_{\mathcal{L}}^{(1)}\|_1^3 + \|\theta_{\mathcal{U}}^{(1)}\|_1^3}{(\|\theta_{\mathcal{L}}^{(1)}\|_1^2 + \|\theta_{\mathcal{U}}^{(1)}\|_1^2)^2} + \frac{\|\theta_{\mathcal{L}}^{(2)}\|_1^3 + \|\theta_{\mathcal{U}}^{(2)}\|_1^3}{(\|\theta_{\mathcal{L}}^{(2)}\|_1^2 + \|\theta_{\mathcal{U}}^{(2)}\|_1^2)^2}} \right)
\end{aligned}
\tag{109}
$$

Therefore,

$$
\text{Risk}(\hat{y}) \leq 4 \exp\left( -(1 - o(1)) \frac{(1 - b)^2}{8} \cdot \theta^* \frac{4}{\frac{\|\theta_{\mathcal{L}}^{(1)}\|_1^3 + \|\theta_{\mathcal{U}}^{(1)}\|_1^3}{(\|\theta_{\mathcal{L}}^{(1)}\|_1^2 + \|\theta_{\mathcal{U}}^{(1)}\|_1^2)^2} + \frac{\|\theta_{\mathcal{L}}^{(2)}\|_1^3 + \|\theta_{\mathcal{U}}^{(2)}\|_1^3}{(\|\theta_{\mathcal{L}}^{(2)}\|_1^2 + \|\theta_{\mathcal{U}}^{(2)}\|_1^2)^2}} \right).
\tag{110}
$$

Taking $C_4 = 4$, we conclude the proof. $\qquad \square$

## K  PROOF OF THEOREM 3

**Theorem 3.** *Suppose the conditions of Corollary 1 hold, where $b_0$ is properly small , and suppose that $\hat{\Pi}_{\mathcal{U}}$ is $b_0$-correct. Furthermore, we assume for sufficiently large constant $C_3$, $\theta^* \leq \frac{1}{C_3}$, $\theta^* \leq \min_{k \in [K]} C_3 \|\theta_{\mathcal{L}}^{(k)}\|_1$, and for a constant $r_0 > 0$, $\min_{k \neq \ell} \{P_{k\ell}\} \geq r_0$. Then, there is a constant $\tilde{c}_2 = \tilde{c}_2(K, C_1, C_2, C_3, c_3, r_0) > 0$ such that $[-\log(\tilde{c}_2 \text{Risk}(\hat{y}))] / [-\log(\inf_{\tilde{y}} \{\text{Risk}(\tilde{y})\})] \geq \tilde{c}_2$.*

*Proof.* On one hand, for any $k, k^* \in [K], k \neq k^*$, using exactly the same proof as in Section J.1, we can show that when $C_3 > C_1$,

$$
\inf_{\tilde{y}} (\mathbb{P}(\hat{y} = k | \pi^* = e_{k^*}) + \mathbb{P}(\hat{y} = k^* | \pi^* = k))
$$

$$
\geq \frac{1}{2} \exp\left( 2 \sum_{i=1}^{n} -\frac{1}{2(1 - (\theta^a)^2)^{\frac{3}{2}}} \left( \sqrt{\theta^* \theta_i P_{kk_i}} - \sqrt{\theta^* \theta_i P_{k^* k_i}} \right)^2 \right)
\tag{111}
$$

where $k_i$ is the true label of node $i$, $\theta^a = \max_{i \in [n]} \max_{k \in [K]} \sqrt{\theta^* \theta_i P_{kk_i}}$. According to DCBM model and condition (8), $\theta^a \leq \frac{C}{C_3}$. Hence take $C_3 \geq \sqrt{2} C_1$, then

$$
\inf_{\tilde{y}} \{\text{Risk}(\tilde{y})\} \geq \max_{k \neq k^* \in [K]} \inf_{\tilde{y}} (\mathbb{P}(\hat{y} = k | \pi^* = e_{k^*}) + \mathbb{P}(\hat{y} = k^* | \pi^* = e_k))
$$

$$\geq \max_{k \neq k^* \in [K]} \frac{1}{2} \exp\left(-2\sqrt{2} \sum_{i=1}^{n} \left(\sqrt{\theta^* \theta_i P_{k k_i}} - \sqrt{\theta^* \theta_i P_{k^* k_i}}\right)^2\right)$$

$$= \frac{1}{2} \exp\left(-2\sqrt{2}\theta^* \min_{k \neq k^* \in [K]} \sum_{l=1}^{K} \|\theta^{(l)}\|_1 \left(\sqrt{P_{kl}} - \sqrt{P_{k^* l}}\right)^2\right) \tag{112}$$

Let $B_0$ be the event that $\hat{\Pi}_{\mathcal{U}}$ is $b_0$-correct. When $\inf_{\tilde{y}}\{\text{Risk}(\tilde{y})\}$ is replaced by the version conditioning on $B_0$, since $X$ and $A$ are independent, and $B_0 \in \sigma(A)$, conditioning on $B_0$ or not does not affect the distribution of $X$. On the other hand, for any $k, k^*$, since $\pi^*$ does not affect the distribution of $A$, the distribution of $A|B_0, \pi^* = e_k$ and $A|B_0, \pi^* = e_{k^*}$ are the same, so their Hellinger distance is still 0. Hence, all the proofs in Section J and above remain unaffected. In other words, one does not gain a lot of information from $B_0$.

On the other hand, notice that proof of Theorem 2 still works conditioning on $B_0$. In other words, there exists constant $C > 0$, such that given $B_0$, for any $\delta \in (0, 1/2)$, with probability $1 - \delta$, simultaneously for $1 \leq k \leq K$,

$$|\hat{\psi}_k(\hat{\Pi}_{\mathcal{U}}) - \psi_k(\hat{\Pi}_{\mathcal{U}})| \leq C \left(\sqrt{\frac{\log(1/\delta)}{\|\theta\|_1 \cdot \min\{\theta^*, \|\theta_{\mathcal{L}}^{(k)}\|_1\}}} + \frac{\|\theta_{\mathcal{L}}^{(k)}\|^2}{\|\theta_{\mathcal{L}}^{(k)}\|_1 \|\theta\|_1}\right).$$

Define $\tilde{\beta}_k = \psi_k(\hat{\Pi}_{\mathcal{U}})$. Replacing $c_0 \beta_n$ by $\tilde{\beta}_k$ and replicating the proof of Corollary 1, we can show that

$$\mathbb{P}(\hat{y} \neq k^* | B_0, \pi^* = e_{k^*}) \leq \bar{C} \sum_{k=1}^{K} \exp\left(-\frac{C_0}{c_0^2} \tilde{\beta}_k^2 \|\theta\|_1 \cdot \min\{\theta^*, \|\theta_{\mathcal{L}}^{(k)}\|_1\}\right)$$

Since $\theta^* \leq \min_{k \in [K]} C_3 \|\theta_{\mathcal{L}}^{(k)}\|_1$, $\min\{\theta^*, \|\theta_{\mathcal{L}}^{(k)}\|_1\} \geq \min\{1, \frac{1}{C_3}\}\theta^*$, therefore,

$$\mathbb{P}(\hat{y} \neq k^* | B_0, \pi^* = e_{k^*}) \leq \bar{C} \sum_{k=1}^{K} \exp\left(-\frac{C_0}{c_0^2} \min\{1, \frac{1}{C_3}\} \tilde{\beta}_k^2 \theta^* \|\theta\|_1\right) \tag{113}$$

Recall that in (20) of Section G, we show that

$$\tilde{\beta}_k = \psi_k(\hat{\Pi}_{\mathcal{U}}) \geq \sqrt{(e_k - e_{k^*})' \tilde{M}(e_k - e_{k^*})} \tag{114}$$

By Lemma 6, denote

$$C_5 = 8K^2 \sqrt{K} C_2^2 b_0 \frac{\|\theta_{\mathcal{U}}\|_1}{\|\theta\|_1} \tag{115}$$

Suppose that $C_5 \leq \frac{1}{4}$. Then, for any vector $\alpha \in \mathbb{R}^k$,

$$|\alpha' \tilde{M} \alpha| \geq \frac{1 - 3C_5}{1 - C_5} |\alpha' \tilde{M}^{(0)} \alpha| \geq \frac{1}{3} |\alpha' \tilde{M}^{(0)} \alpha| \tag{116}$$

where recall that in Section G, we define

$$M^{(0)} = P\left(G_{\mathcal{L}\mathcal{L}}^2 + G_{\mathcal{U}\mathcal{U}}^2\right) P$$

$D_{M^{(0)}} = \text{diag}(M_{11}^{(0)}, ..., M_{KK}^{(0)})$, and $\tilde{M}, \tilde{M}^{(0)} = D_{M^{(0)}}^{-\frac{1}{2}} M^{(0)} D_{M^{(0)}}^{-\frac{1}{2}}$

Hence, when $b_0$ is sufficiently small,

$$\tilde{\beta}_k^2 \geq (e_k - e_{k^*})' \tilde{M}(e_k - e_{k^*}) \geq \frac{1}{3} |(e_k - e_{k^*})' \tilde{M}^{(0)}(e_k - e_{k^*})|$$

$$= \frac{2}{3} \left(1 - \frac{M_{kk^*}^{(0)}}{\sqrt{M_{kk}^{(0)}} \sqrt{M_{k^* k^*}^{(0)}}}\right) \tag{117}$$

Define $\lambda_l = \sqrt{\|\theta_{\mathcal{L}}^{(l)}\|_1^2 + \|\theta_{\mathcal{U}}^{(l)}\|_1^2}, l \in [K]$. Then,

$$M_{kk^*}^{(0)} = \sum_{l=1}^{K} \lambda_l^2 P_{kl} P_{lk^*} = \sum_{l=1}^{K} \lambda_l^2 P_{kl} P_{k^*l}$$

$$M_{kk}^{(0)} = \sum_{l=1}^{K} \lambda_l^2 P_{kl} P_{lk} = \sum_{l=1}^{K} \lambda_l^2 P_{kl}^2$$

$$M_{k^*k^*}^{(0)} = \sum_{l=1}^{K} \lambda_l^2 P_{k^*l} P_{lk^*} = \sum_{l=1}^{K} \lambda_l^2 P_{k^*l}^2$$

Therefore, plugging the above result into (117), we have

$$
\begin{aligned}
\tilde{\beta}_k^2 &\geq \frac{2}{3} \left( 1 - \frac{M_{kk^*}^{(0)}}{\sqrt{M_{kk}^{(0)}} \sqrt{M_{k^*k^*}^{(0)}}} \right) \\
&= \frac{2}{3} \left( 1 - \frac{\sum_{l=1}^{K} \lambda_l^2 P_{kl} P_{k^*l}}{\sqrt{\sum_{l=1}^{K} \lambda_l^2 P_{kl}^2} \sqrt{\sum_{l=1}^{K} \lambda_l^2 P_{k^*l}^2}} \right) \\
&= \frac{2}{3} \left( 1 - \sqrt{1 - \frac{(\sum_{l=1}^{K} \lambda_l^2 P_{kl}^2)(\sum_{l=1}^{K} \lambda_l^2 P_{k^*l}^2) - (\sum_{l=1}^{K} \lambda_l^2 P_{kl} P_{k^*l})^2}{(\sum_{l=1}^{K} \lambda_l^2 P_{kl}^2)(\sum_{l=1}^{K} \lambda_l^2 P_{k^*l}^2)}} \right) \\
&\geq \frac{1}{3} \frac{(\sum_{l=1}^{K} \lambda_l^2 P_{kl}^2)(\sum_{l=1}^{K} \lambda_l^2 P_{k^*l}^2) - (\sum_{l=1}^{K} \lambda_l^2 P_{kl} P_{k^*l})^2}{(\sum_{l=1}^{K} \lambda_l^2 P_{kl}^2)(\sum_{l=1}^{K} \lambda_l^2 P_{k^*l}^2)}
\end{aligned}
\tag{118}
$$

Notice that

$$
\begin{aligned}
&(\sum_{l=1}^{K} \lambda_l^2 P_{kl}^2)(\sum_{l=1}^{K} \lambda_l^2 P_{k^*l}^2) - (\sum_{l=1}^{K} \lambda_l^2 P_{kl} P_{k^*l})^2 \\
&= \sum_{l,\tilde{l} \in [K]} \lambda_l^2 \lambda_{\tilde{l}}^2 (P_{kl} P_{k^*\tilde{l}} - P_{k\tilde{l}} P_{k^*l})^2 \\
&\geq \sum_{l \in [K]} \lambda_l^2 \lambda_k^2 (P_{kl} P_{k^*k} - P_{kk} P_{k^*l})^2 + \sum_{l \in [K]} \lambda_l^2 \lambda_{k^*}^2 (P_{kl} P_{k^*k^*} - P_{kk^*} P_{k^*l})^2 \\
(\text{Identification Condition}) &= \sum_{l \in [K]} \lambda_l^2 \left( \lambda_k^2 (P_{kk^*} P_{kl} - P_{k^*l})^2 + \lambda_{k^*}^2 (P_{kl} - P_{kk^*} P_{k^*l})^2 \right) \\
(\text{Cauchy-Schwartz}) &\geq \sum_{l \in [K]} \lambda_l^2 \frac{1}{\frac{1}{\lambda_k^2} + \frac{1}{\lambda_{k^*}^2}} \left( P_{kk^*} P_{kl} - P_{k^*l} + P_{kl} - P_{kk^*} P_{k^*l} \right)^2 \\
&= \sum_{l \in [K]} \lambda_l^2 \frac{1}{\frac{1}{\lambda_k^2} + \frac{1}{\lambda_{k^*}^2}} (1 + P_{kk^*})^2 (P_{kl} - P_{k^*l})^2 \\
&\geq \frac{1}{\frac{1}{\min_{l \in [K]} \lambda_l^2} + \frac{1}{\min_{l \in [K]} \lambda_l^2}} \sum_{l \in [K]} \lambda_l^2 (P_{kl} - P_{k^*l})^2 \\
&= \frac{1}{2} \min_{l \in [K]} \lambda_l^2 \sum_{l \in [K]} \lambda_l^2 (P_{kl} - P_{k^*l})^2
\end{aligned}
\tag{119}
$$

Substituting (119) into (118), we have

$$\tilde{\beta}_k^2 \geq \frac{1}{6} \frac{\min_{l \in [K]} \lambda_l^2 \sum_{l \in [K]} \lambda_l^2 (P_{kl} - P_{k^*l})^2}{(\sum_{l=1}^{K} \lambda_l^2 P_{kl}^2)(\sum_{l=1}^{K} \lambda_l^2 P_{k^*l}^2)}$$

$$\text{(By Condition (8))} \geq \frac{1}{6C_1^4} \frac{\min_{l\in[K]} \lambda_l^2 \sum_{l\in[K]} \lambda_l^2 (P_{kl} - P_{k^*l})^2}{(\sum_{l=1}^K \lambda_l^2)^2} \tag{120}$$

On one hand, by Cauchy-Schwartz inequality,

$$\begin{aligned}
\lambda_l^2 &= \|\theta_{\mathcal{L}}^{(l)}\|_1^2 + \|\theta_{\mathcal{U}}^{(l)}\|_1^2 \\
&\geq \frac{1}{2}(\|\theta_{\mathcal{L}}^{(l)}\|_1 + \|\theta_{\mathcal{U}}^{(l)}\|_1)^2 \\
&= \frac{1}{2}\|\theta^{(l)}\|_1^2
\end{aligned} \tag{121}$$

So

$$\begin{aligned}
\min_{l\in[K]} \lambda_l^2 &\geq \frac{1}{2}(\min_{l\in[K]} \|\theta^{(l)}\|_1)^2 \\
\text{(By Condition (9))} &\geq \frac{1}{2}(\frac{1}{C_2} \max_{l\in[K]} \|\theta^{(l)}\|_1)^2 \\
&\geq \frac{1}{2}(\frac{1}{KC_2} \|\theta\|_1)^2 \\
&= \frac{1}{2K^2C_2^2}\|\theta\|_1^2
\end{aligned} \tag{122}$$

On the other hand,

$$\begin{aligned}
\sum_{l=1}^K \lambda_l^2 &= \sum_{l=1}^K (\|\theta_{\mathcal{L}}^{(l)}\|_1^2 + \|\theta_{\mathcal{U}}^{(l)}\|_1^2) \\
&\leq (\sum_{l=1}^K \|\theta_{\mathcal{L}}^{(l)}\|_1 + \sum_{l=1}^K \|\theta_{\mathcal{U}}^{(l)}\|_1)^2 \\
&= \|\theta\|_1^2
\end{aligned} \tag{123}$$

Plugging (121), (122), (123) into (120), we obtain

$$\begin{aligned}
\tilde{\beta}_k^2 &\geq \frac{1}{6C_1^4} \frac{\min_{l\in[K]} \lambda_l^2 \sum_{l\in[K]} \lambda_l^2 (P_{kl} - P_{k^*l})^2}{(\sum_{l=1}^K \lambda_l^2)^2} \\
&\geq \frac{1}{6C_1^4} \frac{\frac{1}{2K^2C_2^2}\|\theta\|_1^2 \sum_{l\in[K]} \frac{1}{2}\|\theta^{(l)}\|_1^2 (P_{kl} - P_{k^*l})^2}{(\|\theta\|_1^2)^2} \\
&= \frac{1}{24K^2C_1^4C_2^2} \frac{\sum_{l\in[K]} \|\theta^{(l)}\|_1^2 (P_{kl} - P_{k^*l})^2}{\|\theta\|_1^2} \\
&\geq \frac{1}{24K^2C_1^4C_2^2} \frac{\min_{l\in[K]} \|\theta^{(l)}\|_1 \sum_{l\in[K]} \|\theta^{(l)}\|_1 (\sqrt{P_{kl}} - \sqrt{P_{k^*l}})^2}{\|\theta\|_1^2 (\sqrt{P_{kl}} + \sqrt{P_{k^*l}})} \\
\text{(By (9)}, \min_{k\neq\ell}\{P_{k\ell}\} \geq r_0) &\geq \frac{1}{24K^2C_1^4C_2^2} \frac{\frac{1}{C_2}\max_{l\in[K]} \|\theta^{(l)}\|_1 \sum_{l\in[K]} \|\theta^{(l)}\|_1 (\sqrt{P_{kl}} - \sqrt{P_{k^*l}})^2}{\|\theta\|_1^2 2\sqrt{r_0}} \\
&\geq \frac{1}{24K^2C_1^4C_2^2} \frac{\frac{1}{KC_2}\|\theta\|_1 \sum_{l\in[K]} \|\theta^{(l)}\|_1 (\sqrt{P_{kl}} - \sqrt{P_{k^*l}})^2}{\|\theta\|_1^2 2\sqrt{r_0}} \\
&= \frac{1}{48\sqrt{r_0}K^3C_1^4C_2^3} \frac{\sum_{l\in[K]} \|\theta^{(l)}\|_1 (\sqrt{P_{kl}} - \sqrt{P_{k^*l}})^2}{\|\theta\|_1}
\end{aligned} \tag{124}$$

Substituting (124) into (113), we obtain

$$\mathbb{P}(\hat{y} \neq k^* | B_0, \pi^* = e_{k^*}) \leq \bar{C} \sum_{k=1}^K \exp\left(-\frac{C_0}{c_0^2} \min\{1, \frac{1}{C_3}\}\tilde{\beta}_k^2 \theta^* \|\theta\|_1\right)$$

$$\leq \bar{C} \sum_{k=1}^{K} \exp\Big(-C_8\theta^* \sum_{l=1}^{K} \|\theta^{(l)}\|_1 (\sqrt{P_{kl}} - \sqrt{P_{k^*l}})^2\Big) \tag{125}$$

where $C_8 = \frac{C_0}{c_0^2}\min\{1, \frac{1}{C_3}\}\frac{1}{48\sqrt{r_0}K^3C_1^4C_2^3} = \min\{1, \frac{1}{C_3}\}\frac{C_0}{48\sqrt{r_0}K^3c_0^2C_1^4C_2^3}$.

As a result,

$$\frac{1}{\bar{C}K^2}\mathrm{Risk}(\hat{y}|B_0) = \frac{1}{\bar{C}K^2}\sum_{k*=1}^{K}\mathbb{P}(\hat{y}\neq k^*|B_0, \pi^* = e_{k^*})$$

$$\leq \frac{1}{\bar{C}K^2}\sum_{k*=1}^{K}\bar{C}\sum_{k=1}^{K}\exp\Big(-C_8\theta^*\sum_{l=1}^{K}\|\theta^{(l)}\|_1(\sqrt{P_{kl}} - \sqrt{P_{k^*l}})^2\Big)$$

$$= \frac{1}{K^2}\sum_{k*=1}^{K}\sum_{k=1}^{K}\exp\Big(-C_8\theta^*\sum_{l=1}^{K}\|\theta^{(l)}\|_1(\sqrt{P_{kl}} - \sqrt{P_{k^*l}})^2\Big)$$

$$\leq \max_{k\neq k^*\in[K]}\exp\Big(-C_8\theta^*\sum_{l=1}^{K}\|\theta^{(l)}\|_1(\sqrt{P_{kl}} - \sqrt{P_{k^*l}})^2\Big)$$

$$= \exp\Big(-C_8\theta^*\min_{k\neq k^*\in[K]}\sum_{l=1}^{K}\|\theta^{(l)}\|_1(\sqrt{P_{kl}} - \sqrt{P_{k^*l}})^2\Big) \tag{126}$$

Comparing (112) and (126), we have

$$\frac{-\log(\frac{1}{2\bar{C}K^2}\mathrm{Risk}(\hat{y}|B_0))}{-\log(\inf_{\tilde{y}}\{\mathrm{Risk}(\tilde{y})\})} \geq \frac{\log(2) + C_8\mathcal{I}}{\log(2) + 2\sqrt{2}\mathcal{I}} \tag{127}$$

where $\mathcal{I} = \theta^*\min_{k\neq k^*\in[K]}\sum_{l=1}^{K}\|\theta^{(l)}\|_1(\sqrt{P_{kl}} - \sqrt{P_{k^*l}})^2$ denotes the efficient information in the data.

Notice that since $\mathcal{I} \geq 0$, when $C_8 \geq 2\sqrt{2}$, $\frac{\log(2)+C_8\mathcal{I}}{\log(2)+2\sqrt{2}\mathcal{I}} \geq 1$; when $C_8 \leq 2\sqrt{2}$, $\frac{\log(2)+C_8\mathcal{I}}{\log(2)+2\sqrt{2}\mathcal{I}} \geq \frac{\log(2)\frac{C_8}{2\sqrt{2}}+C_8\mathcal{I}}{\log(2)+2\sqrt{2}\mathcal{I}} = \frac{C_8}{2\sqrt{2}}$. Therefore,

$$\frac{-\log(\frac{1}{2\bar{C}K^2}\mathrm{Risk}(\hat{y}|B_0))}{-\log(\inf_{\tilde{y}}\{\mathrm{Risk}(\tilde{y})\})} \geq \frac{\log(2) + C_8\mathcal{I}}{\log(2) + 2\sqrt{2}\mathcal{I}} \geq \min\{1, \frac{C_8}{2\sqrt{2}}\} \tag{128}$$

Take $\tilde{c}_2 = \min\{1, \frac{C_8}{2\sqrt{2}}, \frac{1}{2\bar{C}K^2}\}$. Then $\tilde{c}_2$ only depends on $K, C_1, C_2, C_3, c_3, r_0$ (recall that both $C_0$ and $c_0$ only depend on $K, C_1, C_2, C_3, c_3, r_0$, and $\bar{C} = 2$), and

$$\frac{-\log(\tilde{c}_2\mathrm{Risk}(\hat{y}|B_0))}{-\log(\inf_{\tilde{y}}\{\mathrm{Risk}(\tilde{y})\})} \geq \frac{-\log(\frac{1}{2\bar{C}K^2}\mathrm{Risk}(\hat{y}|B_0))}{-\log(\inf_{\tilde{y}}\{\mathrm{Risk}(\tilde{y})\})} \geq \min\{1, \frac{C_8}{2\sqrt{2}}\} \geq \tilde{c}_2 \tag{129}$$

This concludes our proof. $\square$

## L   PROOF OF THEOREM 4, 5

### L.1   PROOF OF THEOREM 4

**Theorem 4.** *Consider the DCBM model where (8)-(9) hold. We apply SCORE+ to obtain $\hat{\Pi}_{\mathcal{U}\setminus\{i\}}$ and plug it into the above algorithm. As $n \to \infty$, suppose for some constant $q_0 > 0$, $\min_{i\in\mathcal{U}}\theta_i \geq q_0\max_{i\in\mathcal{U}}\theta_i$, $\beta_n\|\theta_{\mathcal{U}}\| \geq q_0\sqrt{\log(n)}$, $\beta_n^2\|\theta\|_1\min_{i\in\mathcal{U}}\theta_i \to \infty$, and $\beta_n^2\|\theta\|_1\min_k\{\|\theta_{\mathcal{L}}^{(k)}\|_1\} \to \infty$. Then, $\frac{1}{|\mathcal{U}|}\sum_{i\in\mathcal{U}}\mathbb{P}(\hat{y}_i \neq k_i) \to 0$, so the in-sample classification algorithm in section 3 is consistent.*

*Proof.* Let $i^* = arg\max_{i \in \mathcal{U}} \mathbb{P}(\hat{y}_i \neq k_i)$. Then

$$\frac{1}{|\mathcal{U}|} \sum_{i \in \mathcal{U}} \mathbb{P}(\hat{y}_i \neq k_i) \leq \mathbb{P}(\hat{y}_{i^*} \neq k_{i^*})$$

Notice that the assumptions of theorem 4 directly imply the assumptions of Corollary 2 when taking $i^*$ as the new node. Hence, regard $i^*$ as the new node and leveraging on Corollary 2, we have

$$\mathbb{P}(\hat{y}_{i^*} \neq k_{i^*}) \to 0$$

Therefore,

$$\frac{1}{|\mathcal{U}|} \sum_{i \in \mathcal{U}} \mathbb{P}(\hat{y}_i \neq k_i) \to 0$$

In other words, the in-sample classification algorithm in section 3 is consistent. $\square$

### L.2  PROOF OF THEOREM 5

**Theorem 5.** *Suppose the conditions of Corollary 1 hold, where $b_0$ is properly small , and suppose that $\hat{\Pi}_{\mathcal{U} \setminus \{i\}}$ is $b_0$-correct for all $i \in \mathcal{U}$. Furthermore, we assume for sufficiently large constant $C_3$, $\max_{i \in \mathcal{U}} \theta_i \leq \frac{1}{C_3}$, $\max_{i \in \mathcal{U}} \theta_i \leq \min_{k \in [K]} C_3 \|\theta_{\mathcal{L}}^{(k)}\|_1$, $\log(|\mathcal{U}|) \leq C_3 \beta_n^2 \|\theta\|_1 \min_{i \in \mathcal{U}} \theta_i$, and for a constant $r_0 > 0$, $\min_{k \neq \ell} \{P_{k\ell}\} \geq r_0$. Then, there is a constant $\tilde{c}_{21} = \tilde{c}_{21}(K, C_1, C_2, C_3, c_3, r_0) > 0$ such that $[-\log(\tilde{c}_{21} \mathrm{Risk}_{ins}(\hat{y}))]/[-\log(\inf_{\tilde{y}} \{\mathrm{Risk}_{ins}(\tilde{y})\})] \geq \tilde{c}_{21}$, so the in-sample classification algorithm in section 3 is efficient.*

*Proof.* For $i \in \mathcal{U}$, define individual risk $\mathrm{Risk}(\tilde{y}_i) = \sum_{k^* \in [K]} \mathbb{P}(\tilde{y}_i \neq k^* | \pi_i = e_{k^*})$. Then the in-sample risk $\mathrm{Risk}_{ins}(\tilde{y}) = \frac{1}{|\mathcal{U}|} \sum_{i \in \mathcal{U}} \mathrm{Risk}(\hat{y}_i)$.

The minimizer of $\mathrm{Risk}_{ins}(\tilde{y})$ may not exist, so we define $\tilde{y}^{(0)}$ to be an approximate minimizer such that $\mathrm{Risk}_{ins}(\tilde{y}^{(0)}) \leq 2 \inf_{\tilde{y}} \{\mathrm{Risk}_{ins}(\tilde{y})\}$. By the definition of infimum, such $\tilde{y}^{(0)}$ always exists as long as $\inf_{\tilde{y}} \{\mathrm{Risk}_{ins}(\tilde{y})\} > 0$. Notice that for any $i \in \mathcal{U}$, $\inf_{\tilde{y}} \{\mathrm{Risk}_{ins}(\tilde{y})\} \geq \inf_{\tilde{y}_i} \frac{1}{|\mathcal{U}|} \{\mathrm{Risk}(\tilde{y}_i)\}$. Regarding node $i$ as the new node and leveraging on (112), we know that $\inf_{\tilde{y}_i} \{\mathrm{Risk}(\tilde{y}_i)\} > 0$. Hence, $\inf_{\tilde{y}} \{\mathrm{Risk}_{ins}(\tilde{y})\} \geq \inf_{\tilde{y}_i} \frac{1}{|\mathcal{U}|} \{\mathrm{Risk}(\tilde{y}_i)\} > 0$ (note that we are not taking $n$ or $|\mathcal{U}| \to \infty$ here), and $\tilde{y}^{(0)}$ is well-defined.

Let $i^* = arg\max_{i \in \mathcal{U}} \mathrm{Risk}(\hat{y}_i)$, and let $k^*$ be the true label of $i^*$. Regard $[n] \setminus \{i^*\}$ as the existing nodes in the network and $i^*$ as the new node. By (112) and (126), we have

$$-\log(\frac{1}{2\bar{C}K^2} \mathrm{Risk}(\hat{y}_{i^*})) \geq \log(2) + C_8 \mathcal{I}_{i^*} \tag{130}$$

$$-\log(\{\mathrm{Risk}(\tilde{y}_{i^*}^{(0)})\}) \leq -\inf_{\tilde{y}_{i^*}} \log(\{\mathrm{Risk}(\tilde{y}_{i^*})\}) \leq \log(2) + 2\sqrt{2}\mathcal{I}_{i^*} \tag{131}$$

where $\mathcal{I}_{i^*} = \theta_{i^*} \min_{k \neq k^* \in [K]} (\sum_{l=1}^{K} \|\theta^{(l)}\|_1 (\sqrt{P_{kl}} - \sqrt{P_{k^*l}})^2 - \theta_{i^*}(1 - \sqrt{P_{kk^*}})^2)$ denotes the efficient information in the data for classifying node $i^*$.

As a result, we have

$$\frac{-\log(\frac{1}{4\bar{C}K^2} \mathrm{Risk}_{ins}(\hat{y}))}{-\log(\inf_{\tilde{y}} \{\mathrm{Risk}_{ins}(\tilde{y})\})} \geq \frac{-\log(\frac{1}{4\bar{C}K^2} \mathrm{Risk}_{ins}(\hat{y}))}{-\log(\frac{1}{2} \mathrm{Risk}_{ins}(\tilde{y}^{(0)}))}$$

$$= \frac{-\log(\frac{1}{4\bar{C}K^2} \frac{1}{|\mathcal{U}|} \sum_{i \in \mathcal{U}} \mathrm{Risk}(\hat{y}_i))}{-\log(\frac{1}{2|\mathcal{U}|} \sum_{i \in \mathcal{U}} \mathrm{Risk}(\tilde{y})_i^{(0)})}$$

$$\geq \frac{-\log(\frac{1}{4\widehat{C}K^2}\text{Risk}(\hat{y}_{i^*}))}{-\log(\frac{1}{|2\mathcal{U}|}\text{Risk}(\tilde{y})_{i^*}^{(0)})}$$

$$(By\ (130),(131)) \geq \frac{\log(4)+C_8\mathcal{I}_{i^*}}{\log(4)+2\sqrt{2}\mathcal{I}_{i^*}+\log(|\mathcal{U}|)} \tag{132}$$

Notice that

$$\mathcal{I}_{i^*}$$

$$= \theta_{i^*} \min_{k\neq k^*\in[K]} (\sum_{l=1}^{K}\|\theta^{(l)}\|_1(\sqrt{P_{kl}}-\sqrt{P_{k^*l}})^2 - \theta_{i^*}(1-\sqrt{P_{kk^*}})^2)$$

$$\geq \theta_{i^*} \min_{k\neq k^*\in[K]} (\|\theta^{(k)}\|_1(\sqrt{P_{kk}}-\sqrt{P_{k^*k}})^2 + \|\theta^{(k^*)}\|_1(\sqrt{P_{kk^*}}-\sqrt{P_{k^*k^*}})^2 - \theta_{i^*}(1-\sqrt{P_{kk^*}})^2)$$

(By identification condition that $P_{kk}=P_{k^*k^*}=1$)

$$= \theta_{i^*} \min_{k\neq k^*\in[K]} ((\|\theta^{(k)}\|_1 + \|\theta^{(k^*)}\|_1 - \theta_{i^*})(1-\sqrt{P_{k^*k}})^2)$$

(The true label of node $i^*$ is $k^*$)

$$\geq \theta_{i^*} \min_{k\neq k^*\in[K]} (\|\theta^{(k)}\|_1(1-\sqrt{P_{k^*k}})^2) \tag{133}$$

By assumption 8, for any $k\in[K]$,

$$(1-\sqrt{P_{k^*k}})^2 = \frac{(1-P_{k^*k})^2}{(1+\sqrt{P_{k^*k}})^2}$$

$$= \frac{(2-2P_{k^*k})^2}{4(1+\sqrt{P_{k^*k}})^2}$$

(By identification condition that $P_{kk}=P_{k^*k^*}=1$)

$$= \frac{(P_{kk}+Pk^*k^*-2P_{k^*k})^2}{4(1+\sqrt{P_{k^*k}})^2}$$

$$= \frac{((e_k-e_{k^*})'P(e_k-e_{k^*}))^2}{4(1+\sqrt{P_{k^*k}})^2}$$

$$\geq \frac{(|\lambda_{min}(P)|\|e_k-e_{k^*}\|^2)^2}{4(1+\sqrt{C_1})^2}$$

$$\geq \frac{\beta_n^2}{(1+\sqrt{C_1})^2} \tag{134}$$

By assumption 9, for any $k\in[K]$

$$\|\theta^{(k)}\|_1 \geq \min_{k\neq k^*\in[K]} \|\theta^{(k)}\|_1$$

$$\geq \frac{1}{C_2} \max_{k\neq k^*\in[K]} \|\theta^{(k)}\|_1$$

$$\geq \frac{1}{KC_2} \sum_{k\neq k^*\in[K]} \|\theta^{(k)}\|_1$$

$$= \frac{1}{KC_2}\|\theta\|_1 \tag{135}$$

From the assumption, $\log(|\mathcal{U}|) \leq C_3\beta_n^2\|\theta\|_1\min_{i\in\mathcal{U}}\theta_i$. Plugging (134) (135) into (133), we obtain

$$\mathcal{I}_{i^*} \geq \theta_{i^*} \min_{k\neq k^*\in[K]} (\frac{1}{KC_2(1+\sqrt{C_1})^2}\beta_n^2\|\theta\|_1)$$

$$\geq \frac{1}{KC_2(1+\sqrt{C_1})^2}\beta_n^2\|\theta\|_1 \min_{i\in\mathcal{U}}\theta_i$$

$$\geq \frac{1}{KC_2C_3(1+\sqrt{C_1})^2}log(|\mathcal{U}|) \tag{136}$$

In other words,

$$log(|\mathcal{U}|) \leq KC_2C_3(1+\sqrt{C_1})^2\mathcal{I}_{i^*} \tag{137}$$

Substituting (137) into (132), we have

$$\frac{-\log(\frac{1}{4\bar{C}K^2}\mathrm{Risk}_{ins}(\hat{y}))}{-\log(\inf_{\tilde{y}}\{\mathrm{Risk}_{ins}(\tilde{y})\})} \geq \frac{\log(4)+C_8\mathcal{I}_{i^*}}{\log(4)+(2\sqrt{2}+KC_2C_3(1+\sqrt{C_1})^2)\mathcal{I}_{i^*}} \tag{138}$$

Similar to the proof of Theorem 3, notice that since $\mathcal{I}_{i^*} \geq 0$, when $C_8 \geq 2\sqrt{2}+KC_2C_3(1+\sqrt{C_1})^2$,

$$\frac{\log(4)+C_8\mathcal{I}_{i^*}}{\log(4)+(2\sqrt{2}+KC_2C_3(1+\sqrt{C_1})^2)\mathcal{I}_{i^*}} \geq 1 \tag{139}$$

; when $C_8 \leq 2\sqrt{2}+KC_2C_3(1+\sqrt{C_1})^2$,

$$\frac{\log(4)+C_8\mathcal{I}_{i^*}}{\log(4)+(2\sqrt{2}+KC_2C_3(1+\sqrt{C_1})^2)\mathcal{I}_{i^*}} \geq \frac{\log(4)\frac{C_8}{2\sqrt{2}+KC_2C_3(1+\sqrt{C_1})^2}+C_8\mathcal{I}_{i^*}}{\log(4)+(2\sqrt{2}+KC_2C_3(1+\sqrt{C_1})^2)\mathcal{I}_{i^*}}$$
$$= \frac{C_8}{2\sqrt{2}+KC_2C_3(1+\sqrt{C_1})^2} \tag{140}$$

Therefore,

$$\frac{-\log(\frac{1}{4\bar{C}K^2}\mathrm{Risk}_{ins}(\hat{y}))}{-\log(\inf_{\tilde{y}}\{\mathrm{Risk}_{ins}(\tilde{y})\})} \geq \frac{\log(4)+C_8\mathcal{I}_{i^*}}{\log(4)+(2\sqrt{2}+KC_2C_3(1+\sqrt{C_1})^2)\mathcal{I}_{i^*}}$$
$$\geq \min\{1, \frac{C_8}{2\sqrt{2}+KC_2C_3(1+\sqrt{C_1})^2}\} \tag{141}$$

Take $\tilde{c}_{21} = \min\{1, \frac{C_8}{2\sqrt{2}+KC_2C_3(1+\sqrt{C_1})^2}, \frac{1}{4\bar{C}K^2}\}$. Then $\tilde{c}_{21}$ only depends on $K, C_1, C_2, C_3, c_3, r_0$ (recall that both $C_0$ and $c_0$ only depend on $K, C_1, C_2, C_3, c_3, r_0$, and $\bar{C} = 2$), and

$$\frac{-\log(\tilde{c}_{21}\mathrm{Risk}_{ins}(\hat{y}))}{-\log(\inf_{\tilde{y}}\{\mathrm{Risk}_{ins}(\tilde{y})\})} \geq \frac{-\log(\frac{1}{4\bar{C}K^2}\mathrm{Risk}_{ins}(\hat{y}))}{-\log(\inf_{\tilde{y}}\{\mathrm{Risk}_{ins}(\tilde{y})\})}$$
$$\geq \min\{1, \frac{C_8}{2\sqrt{2}+KC_2C_3(1+\sqrt{C_1})^2}\}$$
$$\geq \tilde{c}_{21} \tag{142}$$

This concludes our proof. $\qquad\square$

