# OpenReview forum: "Semi-supervised Community Detection via Structural Similarity Metrics"
_ICLR.cc/2023/Conference — ICLR 2023 poster_

### Official Review · Reviewer_Rg2A · 2022-10-24

**Confidence:** 3
**Correctness:** 4
**Technical Novelty And Significance:** 3
**Empirical Novelty And Significance:** 3
**Recommendation:** 8

**Clarity, Quality, Novelty And Reproducibility:**

This paper is well-written and easy to follow. It provides solid theoretical results and convincing experimental results. I think all results are original and reproducible.

**Strength And Weaknesses:**

Strengths:
1. This paper considers a new semi-supervised community detection problem, which can be used in many applications with partially observed labeled data.
2. This paper provides an explicit misclassification error bound for the proposed algorithm.

Weaknesses:
(Minors): The label for equations (8),(9), and (10) seems unmatched with the lemmas and theorems below.

**Summary Of The Paper:**

This paper considers the semi-supervised community detection problem. In particular, they consider the degree-corrected block model (DCBM) where each node belongs to one of K communities and has a degree parameter. Then, the edges are generated between nodes according to the degree parameter of the two nodes and community connection probability.  The problem is given the label of a subset of nodes, to find the label of the remaining nodes. They propose two new algorithms Anglemin and Anglemin+ for this problem. They provide a misclassification error bound for their algorithm and show their algorithm is nearly optimal by comparing it with the information-theoretical lower bound. Finally, their experimental results show their algorithms achieve smaller classification errors than the unsupervised method.

**Summary Of The Review:**

This paper provides a solid contribution to a semi-supervised community detection problem, They provide theoretical guarantees and convincing experimental results. Thus, I tend to accept this paper.

---

> ### Author Response · Authors · 2022-11-18
> **Response to Reviewer Rg2A**
>
> Thank you for the nice comments. We are very glad that you appreciate the practical value and theoretical value of our results and think our contribution is "solid" and "original".
>
> **Comment**:  "The label for equations (8),(9), and (10) seems unmatched with the lemmas and theorems below.”
>
> **Response**: Thank you for catching this typo. It was caused by a cross-referencing issue when we added/deleted equations in editing the latex file. In this revision, we have fixed this problem.

---

### Official Review · Reviewer_1PMk · 2022-10-25

**Confidence:** 4
**Correctness:** 3
**Technical Novelty And Significance:** 2
**Empirical Novelty And Significance:** 2
**Recommendation:** 6

**Clarity, Quality, Novelty And Reproducibility:**

This paper is reader friendly. I believe the empirical result is easy to reproduce.

**Strength And Weaknesses:**

Strength

This paper is easy to read. I can understand the algorithm and theorems without any trouble.

Weaknesses

1. The problem can be easily solved by existing algorithms with slight modification. For example, we can run the community detection algorithm again on the whole network. If we are only allowed to apply local refinement on the network to reduce computational complexity, the following steps will also work.

(i) Apply prototypical refinement, e.g., Algorithm 2 of Gao et al. (2018), with known labels, then we can assign labels to nodes in $\mathcal U$.

(ii) Now we have labels for all nodes. we update the labels for every node by applying the same refinement procedure with initial labels for all node.

Note that the reason we update the labels in step (ii) is to use the information in $A_{\mathcal U\mathcal U}$. The method in Gao et al. (2018) may not be applied to non-assortative block models, but it is not hard to replace their procedure by a local likelihood ratio classifier.

2. The authors propose a method based on the cosine distance. This classifier may be equivalent to likelihood ratio classifier. If this is the case, it should achieve the information theoretical lower bound, but the authors fail to show this in Lemma 2. It is claimed in the paper that "we bound the total variation distance by the Hellinger distance (the total variation distance is hard to analyze directly)". It can be shown that the exponent of total variation distance and Hellinger distance are the same up to a $1+o(1)$ factor. See [1] for details. If the connectivity matrix is more general, you can use Chernoff information to bound the total variation distance. See Section 11 of [2] for more details.

[1] Zhang, Anderson Y., and Harrison H. Zhou. "Minimax rates of community detection in stochastic block models." The Annals of Statistics 44.5 (2016): 2252-2280.

[2] Thomas, M. T. C. A. J., and A. Thomas Joy. Elements of information theory. Wiley-Interscience, 2006.

**Summary Of The Paper:**

This paper studies the a semi-supervised community detection problem on a degree-corrected stochastic block model. Given some known labels of the nodes in the network, the authors propose a method to classified the new node based on cosine similarity with the in-sample data.

**Summary Of The Review:**

This paper is well written. However, the proposed method is too simple and the theoretical result cannot support the output of the algorithm can achieve the optimal error rate.

---

> ### Author Response · Authors · 2022-11-14
> **Response to Reviewer 1PMk (Part 1)**
>
> Thanks for your comments and for appreciating the clarity of our paper.
>
> ---
>
> #### **Comment:** "The problem can be easily solved by existing algorithms with slight modification."
>
> #### **Response:** You seemed to suggest the following:
>
> 1. When communities are assortative, one can apply the local refinement procedure in Gao et al. (2018) to obtain a semi-supervised community detection algorithm.
>
> 2. When communities are dis-assortative, a similar idea works by replacing the local refinement procedure by a local likelihood ratio classifier.
>
> These ideas actually do not work for our problem.
>
> **Why Idea 1 does not work:** We illustrate it with an example where $K=2$, $n=4m$, and each community has $m$ labeled nodes and $m$ unlabeled nodes. Let
> $$
> P=\begin{pmatrix}1&0.9\\\\0.9&1\end{pmatrix}, \qquad \theta^{\text{label}}_i=\begin{cases}0.8,& i\in \mathcal{C}_1,\\\\0.5,&i\in \mathcal{C}_2.\end{cases} \qquad \theta^{\text{unlabel}}_i=\begin{cases}0.6&i\in \mathcal{C}_1,\\\\0.7&i\in \mathcal{C}_2.\end{cases}
> $$
>
> This example satisfies all of our regularity conditions. Therefore, as $m\to\infty$, the error rate of our method converges to $0$. However, the error rate of your proposed method is at least $0.5$ (as explained below).
>
> The local refinement procedure in Gao et al. (2018) is as follows: Let $\hat{y}^0\in[K]^n$ be a vector of preliminary community labels. For each node $i$, this procedure assigns a community label $\hat{y}_i$ by
> $$
> \hat{y}_i = \mathrm{argmax}\_{k\in [K]}\left\\{ \frac{\sum\_{j: \hat{y}^0_j=k}A\_{ij}}{ |\{j: \hat{y}^0_j=k\}|} \right\\}.
> $$
>
> We first consider an "ideal" case where $\hat{y}^0$ contains *true* labels. By our model, $\mathbb{E}A_{ij}=\theta_i\theta_j$ for nodes in the same community, and $\mathbb{E}A_{ij}=0.9\theta_i\theta_j$ for nodes from distinct communities. Hence, when the new node $i$ is from community 2, for a labeled node $j\in \mathcal{C}\_1$, $\mathbb{E}A_{ij}=0.9\times \theta_i\times 0.8=0.72\theta_i$; for an unlabeled node $j\in \mathcal{C}\_1$, $\mathbb{E}A_{ij}=0.9\times \theta_i\times 0.6=0.54\theta_i$;
> for a labeled node $j\in \mathcal{C}\_2$, $\mathbb{E}A_{ij}=0.5\theta_i$; for an unlabeled node $j\in \mathcal{C}\_2$, $\mathbb{E}A_{ij}=0.7\theta_i$. Since $A_{ij}$'s are independent Bernoullis, it can be easily shown that
> $$
> \frac{\sum\_{j: \hat{y}^0_j=k}A\_{ij}}{ |\{j: \hat{y}^0_j=k\}|}\quad \overset{a.s.}{\longrightarrow}\quad \begin{cases} \frac{1}{2}(0.72\theta_i+0.54\theta_i) = 0.63 \theta_i, & k=1,\\\\ \frac{1}{2}(0.5\theta_i+0.7\theta_i) = 0.6 \theta_i, & k=2. \end{cases}
> $$
>
> Therefore, with an overwhelming probability, any new node $i$ from community 2 is incorrectly classified to community 1. This leads to an error rate of at least $0.5$.
>
> We then consider the "real" case where $\hat{y}^0$ is obtained from data. For example, you suggested to (i) apply the refinement procedure to impute community labels for those unlabeled nodes and (ii) take the observed \& imputed labels together as $\hat{y}^0$ and apply the refinement procedure again to assign a label for a new node $i$.  It can be similarly shown that with an overwhelming probability,  in Step (i),  all unlabeled nodes are classified to Community 1; and in Step (ii), a new node is always classified to Community 1, leading to an error rate of $0.5$.
>
> **Why Idea 2 does not work:** We point out that the refinement procedure in Gao et al. (2018) is *not* equivalent to a "local likelihood ratio classifier", as explicitly stated in the paragraph between their Equation (10) and Equation (11).  Gao et al. (2018) even argued that it is impossible to compute the likelihood ratio tests in practice, because of the large number of unknown degree parameters $\theta_i$. Their refinement procedure is specially designed for assortative communities.
>
> Meanwhile, they did mention an extension to dis-assortative communities, by replacing "argmax" by "argmin" in the definition of $\hat{y}_i$ above. However, this modification requires to know *in priori* whether communities are assortative or dis-assortative. In most application scenarios, we do not know which case it is. Furthermore, even for dis-assortative communities, we can similarly construct an example as above such that the error rate of our method converges to $0$ but the error rate of local refinement converges to $0.5$ (see more explanations below).
>
> The local refinement procedure belongs to the class of "majority vote" procedures. For such methods to work, besides assortativity, it also requires that the average of $\theta_i$'s are nearly the same across different communities:
> $ \max\_k \\{(1/n_k)\sum_{i\in \mathcal{C}\_k}\theta_i\\} =[1+o(1)]\cdot \min\_k \\{(1/n_k)\sum_{i\in \mathcal{C}\_k}\theta_i\\}$.
> In the example above, this condition is not satisfied, so local refinement works unsatisfactorily.
> Our proposed "structural similarity" approach avoids such a condition.

---

> > ### Comment · Reviewer_1PMk · 2022-11-18
> > **Response to authors**
> >
> > Thanks for your reply. After reading the response, I would like to change the score to 6.

---

> > > ### Author Response · Authors · 2022-11-18
> > > **Thank you for appreciating our response!**
> > >
> > > We are very glad that you changed the score to a higher one. Thank you for your interesting comments, especially the pointer to literature and the suggestions to improve theory.

---

> ### Author Response · Authors · 2022-11-15
> **Response to Reviewer 1PMk (Part 2)**
>
> (Response continues here)
>
> ---
>
> **Comment**: "This classifier may be equivalent to likelihood ratio classifier ... it should achieve the information theoretical lower bound, but the authors fail to show this in Lemma 2 ... it can be shown that the exponent of total variation distance and Hellinger distance are the same up to 1+o(1); see [1] for details"
>
> **Response**: Thanks for suggesting possible ways to improve theory. However, we wish to clarify that our method is *not* equivalent to any likelihood ratio classifier.  A likelihood ratio procedure must first estimate all the nuisance parameters $\theta_i$, but our method does not need to estimate any $\theta_i$. Since our method is not likelihood-based, the constants in the upper/lower bounds in Lemma 2 are not necessarily the same. We stay valid in our arguments following Lemma 2.
>
> We also carefully checked Zhang and Zhou (2016) but did not find any claim like "the exponent of total variation distance and Hellinger distance are the same up to 1+o(1)". They studied the  Rényi divergence of order 1/2 and made a connection to the Hellinger distance (in Lemma B.1 of their supplementary material). Maybe this is what you meant? But this result is not about the total variation distance. Moreover, these results are based on stochastic block models where $\theta_i$'s are all equal to each other. They are not guaranteed to hold under the DCBM model.
>
> ---
>
> **Comment**: "The proposed method is too simple and the theoretical result cannot support the output of the algorithm can achieve the optimal error rate"
>
> **Response**: The implementation of our method is simple, but it does not mean the derivation of our method is trivial. Our goal is to have a method such that:
> - It works for both assortative and dis-assortative communities.
> - It avoids estimating any nuisance parameter in DCBM.
> - It tolerates an arbitrary permutation of the estimated communities on unlabeled nodes, without affecting the classification accuracy.
> - Its error rate matches with the information theoretic lower bound.
>
> How to find a method that simultaneously satisfies all these requirements is non-trivial. A likelihood-based procedure is usually optimal, but it is practically infeasible (because a likelihood-based procedure must first accurately estimate all nuisance parameters $\theta_i$, which is impossible in practice, especially when the network is sparse). An adaption of local refinement (as you suggested in your comment) is easy to implement, but it doest not guarantee accuracy (please see the example in our response to your first comment) and requires prior knowledge of assortativity. As far as we see, our method is the only one that satisfies all the above requirements.
>
> Regarding optimality, we already prove the "optimality in rate": Our error rate matches with that of the information theoretic lower bound, up to a constant factor in the exponent. For the "optimality in constant", it is yet unclear whether it can be achieved by any practically feasible  method. We think our theoretical result is already the best so far for semi-supervised community detection.
>
> ---
>
> **Comment**: "We can run the community detection algorithm again on the whole network."
>
> **Response**: We have already included this method in simulations. It cannot take advantage of labeled nodes and frequently  underperforms our proposed method.
>
> ---

---

### Official Review · Reviewer_VCTg · 2022-10-29

**Confidence:** 4
**Correctness:** 3
**Technical Novelty And Significance:** 3
**Empirical Novelty And Significance:** 3
**Recommendation:** 6

**Clarity, Quality, Novelty And Reproducibility:**

- It is my understanding that when references are not included as part of the sentence, it should be (Ji et al. (2022)) rather than Ji et al. (2022). However, the citation format seems to be mixed throughout the paper, such as in the first paragraph of the introduction. I encourage the authors to fix the misused reference formats.

- In the third paragraph of the introduction, two real-world examples are given to motivate the research question. Maybe add some references to support the first example just like Ji et al. (2022) supports the second example.

- I have major reservations about the experiments on real-world networks.

1) Is it a fair comparison between this model and GCN, when your setting excludes node attributes while GCN specifically focuses on propagating messages based on node attributes? Although three measures are adopted to add synthetic node attributes to do GCN (i: all 1 vectors, ii: random attributes, iii: adjacency vectors), I am not sure whether these are standard practices to use GNNs on networks with no node attributes. Propagating messages with all-1 vectors seems to be resulting in over-smoothing. Using adjacency vectors as node features means that the feature transformation linear layers' size changes with respect to the number of nodes in a network, which is not an inductive setting and could heavily overfit due to too my parameters. If these statements are true, then the consistently low performance of GCNs in Table 1 could be a result of the undesirable experiment setting, not necessarily indicating the proposed approach's superiority over GCN. Setting aside whether it's fair to compare non-attribute approaches with GCN, I would encourage the author to check out standard practices to add synthetic node attributes in GNN research.

2) Since (a) the authors acknowledged that the proposed approach could be extended to handle attributed graphs (remark 4 in Section 2.2), (b) GCNs rely on meaningful node attributes, and (c) an increasing amount of real-world networks have nodes with clear attributes or semantic information, I wonder if it might be better and fairer to evaluate on the datasets used in [1] and/or other large-scale real-world networks [2,3].

3) Let's assume that the current setup, adding synthetic features to GCN as a baseline, is fair. In that way, there seem to be more advanced semi-supervised community detection approaches based on better GNNs and node attributes [1,4,5,6]. The authors are encouraged to compare with them, or at least discuss them in the related work.

- The authors stated that "we do not think that our work has any potential negative influence" and I would like to propose one. In the third paragraph of the introduction, the authors gave an application case of how to determine the community belongings of social media users with the help of a small fraction of nodes with known community labels. Similarly, the proposed approach could be manipulated by malicious and authoritarian actors to identify dissenters on social media with "dissenter" as a community. If the authors agree, I would suggest expanding the discussion in the ethics statement.

[1] Jin, Di, et al. "Graph convolutional networks meet markov random fields: Semi-supervised community detection in attribute networks." Proceedings of the AAAI conference on artificial intelligence. Vol. 33. No. 01. 2019.

[2] Tang, Jie, et al. "Arnetminer: extraction and mining of academic social networks." Proceedings of the 14th ACM SIGKDD international conference on Knowledge discovery and data mining. 2008.

[3] Feng, Shangbin, et al. "TwiBot-22: Towards Graph-Based Twitter Bot Detection." arXiv preprint arXiv:2206.04564 (2022).

[4] Wu, Xixi, et al. "CLARE: A Semi-supervised Community Detection Algorithm." Proceedings of the 28th ACM SIGKDD Conference on Knowledge Discovery and Data Mining. 2022.

[5] Bakshi, Arjun, Srinivasan Parthasarathy, and Kannan Srinivasan. "Semi-supervised community detection using structure and size." 2018 IEEE International Conference on Data Mining (ICDM). IEEE, 2018.

[6] Li, Pan, I. Chien, and Olgica Milenkovic. "Optimizing generalized pagerank methods for seed-expansion community detection." Advances in Neural Information Processing Systems 32 (2019).

**Strength And Weaknesses:**

Strengths:
+ the proposed approach is well motivated
+ solid theoretical analysis

Weaknesses:
- major issues in the empirical experiments on real-world networks
- minor issues such as citation format and ethics statement

**Summary Of The Paper:**

This paper proposes a semi-supervised community detection method based on the idea that "a community is a group of structurally equivalent nodes". To this end, the authors propose structural similarity metrics, which enable the model to handle non-assortative networks and degree heterogeneity. Theoretical guarantees are also provided to support the proposed approach.

**Summary Of The Review:**

This paper proposes a well-grounded solution to an interesting problem with theoretical guarantees. Though I appreciate this work, I have some reservations about the empirical experiments. I look forward to discussing with the authors to revisit my review and/or score. :)

---

> ### Author Response · Authors · 2022-11-19
> **Response to Reviewer VCTg (Part 1)**
>
> Thank you for your detailed comments about GNN and pointers of recent literature. We learnt a lot from your comments.
> We are glad that you think our approach is "well-motivated" and we provide "solid theoretical analysis".
>
> ---
>
> **Our motivation of including GNN in real data analysis**:
>
> We realize that you and we are actually on the same page. It seems that your concern can be resolved by improving the writing of the paper.
>
> You and we all agree that GNN and our method target on two different scenarios:
> - *Scenario A*: There are a large number of attributes for each node, which can be used to infer community labels. The key question is how to utilize the graph to better propagate messages. This is where GNN focuses on.
> - *Scenario B*: There is no node attribute available. The key question is how to infer community labels merely from the graph structure. This is where our method focuses on.
>
> To this end, GNN and our method are not competitors of each other. We decided to include GNN in real data analysis, because we were frequently asked by other people: "How does GNN perform in Scenario B if one uses randomly generated attributes or the 1-hop representations as attributes?" The message we would like to deliver is that "GNN, with straightforward modifications, does not work well for Scenario B (a scenario where GNN is *not* designed for)".
>
> Therefore, what you pointed out seems to be an issue of writing. If we understand correctly, you do not want the readers to have the wrong impression that GNN and our method are direct competitors of each other and that our method dominates GNN. We completely agree with you. We feel that this issue can be easily resolved by clarifying our motivation of including GNN in real data analysis. In this revision, we have edited the paragraph of GNN in Section 1 to make this point clear.
>
> ---
> **Comment**: "Is it a fair comparison between this model and GCN? ... I would encourage the author to check out standard practices to add synthetic node attributes in GNN research."
>
> **Response**: As we have explained, we did not mean to use Table 1 to show our superiority over GCN. The two methods deal with different problems and are not directly comparable. However, since GNN is popular in graph data analysis and we were often asked about the comparison with GNN, we think the readers may be interested in seeing whether a naive modification of GNN works for our setting. In this revision, we have edited the text to make this point clear.
>
> We really like your insights about why the three choices of attributes do not work. We have absorbed them into the text.
>
> Following your suggestion, we searched for "standard practices of adding synthetic node attributes in GNN research". But we have not found any method for the case where there are completely no node attributes. We did find research that investigates how to "leverage long paths of graphs without over-smoothing the vertex features". It considers applying transformation to the graph or the features, but it still requires there are existing features. We tried using the landing probability (LP) in Li, Chien and Milenkovic (2019) as node attributes, as the authors argued that "LP encodes rich information regarding graph topology". We also tried using the node2Vec embeddings (Grover and Jure Leskovec, 2017) as node attributes. However, these new choices do not improve the error rates of GCN. Please see the last two columns of the extended Table 1. Our experiments suggest that GNN is generally not suitable for our setting where there are no node attributes.
>
> ---
>
> **Comment**: "Since ... the proposed approach could be extended to handle attributed graphs, I wonder if it might be better and fairer to evaluate on the datasets used in [1] and/or other large-scale real-world networks [2,3]."
>
> **Response**: Following your suggestion, we implemented an extension of our method to attributed graphs on the Citeseer data set. The reported error rate by Kipf & Welling (2016) - the original CCN approach in our experiments - is 0.297, and the reported error rate by Jin et al. (2019) - one of the references you mentioned - is 0.268. The error rate of the extension of AngleMin+ to attributed networks is 0.334.
>
> We note that the other two approaches are fine-tuned for these real networks, but for our method, we just use a very simple fusion approach to extend AngleMin+ to attributed networks. The error rate may be further improved by carefully choosing a fusion approach.  Additionally, our experiment setting is not exactly the same as those in the above papers. We consider 10 random data splits, but they only considered 1 data split. When we re-ran GCN with 10 random splits, their error rate increased from 0.297 to 0.321 and got closer to our error rate.

---

> ### Author Response · Authors · 2022-11-19
> **Response to Reviewer VCTg (Part 2)**
>
> (Response continues here)
>
> ---
>
> **Comment**: "There seem to be more advanced semi-supervised community detection approaches based on better GNNs and node attributes [1,4,5,6]. The authors are encouraged to compare with them, or at least discuss them in the related work."
>
> **Response**: Thanks for pointing out these references. We have added all of them into the paper.
>
> In this revision, we added a comparison with the GNN approach in [1] on the Citeseer data set, where we quoted their reported error rate. For the other three data sets (Caltech, Simmons, Polblogs), there were no results reported in [1]. We did not manage to find and run their code during this rebuttal period, but based on the results of GCN in Table 1, we conjecture that the approach in [1] does not work well either, as these networks have no node attributes.
>
> ---
>
> **Comment**: "When references are not included as part of the sentence, it should be (Ji et al. (2022)) rather than Ji et al. (2022) ... I encourage the authors to fix the misused reference formats"
>
> **Response**: Thanks for catching this issue. We have fixed all the misused reference formats.
>
> ---
>
> **Comment**: "Two real-world examples are given to motivate the research question. Maybe add some references to support the first example just like Ji et al. (2022) supports the second example."
>
> **Response**: This is a great idea. We have added a reference: Shapira, et al. (2013) Facebook single and cross domain data for recommendation systems.
>
> ---
>
> **Comment**: "The proposed approach could be manipulated by malicious and authoritarian actors to identify dissenters on social media with "dissenter" as a community ... I would suggest expanding the discussion in the ethics statement."
>
> **Response**: Thanks for bringing this to our attention. We have expanded the ethics statement following your suggestions.

---

> > ### Comment · Reviewer_VCTg · 2022-11-19
> > **Thank you for your response.**
> >
> > Thank you for your detailed response. Most of my concerns are adequately addressed, and I now understand better the point of bringing in GNNs in the experiments. I have revised my score to reflect that.

---

> > > ### Author Response · Authors · 2022-11-19
> > > **Thank you for appreciating our response!**
> > >
> > > We are very glad that you changed the score to a higher one. Thank you for providing the useful insights and references on GNN.
> > >
> > > An update: We further revised the paper. 1) We edited the paragraph of GNN in Section 1 to clarify why we include GNN in experiments. 2) We added the new results of GNN and rephrased the text in the section of numerical studies (in particular, we mention the insights you give about why the constant vector and adjacency vector are not good choices of features). We hope this adequately addresses your comments.

---

### Official Review · Reviewer_wECN · 2022-10-31

**Confidence:** 4
**Correctness:** 3
**Technical Novelty And Significance:** 3
**Empirical Novelty And Significance:** 3
**Recommendation:** 6

**Clarity, Quality, Novelty And Reproducibility:**

The author proposed a semi-supervised community detection method that can predict the labels for the unlabeled nodes in networks. The authors clearly described their works. The novelties of their work include a new structural similar metric and theoretical guarantees. Their efforts, in my opinion, do contribute to the development of community label prediction.

**Strength And Weaknesses:**

Strength:
(1) They propose a fast method.
(2) Their method is more generalized because their semi-supervised method models degree heterogeneity and can handle both assortative and non-assortative communities.
(3) The authors first offer theoretical guarantees on the community detection problems.

Weaknesses:
(1) In the Introduction part, The authors suppose that "the nodes partition into K non-overlapping communities". It could be clearer if they can explain how to process the unlabeled nodes. It means it is not clear how to decide the unlabeled nodes to which community C_i. I believe the way, how to initiate the unlabeled nodes, may influence their equation (4) which needs to consider the labeled and unlabeled nodes in each community.
(2) On page 3, if we look at equation (3), it would be clearer if the authors can add a notation on "𝛳/i" and "𝛳/j".
(3) The authors claimed that their method is fast. However, they didn't give an analysis of time complexity or a comparison of running time. I think it would be better if the authors can give a comparison of running time.
(4)It would be clearer if the authors can offer a pseudo code or a figure of their algorithm.


**Summary Of The Paper:**

The authors created a semi-supervised approach for the crucial and challenging subject of community detection.
To solve the problem, they are facing the following challenges: (1) Non-assortative communities exist in real networks; (2) Degree heterogeneity exists in the network; (3) Optimization-based techniques may obtain local minima; and (4) No existing solution offers theoretical assurances.
Their method can give the predicted labels of unlabeled nodes in a graph based on the structural similarity metric between the new node and the observed labeled communities. In this work, their contributions include (1)Their new semi-supervised method models degree heterogeneity; (2) The authors' method allows for both assortative and non-assortative communities; (3) The authors designed a structural similar metric to measure the similarity between a new node and the other observed communities. (4) They first showed the theoretical guarantees of their method.
Their efforts do aid in the advancement of community label prediction.

**Summary Of The Review:**

Based on the innovations in their work as well as their strengths and weaknesses. Their work is, in my opinion, above the standard for acceptance. But it falls short of becoming strongly accepted.

---

> ### Author Response · Authors · 2022-11-18
> **Response to Reviewer wECN**
>
> Thank you for a nice summary of our contributions. We are glad that you think our efforts "aid in the advancement of community label prediction".
>
> ---
>
> **Comment 1**:  “It could be clearer if they can explain… how to initiate the unlabeled nodes”
>
> **Response**: This is a great point. It was usually expected that a semi-supervised community detection algorithm would need a "good" initialization on unlabeled nodes, but our method does not need it. This is one advantage of our method.
>
> We provide two versions, AngleMin and AngleMin+. AngleMin does not need any initial estimates of the community labels of unlabeled nodes. To see this, we note that Equation (4) requires computing $X$ and $A^{(k)}$. $X$ is the $n$-dimensional vector consisting of edges between the new node and all the existing nodes. $A^{(k)}$ is an $n$-dimensional vector for community $k$, where for any $1\leq j\leq n$, the $j$th entry of $A^{(k)}$ is the total number of edges between node $j$ and all labeled nodes in community $k$; this quantity is computed without knowledge of the true community label of node $j$.
>
> AngleMin+ needs initial estimates of the community labels of unlabeled nodes, but these estimates can be rather inaccurate. In our theory, we allow the initial estimates on unlabeled nodes to have a *constant* error rate (see Definition 1 of Section 3.1). We show that AngleMin+ can achieve an $o(1)$ error rate even when these initial labels have a constant error rate. In our design of AngleMin+,  these initial labels are used to provide a "projection matrix" (please Remark 2 in our paper) but not directly used to label a new node. This is why AngleMin+ is resistant to (potential) unsatisfactory performance of unsupervised community detection on $A_{\cal UU}$.
>
> Furthermore, AngleMin+ continues to work even if we arbitrarily permute the initial estimated "communities"; e.g., "Community 1" on unlabeled nodes does not need to correspond to the true community 1. This avoids a one-to-one matching between the $K$ communities on labeled nodes and the $K$ estimated communities on unlabeled nodes (when $K$ is large, such a matching could be hard in practice).
>
> In our paper, we recommend applying an unsupervised community detection algorithm, SCORE+, on $A_{\cal UU}$ to get initiate estimates on unlabeled nodes. Our theory continues to hold if SCORE+ is replaced by any unsupervised community detection algorithm that is better than random guessing (please see Definition 1 and Corollary 1).
>
> ---
>
>
> **Comment**: “On page 3, if we look at equation (3), it would be clearer if the authors can add a notation on "𝛳/i" and "𝛳/j".”
>
> **Response**: Thank you for the suggestion. We have added extra explanations of the degree parameters $\theta_i$ and $\theta_j$ in the paragraph above Equation (3).
>
> ---
>
> **Comment**: "It would be better if the authors can give a comparison of running time”
>
> **Response**: In the revision, we have taken your advice and added the running time comparison in Section B of the appendix. For each of the three real networks, Caltech, Simons and Polblogs, we run the algorithms on many random splits into training/testing and report the average running time per data split. For example, for Polblogs ($n=1222$), the average running time of our method is $0.02$ to $0.06$ seconds, the average running time of SNMF is $0.3$ to $0.5$ seconds, and the average running time of GNN is $0.3$ to $1.5$ seconds. Recalling that our method is optimization-free, we expect the advantage of running time to be even more significant for larger networks.
>
>
> ---
>
> **Comment**: ”It would be clearer if the authors can offer a pseudo code or a figure of their algorithm.”
>
> **Response**: In the revision, we have added the pseudo code of our algorithm into Section A of the appendix. We tried hard to include it in the main paper, but there is no additional space. We hope it is okay to put it in the appendix.

---

### Decision · Program_Chairs · 2023-01-20

**Decision:**

Accept: poster

**Justification For Why Not Higher Score:**

Experimental results are interesting but they are not too strong. It is also not clear if the method could be directly applied in practice.

**Justification For Why Not Lower Score:**

Nice theoretical analysis of natural model, interesting experimental results

**Metareview: Summary, Strengths And Weaknesses:**

The authors propose new algorithms to recover community in the degree-corrected stochastic block model in the semi-supervised setting. In this setting, few labels are revealed to the algorithm and the goal is to use such labels to recover the full community structure. The authors present algorithms for this task and they study their performances both in theory and in practice showing that their method is very effective.

Overall, the result is a nice and novel contribution on a well-studied problem and it would be nice to include it in the ICLR program.

**Note From Pc:**

if the above contains the word "oral" or "spotlight" please see: "oral" presentation means -> notable-top-5% and "spotlight" means -> notable-top-25%. As stated in our emails, we are disassociating presentation type from AC recommendations